# SWE-agent: Agent-Computer Interfaces Enable Automated Software Engineering

**John Yang**\*   **Carlos E. Jimenez**\*   **Alexander Wettig**   **Kilian Lieret**

**Shunyu Yao**   **Karthik Narasimhan**   **Ofir Press**

Princeton Language and Intelligence, Princeton University

## Abstract

Language model (LM) agents are increasingly being used to automate complicated tasks in digital environments. Just as humans benefit from powerful software applications, such as integrated development environments, for complex tasks like software engineering, we posit that LM agents represent a new category of end users with their own needs and abilities, and would benefit from specially-built interfaces to the software they use. We investigate how interface design affects the performance of language model agents. As a result of this exploration, we introduce SWE-agent: a system that facilitates LM agents to autonomously use computers to solve software engineering tasks. SWE-agent's custom agent-computer interface (ACI) significantly enhances an agent's ability to create and edit code files, navigate entire repositories, and execute tests and other programs. We evaluate SWE-agent on SWE-bench and HumanEvalFix, achieving state-of-the-art performance on both with a pass@1 rate of $12.5\%$ and $87.7\%$, respectively, far exceeding the previous state-of-the-art achieved with non-interactive LMs. Finally, we provide insight on how the design of the ACI can impact agents' behavior and performance.

## 1   Introduction

Recent work has demonstrated the efficacy of LM agents for code generation with execution feedback [39]. However, applying agents to more complex code tasks like software engineering remains unexplored. To solve programming tasks, LM agents are typically designed to use existing applications, such as the Linux shell or Python interpreter [53, 57, 59]. However, to perform more complex programming tasks such as software engineering [20], human engineers benefit from sophisticated applications like VSCode with powerful tools and extensions. Inspired by human-computer interaction (HCI) studies on the efficacy of user interfaces for humans [7], we investigate whether LM agents could similarly benefit from better-designed interfaces for performing software engineering tasks.

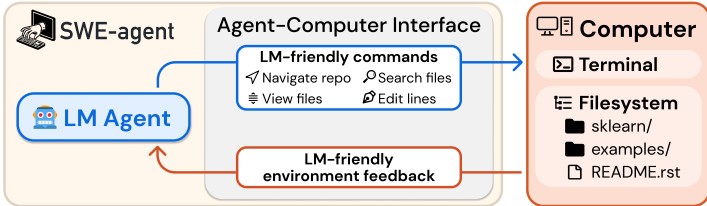

Figure 1: SWE-agent is an LM interacting with a computer through an agent-computer interface (ACI), which includes the commands the agent uses and the format of the feedback from the computer.

---

\*Equal contribution. Correspondence to johnby@stanford.edu, carlosej@princeton.edu.
Data, code, and leaderboard at swe-agent.com

38th Conference on Neural Information Processing Systems (NeurIPS 2024).

Consider the simple setting of an agent interacting directly with a Linux shell [59]. In practice, we find that LM agents can struggle to reliably take actions in this environment. For example, it fails to provide simple commands to edit a small file segment, and does not provide any feedback if the user makes an invalid edit. These deficits substantially hamper performance, motivating the need for an agent-computer interface (ACI), i.e., an abstraction layer between the LM agent and computer, to enhance the LM agent's abilities in computer environments (Figure 1).

From this effort, we introduce SWE-agent, an agent composed of an LM and ACI, that can interact with a computer to solve challenging real-world software engineering problems, such as those proposed in SWE-bench [20]. In contrast to the Linux Shell's granular, highly configurable action space, SWE-agent's ACI instead offers a small set of simple actions for viewing, searching through and editing files. The ACI uses guardrails to prevent common mistakes, and an agent receives specific, concise feedback about a command's effects at every turn. *We show that ACIs tailored specifically for LMs outperform existing user interfaces* (UIs) *designed for human users*, such as the Linux shell.

Using GPT-4 Turbo as a base LM, SWE-agent solves 12.47% of the 2,294 SWE-bench test tasks, substantially outperforming the previous best resolve rate of 3.8% by a non-interactive, retrieval-augmented system [20]. We perform an ablation study on a subset of 300 SWE-bench test instances (SWE-bench Lite) to analyze our ACI design choices. The results show that SWE-agent solves 10.7 percentage points *more* instances than the baseline agent, which uses only the default Linux shell. Although our ACI was developed for GPT-4 Turbo, we show that it is portable to a different LM; SWE-agent with Claude 3 Opus can solve 10.5% of the benchmark tasks.

Our contributions are twofold. First, we introduce the concept of the agent-computer interface (ACI) and demonstrate how careful ACI design can substantially improve LM agent performance without modifying the underlying LM's weights. Second, we build, evaluate, and open-source SWE-agent, a system that provides LMs an ACI for solving real-world software engineering tasks. Unlike prior works that independently explore the merits of tool use, prompting techniques, and code execution in interactive settings, our approach unifies these factors within the ACI framework. We show that crafting LM-centric interactive components has meaningful effects on downstream task performance.

## 2    The Agent-Computer Interface

An LM acts as an agent when it interacts with an environment by iteratively taking actions and receiving feedback [42, 62]. Typically, the environment has hard constraints, as in robotics, where agents control actuators in the physical world. On the other hand, digital environments can be molded by abstractions in the form of application programming interfaces and user interfaces for software and humans respectively. Naturally, existing interfaces have been designed with one of these users in mind. We argue that LM agents represent a new category of end user, with their own needs and abilities. We refer to the interface LM agents use to interact with computers as the *agent-computer interface* (ACI). Figure 2 illustrates how ACIs provide LM agents with important functionality to interface with computers, similar to how code editors also help humans use computers more effectively.

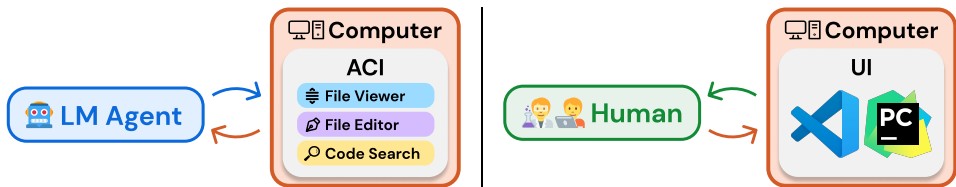

Figure 2: Specialized applications like IDEs (e.g., VSCode, PyCharm) make scientists and software engineers more efficient and effective at computer tasks. Similarly, ACI design aims to create a suitable interface that makes LM agents more effective at digital work such as software engineering.

Disparities in humans' and LMs' abilities and limitations motivates different interface design guidelines. For instance, the current generation of LMs lack the visual understanding abilities to directly operate GUI-based applications with rich visual components and signals. However, many of the features provided by these applications, such as syntax checking and navigation tools, could be useful to LM agents if they were presented in a suitable manner. Additionally, humans can flexibly ignore unnecessary information, whereas all content has a fixed cost in memory and computation for LMs

and distracting context can harm performance [27]. Therefore, LM agents may be more effective at interacting with computers when provided an interface that was built informed by these differences.

Ultimately, a well-designed ACI should help the LM agent understand the state of the application given previous changes, manage history to avoid unnecessary context from prior observations, and provide actions that models can use efficiently and reliably. The ACI specifies both the commands available to the LM and how the environment state is communicated back to the LM. It also tracks the history of all previous commands and observations and, at each step, manages how these should be formatted and combined with high-level instructions into a single input for the LM.

In this paper, we assume a fixed LM and focus on designing the ACI to improve its performance. This means that we shape the actions, their documentation, and environment feedback to complement an LM's limitations and abilities. We draw inspiration from the field of HCI, where user studies elicit insights about how compatible different interfaces are with respect to human intuition and performance [7]. We use two approaches to enhance performance on a development set: (1) manually inspect agent behavior to identify difficulties and propose improvements, and (2) run a grid search to select the best ACI configuration.

Taking these two actions resulted in several insights about design principles that seem especially important for building effective ACIs:

1. **Actions should be simple and easy to understand for agents.** Many bash commands have documentation that includes dozens of options. Simple commands with a few options and concise documentation are easier for agents to use, reducing the need for demonstrations or fine-tuning. This is a defining principle for all SWE-agent commands that we describe in Section 3.

2. **Actions should be compact and efficient.** Important operations (e.g., file navigation, editing) should be consolidated into as few actions as possible. Efficient actions help agents make meaningful progress towards a goal in a single step. A poor design would therefore have many simple actions that must be composed across multiple turns for a higher order operation to take effect. We show this idea in action in the Editing and Search interface analyses in Section 5.1.

3. **Environment feedback should be informative but concise.** High quality feedback should provide the agent with substantive information about the current environment state (and the effect of the agent's recent actions) without unnecessary details. For instance, when editing a file, updating the agent about revised content is helpful. Figures 3a, 3b and Table 3 show this.

4. **Guardrails mitigate error propagation and hasten recovery.** Like humans, LMs make mistakes when editing or searching and can struggle to recover from these errors. Building in guardrails, such as a code syntax checker that automatically detects mistakes, can help agents recognize and quickly correct errors. We show the effect of editing guardrails in Table 3.

Analysis and ablation studies in Section 5 demonstrate how alternative ACIs affect LM performance. Our studies shows how these principles appear recurrently across actions, feedback, and workflows.

## 3    SWE-agent: Designing an ACI for Software Engineering

Here we describe how SWE-agent provides an ACI for LMs to act as software engineering agents, enabling them to effectively search, navigate, edit, and execute code commands. The ACI comprises several principal components, including search/navigation, file viewer, file editor, and context management. At each step, SWE-agent generates a thought and a command, then incorporates the feedback from the command's execution in the environment (ReAct; Yao et al. [62]). Built atop the Linux shell, SWE-agent also allows access to common Linux commands and utilities when needed.

**Search and navigation.** Navigating codebases requires finding the relevant file and content. A common strategy to do this involves looking up terms that might be useful, e.g., files, functions, or class definitions mentioned in an issue. We introduce the special commands `find_file`, `search_file`, and `search_dir`, which output a summary of search results when searching for filenames and strings within files or directories. Figure 10 shows examples of these search result formats. The `find_file` command searches for filenames in the repository, while the `search_file` and `search_dir` locates strings in a file(s) of a subdirectory. Our interface encourages efficient searches by suppressing verbose results. The search commands return at most 50

```
[File: solvers/diophantine.py (3093 lines total)]
(400 lines above)
401:    elif total_degree == 3:
402:        diop_type = "cubic_thue"
403:
404:    elif (total_degree > 3):
405:        if all(coeff[k] == 1):
406:            diop_type = 'general_sum_even'
407:
408:    if diop_type is not None:
409:        return var, coeff, diop_type
(2684 lines below)
```

```
We need to add a check to ensure that all powers
are the same. Let's edit the code to include this
check.

edit 404:407
    elif (total_degree > 3 and
          len(set(k.exp[k.is_Pow])) == 1):
        if all(coeff[k] == 1):
            diop_type = 'general_sum_even'
end_of_edit
```

(a) Observation from the file viewer.      (b) Action using the edit interface.

Figure 3: The file viewer and edit command are integrated. (a) The file viewer shows the agent the open file's content with line numbers. (b) The agent invokes the edit function to replace lines 404-407 in the open file. After the edit, the file viewer shows the agent the now updated version of the file.

results for each search query; if a search exceeds this number, we do not report the results and instead suggest that the agent write a more specific query.

**File viewer.** After finding a file they want to view, agents use the interactive file viewer by calling the command `open` on the relevant file path. The file viewer presents a window of at most 100 lines of the file at a time. The agent can move this window with the commands `scroll_down` and `scroll_up` or access a specific line with the `goto` command. To facilitate in-file navigation and code localization, we display: the full path of the open file, the total number of lines in the file, the number of lines omitted before and after the current window, and the line number (prepended to each visible line). Figure 3a shows an example of this interface.

**File editor.** We provide a few commands that let LMs create and edit files. The `edit` command works in conjunction with the file viewer, allowing agents to replace a specific range of lines in the open file. This command takes 3 required arguments: the start line, end line, and replacement text. In a single step, agents can replace all lines between the start and end lines with the replacement text, as shown in Figure 3b. After edits are applied, the file viewer automatically displays the updated content, helping the agent observe the effects of its edit immediately without invoking additional commands. Figure 3b shows an example agent response, including a file edit.

Similar to how humans can use tools like syntax highlighting to help them notice format errors when editing files in an IDE, we integrate a code linter into the `edit` function to alert the agent of mistakes it may have introduced when editing a file. Select errors from the linter are shown to the agent along with a snippet of the file contents before/after the error was introduced. Invalid edits are discarded, and the agent is asked to try editing the file again.

**Context management.** The SWE-agent system uses informative prompts, error messages, and history processors to keep agent context concise and informative. Agents receive instructions, documentation, and demonstrations on the correct use of bash and ACI commands. At each step, the system instructs them to generate both a *thought* and an *action* [62]. Malformed generations trigger an error response, shown in Figure 32, asking the agent to try again, which is repeated until a valid generation is received. Once received, all past error messages except the first are omitted.

The agent's environment responses display computer output using the template shown in Figure 30; however, if no output is generated, a specific message ("Your command ran successfully and did not produce any output") is included to enhance clarity. To further improve context relevance, observations preceding the last 5 are each collapsed into a single line, shown in Figure 31. By removing most content from prior observations, we maintain essential information about the plan and action history while reducing unnecessary context, which allows for more interaction cycles and avoids showing outdated file information. §A provides further implementation details.

## 4 Experimental Setup

**Datasets.** We primarily evaluate on the SWE-bench dataset, which includes 2,294 task instances from 12 different repositories of popular Python packages [20]. We report our main agent results on the full SWE-bench test set and ablations and analysis on the SWE-bench Lite test set, unless

otherwise specified. SWE-bench Lite is a canonical subset of 300 instances from SWE-bench that focus on evaluating self-contained functional bug fixes. We also test SWE-agent's basic code editing abilities with HumanEvalFix, a short-form code debugging benchmark [32].

**Models.** All results, ablations, and analyses are based on two leading LMs, GPT-4 Turbo (`gpt-4-1106-preview`) [34] and Claude 3 Opus (`claude-3-opus-20240229`) [6]. We experimented with a number of additional closed and open source models, including Llama 3 and DeepSeek Coder [14], but found their performance in the agent setting to be subpar. Many LMs' context window is too small, such as Llama 3's context window of 8k. GPT-4 Turbo and Claude 3 Opus have 128k and 200k token context windows, respectively, which provides sufficient room for the LM to interact for several turns after being fed the system prompt, issue description, and optionally, a demonstration.

**Baselines.** We compare SWE-agent to two baselines. The first setting is the non-interactive, retrieval-augmented generation (RAG) baselines established in Jimenez et al. [20]. Here, a BM25 retrieval system retrieves the most relevant codebase files using the issue as the query; given these files, the model is asked to directly generate a patch file that resolves the issue.

The second setting, called Shell-only, is adapted from the interactive coding framework introduced in Yang et al. [59]. Following the InterCode environment, this baseline system asks the LM to resolve the issue by interacting with a shell process on Linux. Like SWE-agent, model prediction is generated automatically based on the final state of the codebase after interaction.

**Metrics.** We report **% Resolved** or **pass**@1 as the main metric, which is the proportion of instances for which all tests pass successfully after the model generated patch is applied to the repository [20]. We also report the **$ Avg. Cost** metric, the API inference cost incurred by SWE-agent averaged over all successfully resolved instances. Due to budget constraints, we set the per-instance budget to $4; if a run exceeded this budget, existing edits were submitted automatically.

**Configuration search.** During the design process of SWE-agent, we arrived at the final ACI design through qualitative analysis of system behavior on a small set of hand-picked examples from the development split of SWE-bench. For the remaining hyperparameter choices, we performed a sweep over the window size, history processing, and decoding temperature, shown in §B.1.

## 5   Results

Across all systems, SWE-agent w/ GPT-4 Turbo achieves the best performance all-around, successfully solving 12.47% (286/2,294) of the full SWE-bench test set and 18.00% (54/300) of the Lite split. As shown in Table 1, compared to RAG on Lite, SWE-agent is 8-13x more costly but yields a 6.7-fold improved % Resolved rate. An LM-friendly ACI's value is confirmed by SWE-agent's 64% relative increase compared to Shell-only, both with GPT-4 Turbo.

In Table 2, SWE-agent yields strong performance on HumanEvalFix with 88.3% pass@1 rate. Figure 4 reveals that average performance variance is relatively low, but per-instance resolution can change considerably. More results are given in the appendix: §B.2 shows that the success rate is uncorrelated to the issue age (controlling for possible test pollution), B.5 presents more details on performance variance and pass@$k$, and B.7 discusses extra evaluation details.

### 5.1   Analysis of ACI Design

We perform several ablations of the SWE-agent interface, specifically with respect to the SWE-agent w/ GPT-4 configuration, summarized in Table 3. Our case studies shed light on interesting agent behavior along with the impact of different ACI designs.

**Human user interfaces are not always suitable as agent-computer interfaces.** Current LMs are vulnerable to a number of pitfalls when searching for relevant content in a Linux shell environment. Some exploration patterns (e.g., chains of `cd`, `ls`, `cat`) are extremely inefficient. `grep` or `find` look ups can perform better but occasionally produce many lines of irrelevant results. We hypothesize that better localization is possible with faster navigation and a more informative search interface.

---

https://github.com/meta-llama/llama3

Token counts for different models are not directly comparable since they use different tokenizers.

Table 1: Main results for SWE-agent performance on the full and Lite splits of the SWE-bench test set. We benchmark models in the SWE-agent, Basic CLI, and Retrieval Augmented Generation (RAG) settings established in SWE-bench [20].

| | SWE-bench | | SWE-bench Lite | |
|---|---|---|---|---|
| Model | % Resolved | $ Avg. Cost | % Resolved | $ Avg. Cost |
| RAG | | | | |
|   w/ GPT-4 Turbo | 1.31 | 0.13 | 2.67 | 0.13 |
|   w/ Claude 3 Opus | 3.79 | 0.25 | 4.33 | 0.25 |
| Shell-only agent | | | | |
|   w/ GPT-4 Turbo | - | - | 11.00 | 1.46 |
|     w/o Demonstration | - | - | 7.33 | 0.79 |
| SWE-agent | | | | |
|   w/ GPT-4 Turbo | **12.47** | 1.59 | **18.00** | 1.67 |
|   w/ Claude 3 Opus | 10.46 | 2.59 | 13.00 | 2.18 |

Table 2: Pass@1 results on HumanEvalFix [32]. Except for SWE-agent, we use scores as reported in Yu et al. [65].

| Model | Python | JS | Java |
|---|---|---|---|
| CodeLLaMa-instruct-13B | 29.2 | 19.5 | 32.3 |
| GPT-4 | 47.0 | 48.2 | 50.0 |
| DeepseekCoder-CodeAlpaca-6.7B | 49.4 | 51.8 | 45.1 |
| WaveCoder-DS-6.7B | 57.9 | 52.4 | 57.3 |
| SWE-agent w/ GPT-4 Turbo | **87.7** | **89.7** | **87.9** |

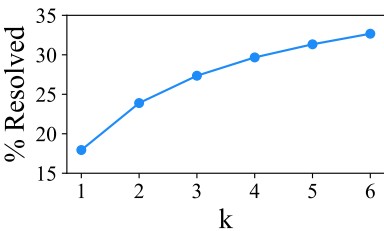

Figure 4: SWE-agent w/ GPT-4 Turbo Pass@$k$ performance across 6 runs on SWE-bench Lite.

Table 3: SWE-bench Lite performance under ablations to the SWE-agent interface, which is denoted by 😈. We consider different approaches to searching and editing (see Figures 5 and 6, respectively). We also verify how varying the file viewer window size affects performance, and we ablate the effect of different context management approaches.

| **Editor** | | **Search** | | **File Viewer** | | **Context** | |
|---|---|---|---|---|---|---|---|
| `edit` action | 15.0 $_{\downarrow 3.0}$ | Summarized 😈 | 18.0 | 30 lines | 14.3 $_{\downarrow 3.7}$ | Last 5 Obs. 😈 | 18.0 |
| w/ linting 😈 | 18.0 | Iterative | 12.0 $_{\downarrow 6.0}$ | 100 lines 😈 | 18.0 | Full history | 15.0 $_{\downarrow 3.0}$ |
| No `edit` | 10.3 $_{\downarrow 7.7}$ | No search | 15.7 $_{\downarrow 2.3}$ | Full file | 12.7 $_{\downarrow 5.3}$ | w/o demo. | 16.3 $_{\downarrow 1.7}$ |

Figure 5 compares the Shell-only setting to two different search interfaces. *Iterative* search, directly inspired by traditional user interfaces for search, e.g., Vim or VSCode, shows results one by one via the file viewer. Agents can look through results using `next` and `prev` actions. Each result displays the matching line along with n surrounding lines of context. An advantage is that an agent can begin editing directly after seeing the relevant code in its search. However, when given a large number of search results, agents tend to look through every match exhaustively, calling `next` until each result has been inspected. This inefficient behavior can exhaust an agent's cost budget or context window, leading to even worse performance than the not having additional search tools at all (15.7% $_{\downarrow 2.3}$ for No search vs. 12.0% $_{\downarrow 6.0}$ with Iterative search).

**Compact, efficient file editing is critical to performance.** SWE-agent's file editor and viewer are designed to consolidate the editing process into a single command that enables easy multi-line edits with consistent feedback and automatically updates the agent's view of the file after editing. In the No `edit` setting, editing options are restrictive and prone to errors; the primary methods available are either replacing entire files through redirection and overwriting or using utilities like `sed` for single-line or search-and-replace edits. Both methods have significant drawbacks. Redirection involves copying and rewriting entire files for even minor changes, which is both inefficient and error-prone. Although `sed` can facilitate specific edits, executing multi-line edits is cumbersome and can lead to unintended consequences that are challenging to detect. Moreover, both strategies

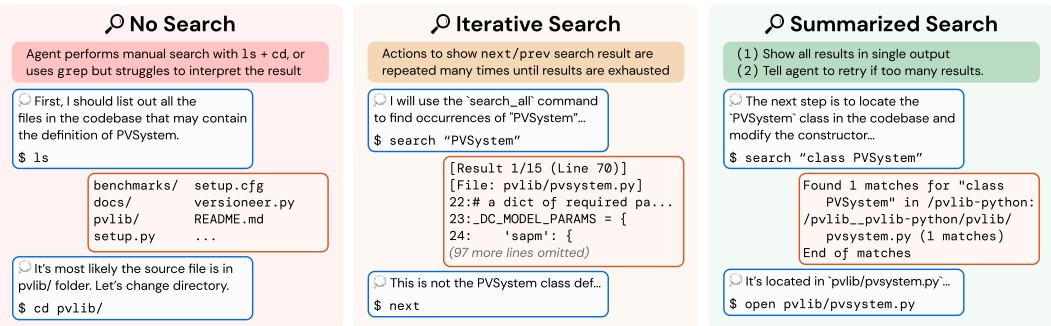

Figure 5: Three different Search interfaces for task instance `pvlib__pvlib-python-1224`. In Shell-only, an agent performs localization using only standard bash commands and utilities. Compared to *Iterative* search, *Summarized* search shows an exhaustive list of search results and provides guidance on refining under-specified queries.

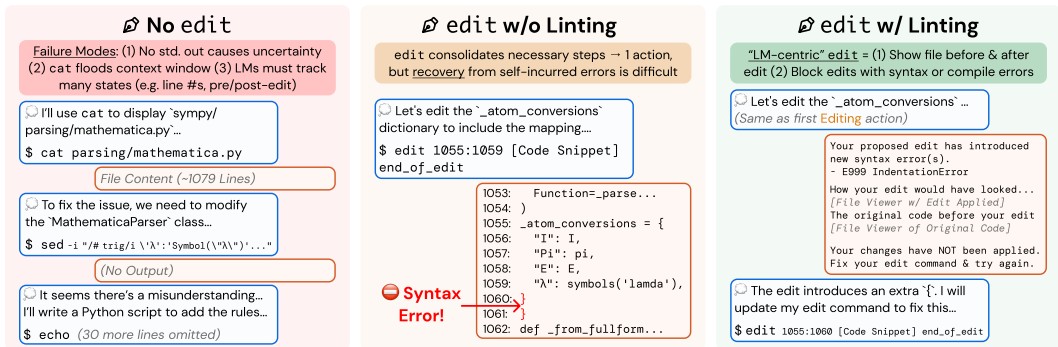

Figure 6: Three different Edit interfaces for task instance `sympy__sympy-24102`. Editing with bash commands requires several actions to successfully modify a file. The *Editing* component defines an `edit` command that leverages the File Viewer component to replace the bash style of editing workflow with a single command. *Linting* is beneficial for stymieing cascading errors that often start with an error-introducing edit by the agent.

lack immediate feedback about file updates, making these silent operations potentially confusing for models to interpret and increasing the risk of errors. Without SWE-agent's file editor interface, performance drops to (10.3% $_{\downarrow 7.7}$). We also find that agents are sensitive to the number of lines the file viewer displays. Either too little content (30 lines, 14.3% $_{\downarrow 3.7}$) or too much (entire file, 12.7% $_{\downarrow 5.3}$) lowers performance.

**Guardrails can improve error recovery.** A prominent failure mode occurs when models repeatedly `edit` the same code snippet. The usual suspect for this behavior is an agent introducing a syntax error (e.g., incorrect indentation, extra parenthesis) via an errant `edit`. As discussed in Section 3, we add an intervention to the `edit` logic that lets a modification apply only if it does not produce major errors. We compare this interface with the No `edit` and `edit` w/o linting alternatives in Figure 6. This intervention improves performance considerably (without linting, 15.0% $_{\downarrow 3.0}$).

## 5.2 Analysis of Agent Behavior

Recurring problem-solving patterns emerge when LMs are equipped with a useful, intuitive ACI. We describe several model behaviors and problem-solving patterns that can be discerned from model performance and each model's corresponding trajectories.

**Reproduction and/or localization is the first step.** SWE-agent usually begins with either writing reproduction code and/or localizing the issue's cause to specific lines of code. As shown in Figure 7, all trajectories begin with either `create` (reproduction) or `find_file`/`search_dir` (localization). To reproduce, models will `create` a new file, add reproduction code to it with an `edit`, then run with `python`; this is the most popular triple of actions in Table 8. Using this feedback along with file

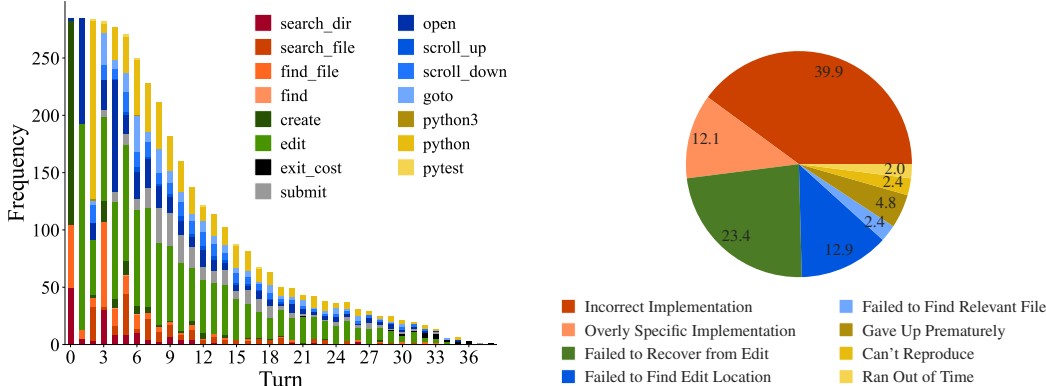

Figure 7: The frequency with which actions are invoked at each turn by SWE-agent w/ GPT-4 for task instances that it solved on the SWE-bench full test set (286 trajectories).

Figure 8: Failure mode distribution for SWE-agent w/ GPT-4 Turbo trajectories of unresolved instances. Each instance is labeled automatically using an LM with the categories from Table 9.

names and symbols in the issue description, an agent will start with a broad, directory-level keyword search, before then zooming into specific files and lines. This is reflected in Figure 22, where the most likely actions following localization sequences like (python, find_file) and (search_dir, open) are search_file and goto, indicative of how an agent "zooms in" on a bug. Extensive analysis on correlations between different groups of actions are discussed in §B.3.3

**Remaining turns are mostly "edit, then execute" loops.** As exhibited in Figure 7, from turn 5 onwards, the most frequent two actions for all turns are edit and python. Captured as high probability next actions following (edit, python) in Figure 22, additional localization operations are often interspersed across these later turns, where agents might look at more in-file code with search_file, scroll_up/down, or other files altogether with search_dir, find_file. This behavior usually arises in response to new information from re-running the reproduction script. Submissions are distributed normally from turn 10 onwards, although resolved task instances correlate more with earlier submits (see §B.3.1). A walk-through of common trajectory phases is in §B.3.2.

**Editing remains challenging for agents.** A non-trivial minority of edit actions raise a linting error; out of 2,294 task instances, 1,185 (51.7%) of SWE-agent w/ GPT-4 Turbo trajectories have 1+ failed edits. While agents generally recover more often than not from failed edits, the odds of recovery decrease as the agent accumulates more failed edits. Recovery refers to a sequence of consecutive failed edits followed immediately by a successful edit. Any attempt at editing has a 90.5% chance of eventually being successful. This probability drops off to 57.2% after a single failed edit. More editing phenomena are discussed in §B.3.3, and data about agents' generated fixes are in §B.6.

**Agents succeed quickly and fail slowly.** We find that runs submitted relatively early are much more likely to be successful compared to those submitted after a larger number of steps or cost. We show in Table 15 the distribution of resolved and unresolved instances, including only instances that did not exhaust their budget. We observe that successful runs complete earlier and at a cheaper cost than unsuccessful ones. In general, successful instances solved by SWE-agent w/ GPT 4 finish with a median cost of $1.21 and 12 steps compared to a mean of $2.52 and 21 steps for unsuccessful ones. Furthermore, we find that 93.0% of resolved instances are submitted before exhausting their cost budget, compared to 69.0% of instances overall. For these reasons, we suspect that increasing the maximum budget or token limit are unlikely to substantially increase performance. More statistics about how trajectories typically conclude are in §B.9.

**Most failures are incorrect implementations.** We use GPT-4o to automatically categorize unresolved trajectories (SWE-agent w/ GPT-4 Turbo on SWE-bench Lite, $n = 248$) into one of 9 manually defined categories described in Table 9. On a hand-labeled validation set, the LM's judgment agrees with the authors' on 87% of instances. From Figure 8, about half (52.0%) of unresolved instances fall into the Incorrect Implementation or Overly Specific Implementation categories, suggesting that agents' proposed solutions often simply fail to functionally address the issue or are insufficiently general solutions. Cascading failed edits make up another 23.4% of failures. More details in §B.4.

# 6 Related Work

## 6.1 Software Engineering Benchmarks

Code generation benchmarks, which evaluate models on the task of synthesizing code from natural language descriptions, have served as a long-standing bellwether for measuring LM performance [5, 1, 15, 30]. Subsequent works have built upon the code generation task formulation to contribute new benchmarks that translate problems to different (programming) languages [3, 49], incorporate third-party libraries [25, 29], introduce derivative code completion tasks [18, 32], increase test coverage [26], change the edit scope [8, 9, 64], and add robustness to dataset contamination [19]. Code generation problems are largely self-contained, with short problem descriptions (∼100 lines) and corresponding solutions that are similarly brief, requiring nothing more complex than basic language primitives. Tests are either handwritten or generated synthetically via fuzz testing. In recent months, the rapid development of LMs has begun to saturate many of these benchmarks. For instance, the top method solves $94.4\%$ of HumanEval [70].

Gauging future trends with the code generation task paradigm can be limited by the simplicity of this setting and cost of human-in-the-loop problem creation. In response, recent efforts have demonstrated that software engineering (SE) can serve as a diverse, challenging testbed for LM evaluation [68, 20, 28]. Repository-level code editing introduces many reasoning challenges grounded in real SE subtasks, such as spotting errant code and identifying cross-file relationships and understanding codebase-specific symbols and conventions. As a field, SE has generally studied tasks in a more isolated manner; prior benchmarks tended to frame problems in isolation from the rest of a codebase [21, 23].

We use SWE-bench because it unites many separate SE tasks, such as automated program repair [10, 40, 55], bug localization [4, 58], and testing [22, 46, 56] under a single task formulation that faithfully mirrors practical SE. Furthermore, SWE-bench task instances are diverse, having been automatically collected from real GitHub issues across 12 different repositories. In addition, SWE-bench performance is based on rigorous, execution-based evaluation with human-written unit tests.

## 6.2 Language Models as Agents

The co-emergence of stronger LMs, increasingly challenging benchmarks, and practical use cases have together motivated a paradigm shift in LMs' inference setting. Instead of traditional zero/few-shot generation, LM agents [17, 42, 47, 54] that interact with a real/virtual world have proliferated as the default setting for web navigation [24, 33, 36, 41, 45, 61, 62, 71], computer control [35, 53, 57], and code generation tasks [16, 50, 63].

Interaction and code generation are increasingly used together, with code as the modality of choice for actions [48, 59], tool construction [13, 51, 69], and reasoning [39, 66, 67]. Coding agents have also been applied to offensive security [11, 37, 60], theorem proving [44], and clinical tasks [38, 43, 52]. To the best of our knowledge, SWE-agent is the first work to explore language agents for end-to-end software engineering (SE).

# 7 Discussion

We introduce SWE-agent, an agent composed of an LM and ACI capable of autonomously solving software engineering tasks. Through our design methodology, results, and analysis, we demonstrate the value of ACIs tailored to leverage LMs' strengths and mitigate their weaknesses. Beyond empirical applications, we hope the further study of ACIs can also make principled use of and contribute to our understanding of language models and agents, analogous to the synergy between human-computer interaction (HCI) and psychology [2]. Humans and LMs have different characteristics, training objectives, specialities, and limitations [12, 31], and the interaction design processes can be seen as systematic behavioral experimentation that could reveal more insights into these differences towards establishing a comparative understanding of human and artificial intelligence.

## Acknowledgements

We thank Austin W. Hanjie, Sam Ainsworth, Xindi Wu, Yuhan Liu, Mengzhou Xia, Dan Friedman, Tianyu Gao, Adithya Bhaskar, Aatmik Gupta, Louisa Nyhus, Alisa Liu, Ori Yoran and Richard Zhu for their valuable feedback and advice. We would also like to thank the broader Princeton Language and Intelligence community for supporting our work. We acknowledge support from an Oracle Collaborative Research award and the National Science Foundation under Grant No. 2239363. Any opinions, findings, conclusions, or recommendations expressed in this material are those of the author(s) and do not necessarily reflect the views of the National Science Foundation

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

# Appendix

In the appendix, we provide additional analyses and more extensive discussions about SWE-agent, agent-computer interface (ACI) design, and model performance on various evaluation benchmarks. We also provide several thorough case studies of SWE-agent behavior on select task instances. Data, code, and leaderboard at swe-agent.com.

# A    SWE-agent Design

In this section, we go into greater discussion about the design methodology, appearance, and implementation of each of the SWE-agent components. As described in Section 3, the SWE-agent interface consists of several components that enable agents to accomplish key sub-tasks that are fundamental to solving software engineering problems. These are generally the following:

1. *Localization*: Identify file(s)/line(s) causing the issue.
2. *Editing*: Generate fixes addressing the given issue.
3. *Testing*: Write new scripts or modify existing test files to reproduce the issue and/or verify if fixes are correct.

To enable LM-based agents to efficiently carry out these individual functions and progress towards the overarching goal of resolving a codebase issue, we provide a file viewer, file editor, search / navigation system, and context management system. In Section A.1, we provide a thorough breakdown of each of these components. In Section A.2, we discuss the technical design decisions and challenges of building SWE-agent. In Section A.3, we discuss how SWE-agent is configured to support the final interface, along with how SWE-agent is built to enable easy extensibility and customization to alter the interface.

## A.1    ACI Design

In this section, we revisit each component discussed in Section 3. Per section, we first briefly review the component. We then discuss the underlying motivation for the component with respect to existing software tools. Finally, we note any additional thoughts that influenced the design process of the component with some occasional discussion of what aspects of the component heavily influence language model behavior.

Figure 9: An overview over the structure of a trajectory: We first present the system prompt, demonstration (optional), and issue statement. The agent then interacts in turn with the environment. Past observations may be *collapsed*, i.e. we truncate any long output, as described in Section 3.

For a quick, text-free overview, comprehensive documentation for all commands, their usage, and docstrings are included in Table 4. Figure 9 visualizes the message history for SWE-agent. Each prompt template is discussed thoroughly in Section C.

**File viewer.** As discussed in Section 3, the File Viewer is fundamental to a language agent's ability to understand file content and understand how different programmatic entities relate to one another. The File Viewer refers to an interface that consists of the four commands, as shown in Table 4, and a customized standard output for displaying n lines of a file at a time. Using the file viewer, an agent can look at n lines of a file at a time and jump around the file. The File Viewer enables agents to perform fine-grained localization steps and also understand relationships between intra-file entities.

First, we discuss why existing software systems and graphical user interfaces are sub-optimal for LM use. In a Shell-only setting, there are several commands that can be used to inspect file content. However, out of the box command line tools are sub-optimal or limiting for language agents for

Table 4: In additional to the standard Linux Bash commands, we provide SWE-agent with specialized tools, including an interactive file viewer, search functionalities, and edit tools for the open file. Required arguments are enclosed in <> and optional arguments are in []. The last column shows the documentation presented to the LM.

| Category | Command | Documentation |
|---|---|---|
| *File viewer* | **open** `<path>` `[<line_number>]` | Opens the file at the given path in the editor. If `line_number` is provided, the window will move to include that line. |
| | **goto** `<line_number>` | Moves the window to show line_number. |
| | **scroll_down** | Moves the window up 100 lines. |
| | **scroll_up** | Moves the window down 100 lines. |
| *Search tools* | **search_file** `<search_term>` `[<file>]` | Searches for `search_term` in file. If file is not provided, searches in the current open file. |
| | **search_dir** `<search_term>` `[<dir>]` | Searches for `search_term` in all files in dir. If dir is not provided, searches in the current directory. |
| | **find_file** `<file_name>` `[<dir>]` | Finds all files with the given name in dir. If dir is not provided, searches in the current directory. |
| *File editing* | **edit** `<n>:<m>` `<replacement_text>` **end_of_edit** | Replaces lines n through m (inclusive) with the given text in the open file. All of the `replacement_text` will be entered, so make sure your indentation is formatted properly. Python files will be checked for syntax errors after the edit. If an error is found, the edit will not be executed. Reading the error message and modifying your command is recommended as issuing the same command will return the same error. |
| | **create** `<filename>` | Creates and opens a new file with the given name. |
| *Task* | **submit** | Generates and submits the patch from all previous edits and closes the shell. |

several reasons. First, commands that print files to standard output (e.g. `cat`, `printf`) can easily flood a language agent's context window with too much file content, the majority of which is usually irrelevant to the issue. Enabling a language agent to filter out distractions and focus on relevant code snippets is crucial to generating effective edits. While commands like `head` and `tail` reduce length to the first/last `n` lines, it is not intuitive to use bash commands to perform in-file navigation. It is either impossible or requires a long list of arguments to show specific file lines. Furthermore, since such Bash commands are stateless, "scrolling" up/down relative to the current file position typically requires regenerating the same lengthy command with minor changes. Interactive tools like `more` and `less` accommodate this, but (1) representing navigation actions (multiple key up/down clicks) is intuitive for humans, but is verbose and costly for language agents, and (2) even if jumping to a specific line number is allowed, it is not possible to quickly identify what classes/methods/symbols are declared in a file and then immediately go to their definitions.

There are a couple features of the File Viewer interface that make it friendlier and more operable than the Shell-only setting. First, the File Viewer standard output contextualizes code snippets with prepended line numbers and indicators of the number of lines above/below the current region. These details give a more focused view of a file without compromising easy viewing of other parts of the codebase. This kind of file presentation also makes precise and consistent editing commands possible, as we discuss more thoroughly in the following section.

**File Viewer**

```
open atmosphere.py
```

```
(255 more lines above)
256:def gueymard94_pw(temp_air, relative_humidity):
257:    r"""
258:    Calculates precipitable water (cm) from ambient air temperature (C)
259:    and relatively humidity (%) using an empirical model. The
260:    accuracy of this method is approximately 20% for moderate PW (1-3
261:    cm) and less accurate otherwise.
262:
263:    The model was developed by expanding Eq. 1 in [2]_:
264:
265:    .. math::
266:
267:        Pw = 0.1 H_v \rho_v

                        (76 lines omitted)

343:
344:
345:def bird_hulstrom80_aod_bb(aod380, aod500):
346:    """
347:    Approximate broadband aerosol optical depth.
348:
349:    Bird and Hulstrom developed a correlation for broadband aerosol
350:    depth (AOD) using two wavelengths, 380 nm and 500 nm.
351:
352:    Parameters
353:    ----------
354:    aod380 : numeric
355:        AOD measured at 380 nm. [unitless]
(180 more lines below)
```

**Find Files**

```
find_file atmosphere.py
```

```
Found 2 matches for "atmosphere.py" in /pvlib__pvlib-python/pvlib:
/pvlib__pvlib-python/pvlib/tests/atmosphere.py
/pvlib__pvlib-python/pvlib/atmosphere.py
```

**Search Directory**

```
search_dir APPARENT_ZENITH_MODELS
```

```
Found 4 matches for "APPARENT_ZENITH_MODELS" in /pvlib__pvlib-
python:
/pvlib__pvlib-python/docs/sphinx/source/whatsnew/v0.3.0.txt (1
matches)
/pvlib__pvlib-python/pvlib/atmosphere.py (2 matches)
/pvlib__pvlib-python/pvlib/location.py (1 matches)
End of matches for "APPARENT_ZENITH_MODELS" in /pvlib__pvlib-
python
```

**Search File**

```
search_file APPARENT_ZENITH_MODELS
```

```
Found 2 matches for "APPARENT_ZENITH_MODELS" in /pvlib__pvlib-
python/pvlib/atmosphere.py:
Line 12:APPARENT_ZENITH_MODELS = ('simple', 'kasten1966',
'kastenyoung1989',
Line 15:AIRMASS_MODELS = APPARENT_ZENITH_MODELS +
TRUE_ZENITH_MODELS
End of matches for "APPARENT_ZENITH_MODELS" in /pvlib__pvlib-
python/pvlib/atmosphere.py
```

Figure 10: The File Viewer and Search components of the SWE-agent interface. The corresponding commands for each component are shown in blue. These examples are copied from trajectories generated by SWE-agent w/ GPT-4 Turbo on the `pvlib__pvlib-python-1603` task instance.

Another advantage of the File Viewer is that the commands are designed to be complementary and grounded in the File Viewer standard output. This saves the model from having to do repetitive or additional actions that unnecessarily increase the potential for error. As a concrete example, if an agent used a `sed` command to view the first 100 lines of a file and wants to look at the next 100 lines, it will have to recalculate parameters such as the start line and end line and reflect these updates correctly in the subsequent generation. As a rule of thumb, reducing the need for models to do this arithmetic by constructing actions and standard output that complement one another and build upon the effects of prior actions is highly preferable.

**File editor.**   The File Editor, working in conjunction with the File Viewer, primarily refers to the `edit` command and the guardrails it enforces to protect models against self-incurred cascading edit errors. Editing and testing are crucial to language agents' success on programming tasks, and a well-designed interface directly influences how well an agent's capabilities can be elicited. In other words, a bad interface undermines model performance.

As discussed in Section 3, editing can be very difficult in a Shell-only setting. Built in commands (e.g., `sed`) often require a lengthy list of arguments, and the mis-specification of an argument can easily throw a model off track as it attempts to correct self-incurred errors. We also observe that when agents use such commands directly, they struggle with the arithmetic skills required to generate an edit. Details such as including the correct indentation level, inserting delimiters at specific points in a line, and adhering to stylistic preferences of the codebase all require some amount of planning or calculation. Similar to the Shell-only file viewing process, file editing may also require repeating many commands. For instance, performing a multi-line edit can only be represented as multiple `sed` calls with requisite, delicate tweaks to the arguments for every turn. Furthermore, as referenced in Section 5.1, editing in Shell-only is usually a "silent" procedure. Confirming whether an edit succeeded and viewing its effects requires additional steps that can bloat the editing process with extra, needless commands.

The `edit` command, documented in Table 4, addresses the Shell-only failure modes by being grounded in the File Viewer standard output. The line numbers argument eliminates the need for any additional arithmetic, and the find-and-replace edit mechanism is a format that existing models are more used to. With this functionality, agents can also perform multi-line edits in a single action.

Finally, as mentioned in Section 5.2, an important feature of the `edit` command is that it does not apply changes which incur a linting error. A fair and verified assumption we make when considering this feature is that the original codebase associated with each task instance is well-formed. In other

words, we assume that codebase maintainers will only push syntactically sound code that can be compiled successfully. When an agent issues an edit, it is applied to the codebase. Then, we run the following linting command (CURRENT_FILE refers to the file that is currently open):

```
flake8 --isolated --select=F821,F822,F831,E111,E112,E113,E999,E902
"$CURRENT_FILE" 2>&1
```

The arguments for `select` are error codes that refer to syntax issues such as indentation. F821 and F822 indicate undefined names/symbols. F831 indicates a duplicate argument in a function definition. E111, E112, E113 are indentation errors. E999 denotes a syntax error and an E902 occurs if flake8 cannot read the source file.

If the edit does not introduce any of these errors, this command will produce no output. The edit is kept and the updated file content is shown using the File Viewer centered around the lines the edit occurred. If however the linting command produces output, which indicates the edit introduces a syntax error, the edit is reverted. In place of the file viewer, a message shown in Figure 11 is displayed to the agent which shows the error that was caused, what the edit would have looked like, and the original file content. During the development process, we experimented with variations to this message, including the omission of one or more parts. Our takeaway was that having all three messages is helpful. Without the error type, the agent might misdiagnose what the mistake was. Without a snippet of the changed file content, the agent will re-issue the same command more frequently. Without a snippet of the original file content, the agent has to attend to the same content from several turns ago; agents also sometimes generate `edit`'s with respect to wrong, errant file content because it is from a more recent turn.

---

**Linting Error Message**

```
Your proposed edit has introduced new syntax error(s). Please
↪   understand the fixes and retry your edit commmand.

ERRORS:
- F821 undefined name 'orientation_strategy'
- F821 undefined name 'orientation_strategy'

This is how your edit would have looked if applied
-------------------------------------------------
[File: /pvlib__pvlib-python/pvlib/modelchain.py (1890 lines total)]
(64 more lines above)
65:def basic_chain(times, latitude, longitude,
66:                module_parameters, temperature_model_parameters,
67:                inverter_parameters,
68:                irradiance=None, weather=None,
69:                surface_tilt=None, surface_azimuth=None,
70:                transposition_model='haydavies',
71:                solar_position_method='nrel_numpy',
72:                airmass_model='kastenyoung1989',
73:                altitude=None, pressure=None,
74:                **kwargs):
(1816 more lines below)
-------------------------------------------------

This is the original code before your edit
-------------------------------------------------
[File: /pvlib__pvlib-python/pvlib/modelchain.py (1891 lines total)]
(64 more lines above)
65:def basic_chain(times, latitude, longitude,
66:                module_parameters, temperature_model_parameters,
67:                inverter_parameters,
68:                irradiance=None, weather=None,
69:                surface_tilt=None, surface_azimuth=None,
70:                orientation_strategy=None,
71:                transposition_model='haydavies',
```

```
72:                solar_position_method='nrel_numpy',
73:                airmass_model='kastenyoung1989',
74:                altitude=None, pressure=None,
75:                **kwargs):
(1816 more lines below)
------------------------------------------------
Your changes have NOT been applied. Please fix your edit command and
↪  try again.
You either need to 1) Specify the correct start/end line arguments or
↪  2) Correct your edit code.
DO NOT re-run the same failed edit command. Running it again will
↪  lead to the same error.
```

Figure 11: A linting error message. This is emitted if a model generates an `edit` command that introduces a syntax error into the codebase. The error message shows the before and after of the proposed edit along with what error messages were thrown. The problem with this edit is that it omits the `orientation_strategy` field in its edit of the `basic_chain` method definition.

The editing guardrail has a drawback. To a certain degree, it forces some edits to be done in a particular order. For instance, in Figure 11, if the model's intention was in fact to remove the `orientation_strategy` argument, due to the SWE-agent editing guardrails, it would have to remove all references from the function implementation either at the same time in a single action, or before removing it from the method header if split into two separate actions. For this particular scenario, the latter is necessary because the file snippet is not large enough to show the entirety of the `basic_chain` implementation. This example highlights the trade-offs between the flexibility and guardrails of a command. Deciding whether to introduce a guardrail depends on how well it reduces common model errors compared to whether such restrictions hamper models' preferred workflows.

**Search & navigation.** The File Viewer and File Editor together allow agents to make edits, write tests, and perform localization at a file level. The Search & navigation module complements these capabilities by giving agents the tools to perform keyword-driven localization at both a directory level and file level.

As discussed, the main struggles with using built in Shell-only search commands such as `grep` and `find` are (1) given a general enough term, they are prone to producing too many search results that can consume an inordinate amount of space in the context window, and (2) they are highly configurable, making search result outcomes potentially inconsistent in appearance. The alternative to these search utilities is to navigate the file system directly with `cd` and look at what's in each folder with variations of `ls` and `cat`; this kind of approach can take a large number of turns without yielding any particularly useful information.

Figure 10 visualizes the standard output for the three different search commands. The `search_dir` and `find_file` helps agents perform directory level searches. The reason we provide two commands is due to the kinds of keywords that are present in an issue description (e.g., class references, file names). The `search_file` command allows agents to search for terms at a file-level, which is helpful for efficient fine-grained localization. Taking a step back, the goal of these search commands is to make it easy for the agent to utilize any signal (e.g., line number, stack trace, natural language) about the root cause of an issue that may be present in the issue description or codebase. Once again, simpler command usage patterns with consistent output formats are easier for agents to use and reduces the chance for mistakes or irrelevant outputs.

The main guardrail in place for all three search commands is curbing the number of search results to 50 or fewer. The downside is that reporting an error forces the model to generate another search query which can be an expensive operation. This reflects a trade-off between keeping observations concise and making additional calls to the base LM.

## A.2 Implementation

The SWE-agent codebase is generally composed of three modules: the environment, the agent, and the logging mechanism for saving task episodes into trajectories and patch generations.

**Environment.** The SWE-agent environment is heavily influenced by the InterCode library [59]. For the general pipeline of agent interactions with the environment, our work directly adopts InterCode's interactive coding task formulation. The environment integrates large parts of the interaction handling logic from the InterCode-Bash environment, which is essentially the Shell-only setting referenced in the paper. As a part of this adoption, SWE-agent also uses Docker containers to ensure reproducible and safe execution. Because of this, SWE-agent's infrastructure makes it easy for a user to swap out the Dockerfile (a domain specific language for defining a container) to support other codebases and programming languages beyond the scope of SWE-bench task instances. One difference is that SWE-agent makes minor adjustments to the underlying communication logic that transfers actions and observations between the Docker container and agent entity.

**Agent.** Beyond serving as an agentic wrapper for facilitating multi-turn queries from an LM, the agent module defines the functions that render the ACI (e.g., context management, commands, interface logic, input/output format) and supports inference for closed/open, API-based/local language models. The main workflow is to define an interface as a class and/or set of commands, which can then be specified via a configuration file, discussed more thoroughly in Section A.3. The commands for the top performing SWE-agent with GPT 4 configuration are shown in Table 4.

**Logging.** For each task episode, the main artifacts produced are the trajectory, which contains a history of the interactions between the agent and environment, and the final patch generation, which can represents a summary of the changes proposed by the agent during the interaction. The patch generation can be used directly for SWE-bench [20] evaluation.

## A.3 Configuration

The SWE-agent system is instantiated by three components: an LM, a SWE-bench style dataset or GitHub issue, and a configuration file. The configuration file serves to specify the design of the ACI. Iteratively refining the configuration file is the main way we achieved better agent performance and carried out different analyses for the main paper. In this section, we will present a thorough review of what a SWE-agent configuration file looks like.

An agent-computer interface is generally made up of four categories of configurable components:

1. Prompt templates: These prompt templates are used to inform the language model of the task setting, show the list of available commands, augment environment responses with the values of state variables, and provide the initial task setting.

2. Command files: These files contain the source code of bash or Python functions and scripts. Commands are easily modified, added, and removed through manipulating these files' code contents directly. Documentation added in these files can also be injected into prompts to inform the model of the available commands.

3. Control flow: Methods for parsing model responses and processing history can be specified through these configuration arguments.

4. Environment variables: Initial values of variables that may interact with commands and the shell can also be specified in the configuration.

In the following Figure 12, we include an annotated example of the contents of a configuration file.

---

**Configuration (.yaml)**

```
# Prompt Templates: Control how observations of environment are shown
↪   to agent
system_template: |
  First `system` message shown to agent
instance_template: |-
  Instance prompt, contains task instance-specific content
next_step_template: |-
  Format template of per-turn observation (Contains standard output
  ↪   from agent's action)
next_step_no_output_template: |-
```

```
    Format template of observation when there is no standard output
    ↪   from the agent's action
format_error_template: |-
    Format template of error message (Used when agent's action causes
    ↪   an error)
demonstration_template: |
    Format template for showing a demonstration to the agent
demonstrations:
- `trajectories/<username>/<experiment folder>/*.traj`
- File is a demonstration of how to solve a task. This could an agent
↪   generated trajectory.
- You can include 1+ demonstrations

# Environment States: Define features of the SWEEnv environment
env_variables:
# Default variables for SWEEnv at the beginning of each instance
    CURRENT_FILE: 0
    CURRENT_LINE:
    OVERLAP:
    SEARCH_FILES:
    SEARCH_INDEX:
    SEARCH_RESULTS:
    WINDOW_SIZE:
    START_INDEX:
    END_INDEX:
    START_CURSOR:
    END_CUROSR:
    START_CURSORS_MARK:
    END_CURSOR_MARK:
state_command: |
# `state_command` allows you to update state variables to reflect any
↪   aspect of the environment (e.g. current working directory)
    name: state
    code: |
        state() { echo '{"pwd": "'$PWD'"}';

# Action Interface: Define how an agent interacts with the SWEEnv
↪   environment
command_files:
- path/to/bash_file.sh
- Each file contains a list of commands implemented in bash
- You can include 1+ command files
parse_command: Reference to functionality for defining command
↪   documentation
history_processor: Reference to functionality for controlling agent's
↪   message history
parse_function: Parser run on agent output
```

Figure 12: An example of the configuration file that defines the SWE-agent ACI. A configuration is represented as a single `.yaml` file, allowing you to define the commands that agents may use, write prompts shown to the agent over the course of a single trajectory, and control the input/output interface that sits between the agent and environment.

The prompt templates are explained in detail in Section C. The environment variables and command files work in tandem; environment variables make the interfaces stateful, and when commands are invoked, the corresponding states are updated to reflect the changes to the environment and interface. The `parse_command`, `parse_function`, and `history_processor` all reference implementations declared within the agent module. The `parse_command` file describes how command documentation should be presented to the agent. The `parse_function` is what enforces the input/output formats for the agent. The `history_processor` points to the logic for controlling and modifying the message history enforced at each turn throughout a single task episode.

The configuration-based workflow of SWE-agent makes it easy to test new ACIs by incorporating novel commands, input/output formats, context managers, and more into the existing codebase. In the following subsections, we showcase existing implementations of several of these components and discuss how they can be extended.

**Commands.** We describe how to implement your own commands for the SWE-agent ACI. As shown in the above Figure 12, commands are declared as a list of one or more file paths in the `command_files` argument. Individual commands must be declared as separate functions in `.py` or `.sh` files. Every command subscribes to the following skeleton code in Figure 13.

---

**Command Skeleton Code**

```
# @yaml
# signature: [command] [argument(s)]
# docstring: [Brief description of what your command does.]
# arguments:
#   [argument 1 name]:
#       type: [type (i.e. integer, string)]
#       description: [Brief description of this argument]
#       required: [true|false]
#   [argument 2 name]:
#       ...
[command]() {
    # Implementation here
}
```

---

Figure 13: The skeleton code for defining a command that can be accessed in the SWE-agent ACI. The function definition includes both the underlying implementation along with several arguments that describe how to use the command, which is compiled into the System template's command documentation at run time.

The choice of Python or Bash based implementations of commands means they can be written to do a wide variety of actions, and the use of Docker means that the commands and system can be co-designed. Here is a list of guidelines around how to implement commands correctly.

- Command arguments can be referenced via positional parameters notation (i.e. `$1`).
- If there are no arguments, omit the `arguments` section.
- The implementation for your command is unconstrained. There are no limitations on the form of the underlying command code.
- The minimal documentation requirements are `signature` and `docstring`.
- Global variables can be used to make stateful changes to the environment. For instance, for the commands associated with the File Viewer, you'll see we define the `CURRENT_LINE` variable for the file viewer. This variable is modified across multiple commands, including `open`, `goto`, `scroll_up`, `scroll_down`, and `edit`.
- Third party libraries can be freely imported and used by commands (e.g., `flake8`).
- To show effects of a command, print to standard output (e.g., with `echo`). The command parsing logic is implemented such that it does not look for a return value.

Once the file path containing the command is added to `command_docs` as an argument, the command is immediately available for use in subsequent task episodes. Including a demonstration that uses more complicated commands can be helpful to showcase proper use and may increase the frequency with which the agent uses the command.

**Input/output format.** The input/output format defines what a correctly formatted response for an agent should look like. Selecting a suitable format greatly affects how well agents can interact with the environment. The methods for communicating and enforcing the input/output format are separated across several arguments. In Figure 12, the value of `parse_function` should point to a class definition that enforces the format and actually parses the agent's responses. Informing the agent of the expectations around the input/output format should take place in `system_template`, and the agent can be reminded of these standards via the `format_error_template`. New input/output formats can be easily devised and enforced by updating these arguments to point to a new class or display different natural language instructions.

**Context management.** Context management is implemented as a class within the agent module. The `history_processor` argument allows one to specify which context manager to use via the configuration file. Underneath the hood, the context manager is invoked per turn of the interactive loop. From the entire recorded history of the agent's interactions so far, the context manager constructs the literal history to be fed to the agent to invoke the next response. The general design of `history_processors` allows for easy experimentation towards more sophisticated strategies for managing history.

# B Extended Results

In this section, we provide additional results, including performance marginalized against different dimensions, patch generation statistics, and problem solving patterns reflected by SWE-agent trajectories. Per analysis, we provide numerical or qualitative evidence that supports our findings, describe our takeaways from each finding, and discuss both the strengths of SWE-agent relative to prior baselines along with future directions based on improving common failure modes.

## B.1 Hyperparameter Sweep

We performed a hyperparameter sweep using a subset of 37 instances sampled randomly from the `dev` split of SWE-bench. We present the results in Table 5, where we perform the sweeps for both the GPT-4 Turbo and Claude 3 Opus models. For GPT-4 Turbo the best configuration has a % Resolved rate of 15.1%, with a temperature of 0.0, window length of 100 and history set to last five observations (described in §3). There is a three way tie for Claude 3 Opus between the aforementioned configuration along with two additional settings (Temperature/Window/History of 0.2/100/Last-5 and 0.2/200/Full). We elect to run inference of both models on the SWE-bench test sets (both full and Lite splits) using the 0.0/100/Last-5 configuration.

Table 5: Hyper parameter sweep results on a subset of the SWE-bench `dev` split. % Resolved shows the mean score across 5 samples.

| Model | Temperature | Window | History | % Resolved |
|---|---|---|---|---|
| GPT-4 Turbo | 0.0 | 100 | Full | 14.1 |
| GPT-4 Turbo | 0.0 | 100 | Last 5 Obs. | **15.1** |
| GPT-4 Turbo | 0.0 | 200 | Full | 9.2 |
| GPT-4 Turbo | 0.0 | 200 | Last 5 Obs. | 10.8 |
| GPT-4 Turbo | 0.2 | 100 | Full | 10.8 |
| GPT-4 Turbo | 0.2 | 100 | Last 5 Obs. | 12.4 |
| GPT-4 Turbo | 0.2 | 200 | Full | 8.7 |
| GPT-4 Turbo | 0.2 | 200 | Last 5 Obs. | 10.8 |
| Claude 3 Opus | 0.0 | 100 | Full | 5.4 |
| Claude 3 Opus | 0.0 | 100 | Last 5 Obs. | **8.1** |
| Claude 3 Opus | 0.0 | 200 | Full | 7.0 |
| Claude 3 Opus | 0.0 | 200 | Last 5 Obs. | 7.1 |
| Claude 3 Opus | 0.2 | 100 | Full | 7.4 |
| Claude 3 Opus | 0.2 | 100 | Last 5 Obs. | **8.1** |
| Claude 3 Opus | 0.2 | 200 | Full | **8.1** |
| Claude 3 Opus | 0.2 | 200 | Last 5 Obs. | 6.8 |

## B.2 Model Performance

We present analyses of model performance marginalized across different dimensions and categories.

**Performance by Repository.** We include a breakdown of model performance by repository on the SWE-bench Lite dataset in Table 6. We also include and adjust the performance of Claude 2 on SWE-bench, inherited from the baseline performances established in the original work. As presented above, SWE-agent performance is superior to prior approaches, solving not only a higher percentage of problems across repositories, but also resolving problems in repositories that were previously nearly or completely unsolved by prior retrieval augmented generation baselines used in the original SWE-bench work (e.g. matplotlib, sympy/sympy).

**Temporal Analysis.** In Table 7, we provide a temporal breakdown that shows the % Resolved statistics for task instances from different years. There is no clear correlation between a task instance's

---

https://github.com/matplotlib/matplotlib/
https://github.com/sympy/sympy

Table 6: % Resolved performance across repositories represented in the SWE-bench Lite dataset. Each row corresponds to a repository while each column is the model's performance for that repository. The numbers in parentheses in the "Repo" column is the number of task instances in SWE-bench Lite that are from the corresponding repository.

| Repo | SWE-agent | | RAG | | |
| --- | --- | --- | --- | --- | --- |
| | GPT 4 | Claude 3 Opus | GPT 4 | Claude 3 Opus | Claude 2 |
| astropy/astropy (6) | 16.67% | 33.33% | 0.00% | 0.00% | 0.00% |
| django/django (114) | 26.32% | 16.67% | 4.39% | 6.14% | 5.26% |
| matplotlib/matplotlib (23) | 13.04% | 13.04% | 0.00% | 0.00% | 0.00% |
| mwaskom/seaborn (4) | 25.00% | 0.00% | 25.00% | 25.00% | 0.00% |
| pallets/flask (3) | 0.00% | 0.00% | 0.00% | 0.00% | 0.00% |
| psf/requests (6) | 33.33% | 16.67% | 0.00% | 0.00% | 0.00% |
| pydata/xarray (5) | 0.00% | 0.00% | 20.00% | 20.00% | 0.00% |
| pylint-dev/pylint (6) | 16.67% | 0.00% | 0.00% | 0.00% | 0.00% |
| pytest-dev/pytest (17) | 17.65% | 5.88% | 0.00% | 5.88% | 5.88% |
| scikit-learn/scikit-learn (23) | 17.39% | 17.39% | 0.00% | 4.35% | 8.70% |
| sphinx-doc/sphinx (16) | 6.25% | 6.25% | 0.00% | 0.00% | 0.00% |
| sympy/sympy (77) | 10.39% | 5.19% | 1.30% | 2.60% | 0.00% |

Table 7: % Resolved performance for task instances from different years represented in the SWE-bench Lite dataset. Each row corresponds to a year while each column is the model's performance for task instances with a `created_at` timestamp from that year. The numbers in parentheses in the Year column is the number of task instances in SWE-bench Lite from that corresponding year.

| Year | SWE-agent | | RAG | | |
| --- | --- | --- | --- | --- | --- |
| | GPT 4 | Claude 3 Opus | GPT 4 | Claude 3 Opus | Claude 2 |
| 2023 (30) | 23.33% | 13.33% | 3.33% | 3.33% | 0.0% |
| 2022 (57) | 21.05% | 17.54% | 5.26% | 7.02% | 1.75% |
| 2021 (42) | 23.81% | 11.90% | 2.38% | 4.76% | 2.38% |
| 2020 (66) | 10.61% | 7.58% | 3.03% | 1.52% | 1.52% |
| Before 2020 (105) | 17.14% | 10.48% | 0.95% | 4.76% | 5.71% |

creation year and its resolution rate across either models or setting. For instance, while the SWE-agent w/ GPT-4 approach solves the highest percentage of problems from 2021, while the RAG w/ GPT-4 and SWE-agent w/ Claude 3 Opus approaches perform better on task instances from 2022.

## B.3 Trajectory Analysis

We present additional characterizations of trajectories corresponding to task instances that were successfully resolved by SWE-agent w/ GPT-4 Turbo (unless otherwise specified).

### B.3.1 Turns to Resolution

Figure 14 visualizes the distribution of the number of turns SWE-agent needed to complete task instances that were successfully resolved. On the full SWE-bench test set, SWE-agent w/ GPT-4 takes an average of 14.71 turns to finish a trajectory, with a median of 12 turns and 75% of trajectories being completed within 18 turns. On the Lite split of the SWE-bench test set, SWE-agent w/ Claude 3 Opus takes an average of 12.71 turns to finish a trajectory, with a median of 13 turns and 75% of trajectories being completed within 15 turns. From the distribution, it is evident that across models and SWE-bench splits, the majority of task instances are typically solved and completed comfortably within the allotted budget.

This also points to a general area of improvement for language agent systems — if a language agent's initial problem solving approach, typically reflected in the first 10 to 20 turns, does not yield a good

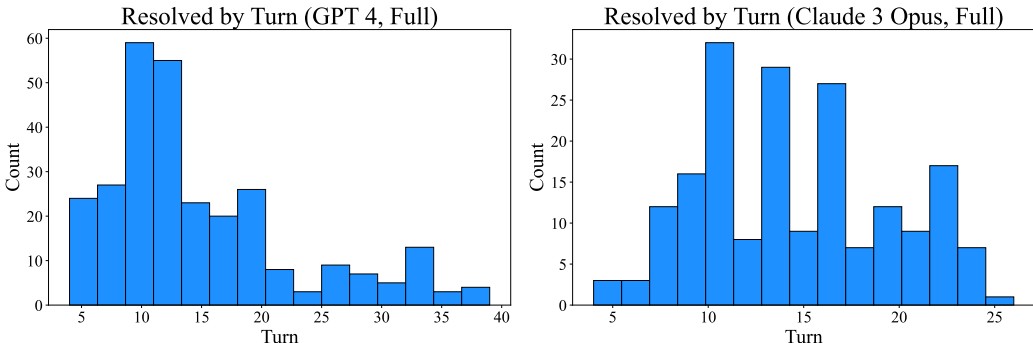

Figure 14: Distribution of the number of turns for interactive trajectories corresponding to solved task instances on SWE-bench. The left histogram shows this distribution for SWE-agent w/ GPT 4 on the full SWE-bench test set (286 trajectories). The right histogram is the performance of SWE-agent w/ Claude 3 Opus on the Lite split of the SWE-bench test set (35 trajectories).

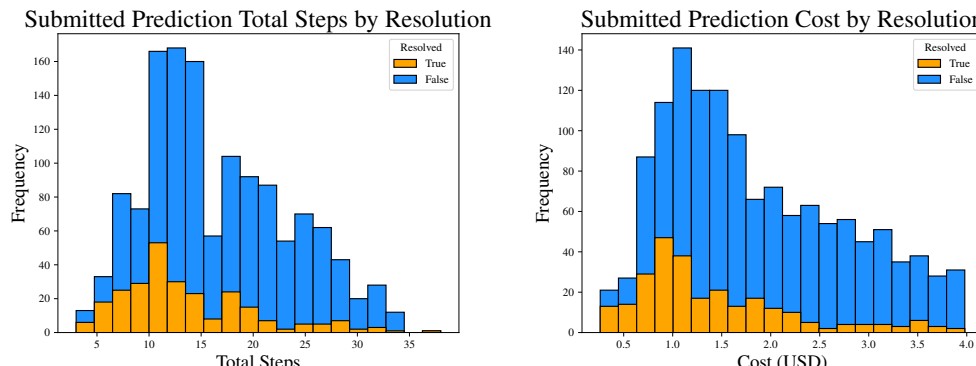

Figure 15: The distribution of agent trajectories by total steps (left) and cost (right) for SWE-agent with GPT-4 Turbo on SWE-bench. The distributions of resolved instances are shown in orange and unresolved are shown in blue. Resolved instances clearly display an earlier mean and fewer proportion of trajectories with many steps or that cost near the maximum budget of $4.00.

solution, it struggles to make use of later turns that build upon past mistakes. To remedy this issue and induce stronger error recovery capabilities in language agents, future directions could consider improving either the model, the ACI, or both.

### B.3.2 Walkthrough of Trajectory Phases

We describe what happens in different phases of an agent's problem solving trajectory. To support our observations, we present several tables and distributions that help highlight consistent trends.

**Initial reproduction, localization steps.** First, the initial steps that SWE-agent usually takes is heavily dominated by Localization and Reproduction operations. The most commonly occurring pattern in general is the `create`, `edit`, `python` triplet. Across these commands, an agent creates an empty python file, adds an executable code snippet via `edit`, and then attempts to run it. As an alternative, the agent also sometimes decides to start off instead with Localization, or identifying the files/lines causing the issue. Depending on how informative the issue description and results for initial search queries are, agents will run additional search queries with finer grained search tools to zoom in on the target problematic code area (e.g., `search_dir`, `open`, `search_file`/`scroll_down`).

These trends are also reflected in Figure 16, which shows a distribution of patterns across turns according to the categories defined in Table 8. The three leftmost bars reflect that Reproduction followed by Localization constitutes the lion's share of operations that occur in the early phases of a trajectory. For a more thorough breakdown, we also include Figure 17, which shows an estimated distribution of each action with respect to different turns, normalized across the total number of times

Table 8: We present a table of the most frequently occurring action patterns at each turn ("frequently" means ≥ 4 times) in trajectories of task instances resolved by SWE-agent w/ GPT-4. For instance, the pattern `create,edit,python` appears 156 times at the first to third turns. In addition, we also manually assign each entry a category (Reproduction, Localization (File), Localization (Line), Editing, Submission) that generally captures the underlying purpose of such a pattern. "Reproduction" refers to the sub-task of recreating the error or request described by the issue. "Localization" refers to the sub-task of identifying the code that is the cause of the issue.

| Turns | Pattern | Count | Category |
|-------|---------|-------|----------|
| 1-3 | `create,edit,python` | 156 | Reproduction |
| 1-3 | `search_dir,open,search_file` | 21 | Localization (File) |
| 1-3 | `search_dir,open,scroll_down` | 12 | Localization (Line) |
| 1-3 | `create,edit,edit` | 11 | Reproduction |
| 1-3 | `search_dir,open,edit` | 10 | Localization (Line) |
| 2-4 | `edit,python,find_file` | 71 | Localization (File) |
| 2-4 | `edit,python,edit` | 37 | Reproduction |
| 2-4 | `edit,python,search_dir` | 26 | Localization (File) |
| 2-4 | `edit,python,open` | 15 | Localization (File) |
| 2-4 | `open,edit,edit` | 13 | Editing |
| 2-4 | `open,edit,create` | 13 | Editing |
| 2-4 | `open,scroll_down,scroll_down` | 9 | Localization (Line) |
| 2-4 | `open,scroll_down,edit` | 5 | Editing |
| 2-4 | `open,edit,submit` | 5 | Submission |
| 3-5 | `python,find_file,open` | 61 | Localization (File) |
| 3-5 | `python,edit,python` | 25 | Editing |
| 3-5 | `search_file,goto,edit` | 24 | Localization (Line) |
| 3-5 | `python,search_dir,open` | 23 | Localization (File) |
| 3-5 | `edit,create,edit` | 13 | Editing |
| 3-5 | `python,edit,edit` | 11 | Editing |
| 3-5 | `python,open,edit` | 7 | Editing |
| 3-5 | `python,find_file,find_file` | 7 | Localization (File) |
| 3-5 | `edit,edit,submit` | 4 | Submission |
| 3-5 | `edit,edit,create` | 4 | Editing |
| 4-6 | `find_file,open,edit` | 28 | Editing |
| 4-6 | `find_file,open,search_file` | 19 | Localization (Line) |
| 4-6 | `edit,edit,python` | 11 | Reproduction |
| 4-6 | `goto,edit,edit` | 8 | Editing |
| 4-6 | `find_file,open,goto` | 8 | Localization (Line) |
| 4-6 | `goto,edit,submit` | 7 | Submission |
| 4-6 | `goto,edit,create` | 7 | Editing |
| 4-6 | `find_file,open,scroll_down` | 6 | Localization (Line) |
| 4-6 | `scroll_down,scroll_down,edit` | 5 | Localization (Line) |
| 4-6 | `find_file,find_file,open` | 5 | Localization (File) |
| 5-7 | `open,search_file,goto` | 29 | Localization (Line) |
| 5-7 | `open,edit,python` | 20 | Editing |
| 5-7 | `open,goto,edit` | 7 | Editing |
| 5-7 | `scroll_down,edit,submit` | 4 | Submission |
| 6-8 | `scroll_down` (x3) | 6 | Localization (Line) |
| 6-8 | `search_file,goto,scroll_down` | 4 | Localization (Line) |
| 7-9 | `edit,python,rm` | 20 | Editing |
| 7-9 | `goto,edit,python` | 12 | Editing |
| 8-10 | `python,rm,submit` | 19 | Submission |
| 8-10 | `search_file,goto,search_file` | 4 | Localization (File) |
| 9-11 | `edit` (x3) | 18 | Editing |
| 9-11 | `edit,open,edit` | 6 | Editing |
| 9-11 | `goto,search_file,goto` | 4 | Localization (Line) |

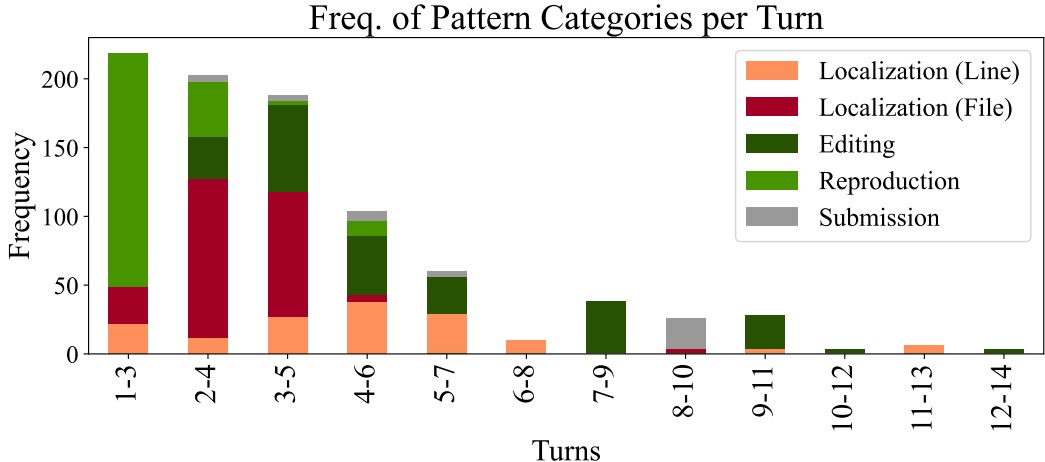

Figure 16: We assign each pattern to one of five categories (as presented in Table 8) and present a histogram of the turns at which patterns from specific categories show up frequently.

the command occurs across all turns. From these graphs, we can see that `create` is invoked much more frequently in the very first turn than in any other turn. The `search_dir` and `search_file` distributions are roughly bi-modal, with a peak of occurrences for both actions showing up in Turn 1 (if the agent decides to do Localization immediately) and the Turn 4 (if the agent decides to do Localization after Reproduction). We also present Figure 18, which communicates similar information as Figure 17, but presented instead as a stacked bar chart with more commands. The chart is created directly from Figure 7, with the frequency of actions at each turn n normalized across the total number of trajectories with a length greater than or equal to n turns.

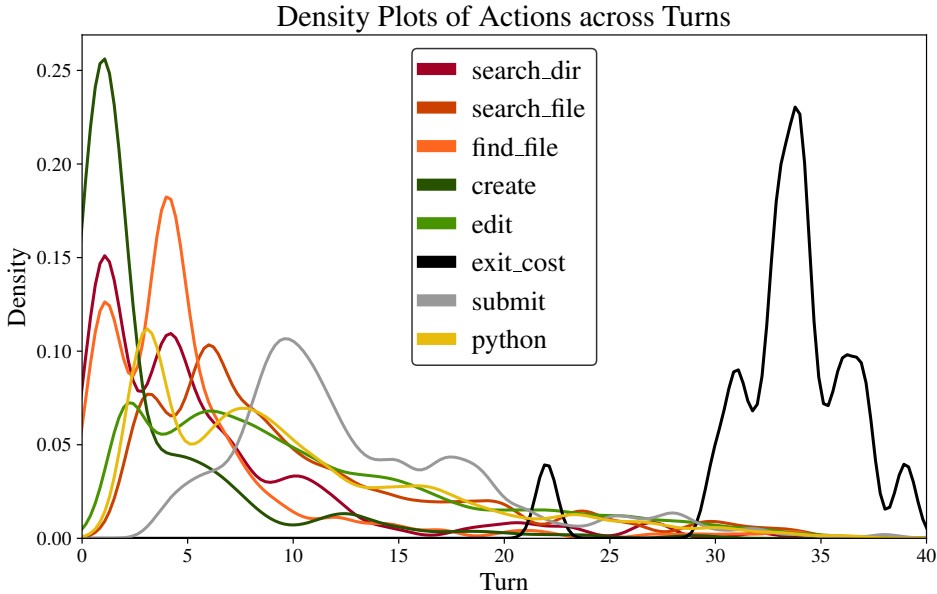

Figure 17: This density plot shows a normalized distribution of actions across different turns of a trajectory. `exit_cost` refers to when the token budget cost was exhausted and the episode's changes are automatically submitted (contrary to an intentional `submit` invoked by the agent).

**Cycle of edit, then evaluate.** From the fifth turn onwards, the distribution of actions per turn can be generally described as alternating `edit` and `python`/`pytest` actions. After reproducing the issue and localizing the file(s) responsible for the problem, agents will typically make edits to the

file, then run the reproduction script or existing tests to check whether the proposed edits resolve the original issue and maintain existing desirable behavior. This pair of actions will often repeat for several turns, as an initial edit usually does not successfully resolve the given issue. Multiple rounds of editing that are supplemented by execution feedback from prior turns are conducive to more well-formed, successful subsequent edits. As reflected in Table 8, for turn 4 onwards, the most popular pattern that begins at each turn usually falls under the Editing category. This is also made obvious by Figure 18, where the `edit` command is the most popular command for Turns 5 to 31, with only one exception (Turn 30). From Figure 17, it is also notably that the distributions of the `edit` and `python` commands are quite similar, as they typically follow one another.

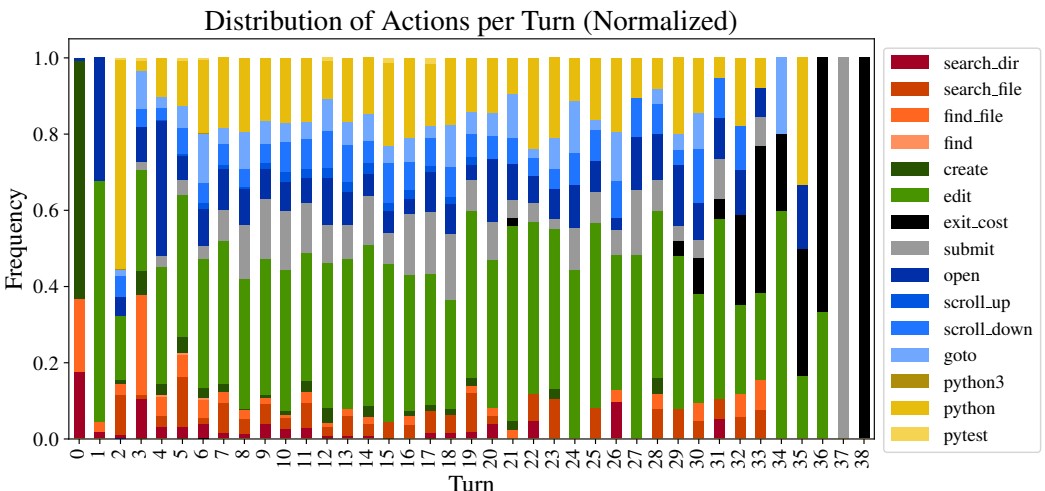

Figure 18: A normalized view of Figure 7. The distributions for turn n are normalized across the number of trajectories that have a length of at least n or more turns.

Interspersed across these later turns are additional Localization operations for inspecting other parts of the current file (e.g., `scroll_down`, `scroll_up`) or opening other files (e.g., `open`, `search_dir/file`, `find_file`). These minor trend lines reflect the tasks that involve multi-line or multi-file edits. Figure 18 displays a steady presence of such actions from Turn 6 onwards. Agents will invoke such actions to read different parts (e.g., documentation, implementation) of a long function, especially when it does not fit entirely within the file viewer's number of lines. After editing one function `A`, running the reproduction script will occasionally propagate an error in a different function `B`, where function `B` invokes `A`. This is a common reason for the additional directory and file level navigation that occurs in the later stages of a trajectory.

**Concluding submission turns.** There is a consistent proportion of `submit` actions per turn, with a relative peak around Turn 10, as shown in Figure 17. As mentioned in Section 5.2 and above, the majority of resolved task instances end with an intentional `submit` command. As suggested by both Figure 15 and Figure 18, submissions are concentrated between Turns 10 and 20, becoming less frequent for each turn beyond this range. This trend reflects how agents struggle to use later turns to their advantage, particularly when the original problem solving approach fails, which is fairly evident by Turn 20. Effectively utilizing later turns to either remedy multiple prior errors or pivot to a different problem solving approach are all viable strategies given the 20+ turns that remain. However, due to overwhelming context or greedy tendencies, agents do not reflect such dynamic behavior, instead opting to focus on continued local editing rather than additional exploration.

Finally, there is a sharp cut off of `exit_cost` actions scattered throughout Turns 30 to 40; this reflects that the $4 cost limit we impose on runs roughly corresponds to this number of turns. The discrepancies mainly comes from variations in the size of observations, with trajectories containing multiple observations that have a high number of tokens corresponding to ones that terminate relatively earlier. Increasing the cost allowance per task episode would directly increase the maximum number of the turns per episode.

### B.3.3 Breakdowns of Action Sequences

In this part, we include more granular examinations of patterns of actions that emerge frequently in trajectories. We also identify consistent associations between groups of actions and how their effects build off one another across several turns.

**Editing Trends.** Editing is a core facet of agents' ability to reproduce issues and propose fixes effectively. It is also the action that models typically struggle with the most. Here, we list several trends we were able to discern about how agents edit.

First, across the full SWE-bench test set, a non-trivial minority of edit actions are unsuccessful, meaning the `edit` invocation raises a linting error. Going forwards, we refer to such an occurrence as a *failed* edit. Out of 2,294 task instances, 1,185 (51.7%) have at least one turn with an failed edit. Of these trajectories, there is a median of 3 failed edits per trajectory, with a max of 33. The rate of failed edits is smaller for resolved task instances. Out of 286 resolved instances, 113 (31.5%) have at least one turn with an failed edit, with a median/mean/max of 2 failed edits per trajectory, with a max of 26. Figure 19 shows corresponding distributions.

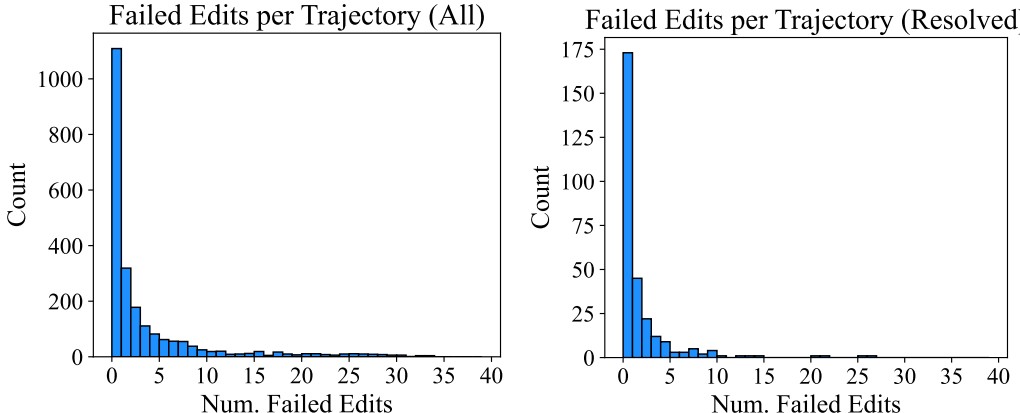

Figure 19: Distribution of the number of failed `edit` actions per trajectory across all (left) and resolved (right) task instances by SWE-agent with GPT-4 Turbo. A "failed" edit refers to an edit action that raised a linting error. The left-most bar for both graphs corresponds to the number of trajectories with no failed edits.

Second, with linting enabled editing, agents "recover" more often than not from failed edits. To understand whether and how effectively agents use linting error feedback to construct a subsequent, well-formed `edit` action, we define two terms. Recovery refers to a sequence of failed edits followed immediately by a successful edit, suggesting the agent used linting feedback to make a well-formatted edit. An unsuccessful recovery is consecutive failed edits followed immediately by a non-edit action.

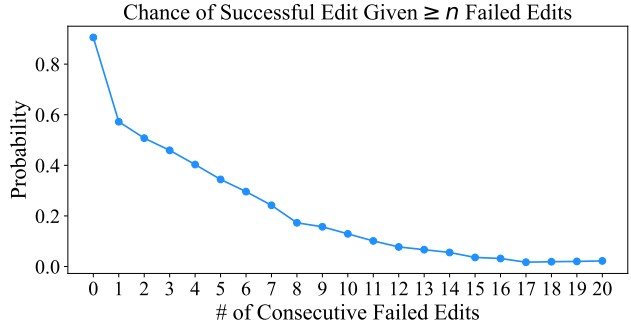

Figure 20: Probability of successful edit after `n` failed edits. The likelihood of recovery decreases as `n` increases.

Across trajectories corresponding to resolved task instances, there are 135 occurrences of 1+ failed edit attempts. Out of these, the agent recovers successfully 104 times. The number of consecutive failed edit attempts before a successful versus failed recovery is also vastly different. Successful recoveries are usually preceded by 2.03 edit attempts, less than the average 4.22 failed edit attempts of unsuccessful recoveries. Across all task instances, the relative rate of unsuccessful recoveries increases, with 810 successful recoveries versus 555 unsuccessful ones. While the number of consecutive failed edit attempts resulting in a recovery remains steady (2.2), it increases significantly for unsuccessful recoveries (5.59).

Third, the odds of recovery decreases as the agent accumulates more failed edit attempts. Figure 20 displays a line plot of the probability of a successful edit given n failed edit attempts in a row. The leftmost data point of $n = 0$ means that any attempt at editing has a 90.5% chance of eventually being successful. This value drops off once the agent incurs a single failed edit; there is only a 57.2% chance the edit is ultimately successful. In other words, there is a 42.8% chance the agent never recovers upon encountering 1 edit error.

**Action sequence analysis.** We calculate the transition probabilities showing the likelihood of the next action given the previous n actions. To perform this analysis, we first determine the 15 most commonly occurring sequences of n actions, for $n \in \{1, 2, 3, 4\}$. We then count how frequently each command appears after this sequence and finally normalize the counts across the total number of occurrences of the sequence to get a likelihood of the "Next Action" with respect to the preceding n sequence of actions.

We show these transition probability heatmaps, with $n = 1$ in Figure 21, $n = 2$ in Figure 22, $n = 3$ in Figure 23, and $n = 4$ in Figure 24. From these graphs, it is immediately obvious that several action sequences emerge consistently across many task instances. The high likelihood cells in these heatmaps suggest that SWE-agent uses common problem solving patterns which correspond to higher order operations such as reproducing an issue, localizing buggy code, and proposing/verifying edits.

In Figure 21, we see direct associations between pairs of actions. There are several obvious trends. All trajectories begin with `create`, `find_file`, `search_dir`, and end on either a `submit` or `exit_cost`. The most popular next action is `edit`; it is the most likely action to follow `create`, `edit`, `goto`, `pytest`, and `python`. Scroll (e.g., `scroll_down/up`) and search (e.g. `find_file`, `search_dir`) actions tend to be repeated.

Other interesting correlations are also present. The edit/evaluate pattern is reflected in the correlation between the `edit` and `python` pair. A variety of localization patterns are also conspicuous. Sometimes, searching for a file turns out to be less fruitful than searching for a keyword, and visa versa. This is reflected in the `find_file` and `search_dir` pair. The invocation of `open` is representative of an agent honing in on a specific file to then continue localizing (`search_file` 0.35, `scroll_down` 0.18, `goto` 0.09) or begin editing (`edit` 0.25).

As the number of prior actions considered increases, more complex operations carried across multiple commands become apparent, echoing the observations from Table 8. In Figure 23, reproduction (e.g. [`create`, `edit`, `python`]) is typically followed by adjustments to the script (`edit` 0.39) or localization (`find_file` 0.31, `search_dir` 0.22). Fruitful localization patterns are once again reflected by [`find_file` / `search_dir`, `open`, `search_file`] are followed by `goto`. In Figure 24, the most popular 4-grams are related to reproduction or editing. The [`edit`, `python`, `rm`, `submit`] pattern is a popular way for trajectories to finish. Common failure modes are also apparent; repeated actions like `edit` (4x) and `scroll_down` (4x) typically continues cascading.

## B.4    Failure Modes

In this section, we provide insight on categorizing common agent failure modes. We perform an automated analysis of the unresolved trajectories ($n = 248$) from the SWE-bench Lite split with our default configuration. We first create a list of possible failure categories based on model behavior analyzed in Sections B.3.2, which are described in full detail in Table 9. A validation set of 15 instances are then sampled from the 248 instances left marked unresolved by SWE-agent and the authors hand-label them according to these categories. Finally, we combine the agent's trajectory with the patch generated by its changes and the gold patch for reference and use an LM to categorize

---

We use `gpt-4o-2024-05-13` from OpenAI.

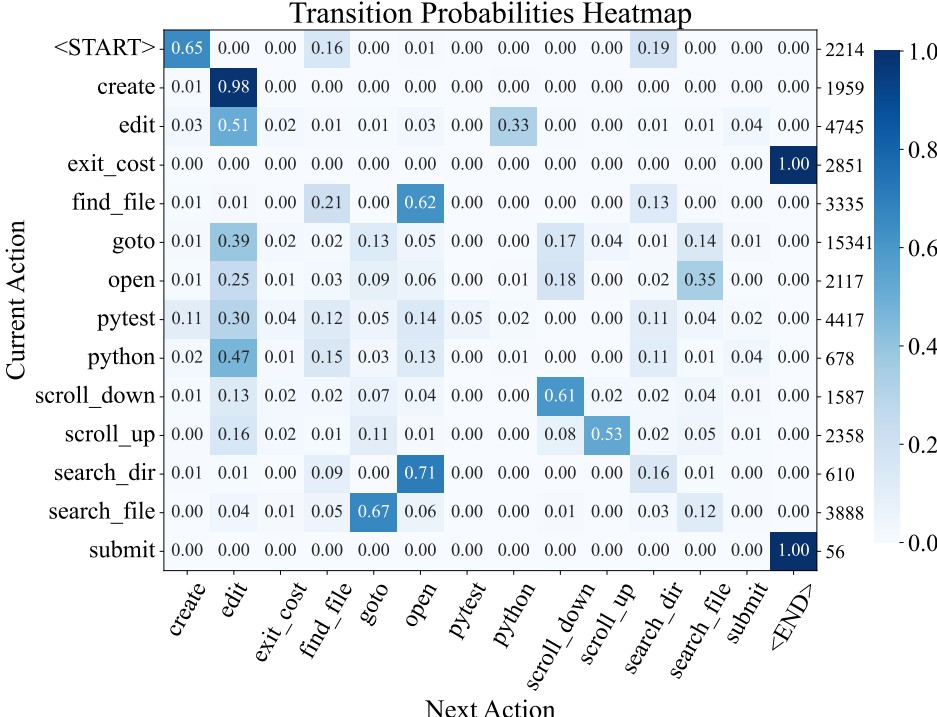

Figure 21: Heatmap displaying the relative frequency of different actions being invoked after the most popular actions in SWE-agent w/ GPT-4 Turbo trajectories across all task instances.

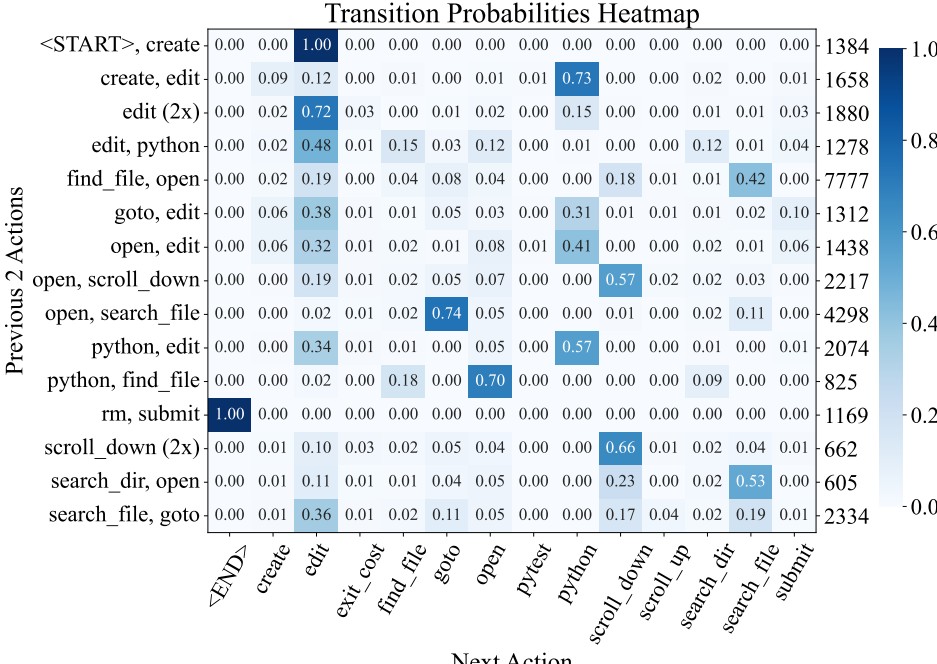

Figure 22: Heatmap displaying the relative frequency of different actions being invoked after the most popular *pairs* of actions in SWE-agent w/ GPT-4 Turbo trajectories across all task instances.

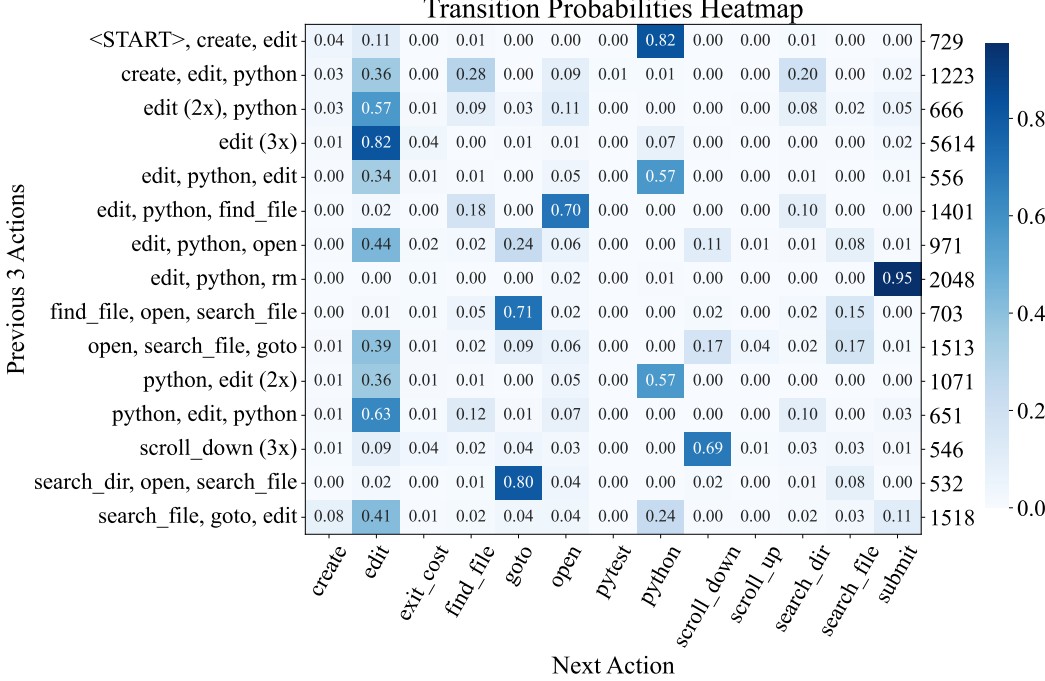

Figure 23: Heatmap displaying the relative frequency of different actions being invoked after the most popular *triplets* of actions in SWE-agent w/ GPT-4 Turbo trajectories across all task instances.

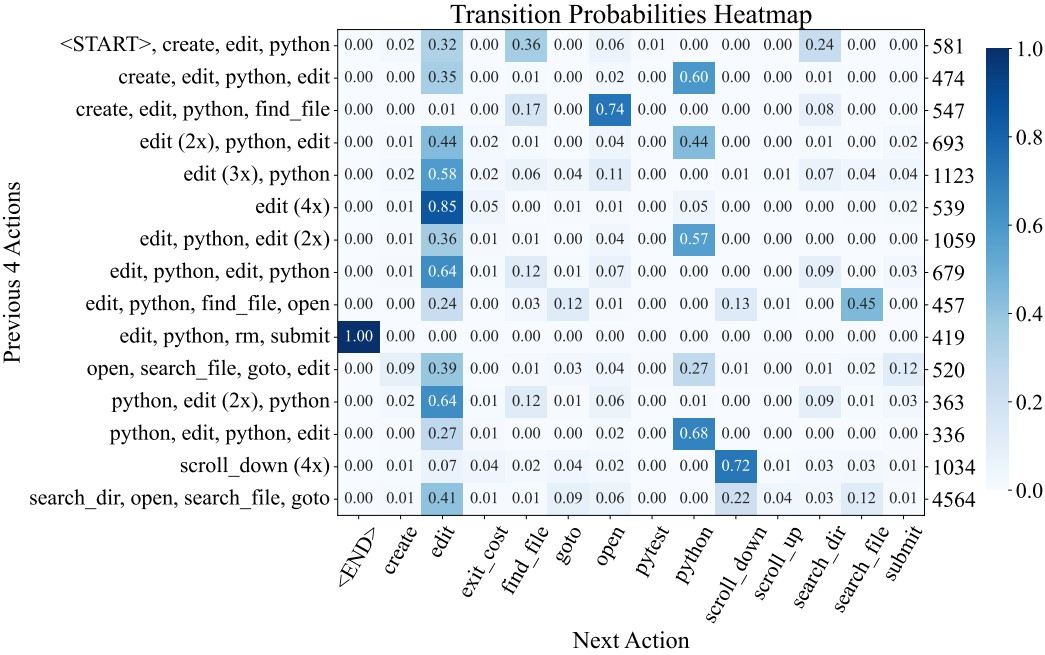

Figure 24: Heatmap displaying the relative frequency of different actions being invoked after the most popular *quadruplets* of actions in SWE-agent w/ GPT-4 Turbo trajectories across all task instances.

each trajectory. In Figure 8, we show the results of this automated categorization. Evaluated on our validation set, the LM generated labels agree with the authors' labels on 87% of instances.

We find that about half (52.0%) of the unresolved instances fall into the Incorrect Implementation or Overly Specific Implementation categories, suggesting that agents' proposed solutions often simply fail to functionally address the issue or are insufficiently general solutions. Another significant category is the Failed Edit Recovery category, making up 23.4% of instances, which happens when models fail to generate well-formed edits to files, which can seriously inhibit their performance. The remaining failure modes make up less than 25% of instances, but highlight different aspects of the challenges faced by the agent in the problem-solving process.

Table 9: Descriptions of failure mode categories.

| Category | Description |
|---|---|
| Failed to Reproduce | The agent tried but was not able to successfully reproduce the problem in the issue. |
| Failed to Find Relevant File | The agent never opened or saw the correct file. |
| Failed to Find Edit Location | The agent opened and viewed the correct file but didn't find or edit a relevant location. |
| Overly Specific Implementation | The agent made a relevant change but its solution was not sufficiently general; in this case it might solve the very specific issue suggested but it does so in a way that might change the behavior of the code in other, more general, cases. |
| Incorrect Implementation | The agent made a change to a reasonable area but their solution didn't correctly address the issue. |
| Ran Out of Budget | The agent seemed to be on the right track to a solution, but the episode ended before they could complete their changes. |
| Failed Edit Recovery | The agent went into an edit loop, making recurrent failing edits without recovering. |
| Gave Up Prematurely | The agent decides to stop solving the problem after encountering some difficulty. |
| Other | There was some other problem that prevented the agent from resolving this issue. |

## B.5 Performance Variance and Pass@k Rate

Since running SWE-agent on SWE-bench can be rather expensive, we perform, all results, unless otherwise stated, are reported using a pass@1 metric (% Resolved). However, we also test our main SWE-agent configuration for a higher number of runs to test the variance and pass@$k$ performance for $k \in \{3, 6\}$. These results are shown in Table 10, suggesting that average performance variance is relatively low, though per-instance resolution can change considerably.

Table 10: Performance for 6 separate runs of SWE-agent with GPT-4 on SWE-bench Lite. The % Resolved rate for each individual run is shown in the first table, and the pass@k rate in the second.

| | SWE-bench Lite | | | | | | |
|---|---|---|---|---|---|---|---|
| | Run 1 | Run 2 | Run 3 | Run 4 | Run 5 | Run 6 | Avg. |
| Resolve % | 17.33 | 18.00 | 18.00 | 18.67 | 17.33 | 18.33 | $17.94_{0.49}$ |

| | Pass@1 | Pass@2 | Pass@3 | Pass@4 | Pass@5 | Pass@6 |
|---|---|---|---|---|---|---|
| Pass@k | 17.94 | 23.89 | 27.35 | 29.67 | 31.33 | 32.67 |

### B.6 Patch Generations

In this section, we present some statistics and analysis around the edits generated by SWE-agent. At the end of a task episode, the edits made by SWE-agent are aggregated and saved as a single `.patch` file, the canonical representation for code changes of a pull request on GitHub. From these patch representations, we can quantitatively characterize an agent's generations and see how they compare to the original solutions written by human codebase maintainers.

Table 11 presents a summary of four basic statistics about the model generations. Lines added and lines removed refer to the total number of lines that were added or deleted in the patch, an indicator of the size of the modification. The number of hunks and files is more indicative of how many "regions" of the codebase were modified. A higher number of hunks and files suggests that there are more distinct, separate places in the codebase where the patch made changes. For both "Resolved" and "All" categories of task instances, models tend to generate "larger" edits (e.g., more lines added, hunks, and files) than the corresponding gold solution. Prior RAG baselines in Jimenez et al. [20] typically produce smaller edits on average. The source of this increase for agent-generated solutions can largely be attributed to additional reproduction code.

Table 11: We show the (median) / (mean) value for several statistics characterizing patch generations. We calculate these statistics across two dimensions. First, the "Resolved" / "All" labels denote whether the patch resolved the issue. Second, for the task instances specific to each model, we calculate the same statistics across the gold patches. To diminish the effect of outliers, we calculate these statistics based on values falling within within the 90th percentile of the distribution.

| Model | Outcome | Lines + | Lines - | Hunks | Files |
|---|---|---|---|---|---|
| SWE-agent w/ GPT-4 Turbo | Resolved | 3.0 / 5.7 | 1.0 / 1.32 | 1.0 / 1.52 | 1.0 / 1.22 |
| | Any | 12.0 / 16.58 | 1.0 / 1.35 | 2.0 / 1.83 | 1.0 / 1.53 |
| Gold | Resolved | 2.0 / 3.58 | 1.0 / 1.98 | 1.0 / 1.3 | 1.0 / 1.0 |
| | Any | 7.0 / 11.67 | 2.0 / 4.05 | 2.0 / 2.45 | 1.0 / 1.24 |
| SWE-agent w/ Claude 3 Opus | Resolved | 3.0 / 5.09 | 1.0 / 1.59 | 1.0 / 1.56 | 1.0 / 1.26 |
| | Any | 11.0 / 15.25 | 1.0 / 1.79 | 2.0 / 2.14 | 2.0 / 1.87 |
| Gold | Resolved | 3.0 / 3.91 | 1.0 / 1.94 | 1.0 / 1.4 | 1.0 / 1.0 |
| | Any | 6.0 / 10.68 | 2.0 / 3.61 | 2.0 / 2.22 | 1.0 / 1.13 |

When comparing the "Resolved" and "All" categories, we see that successfully resolved edits are relatively smaller than the original distribution. This trend is consistent with the RAG based solutions; issues that require multiple edits across a codebase remains challenging for agents.

### B.7 HumanEvalFix Evaluation

In this section, we include further discussion about our evaluation of SWE-agent on HumanEvalFix. We choose to evaluate on the HumanEvalFix task because it focuses on code editing and debugging, which was empirically demonstrated in Muennighoff et al. [32] to be a more difficult task for LMs (as reported in their work, GPT 4 scores 78.3% on HumanEval, compared to 47.8% on HumanEvalFix). The code editing task can also be thought of as a "subtask" in SWE-bench; being able to identify and fix bugs is a major part of software engineering.

We adopt the HumanEvalFix dataset (164 problems per language) to be compatible with the SWE-agent setting. Following the documentation in Muennighoff et al. [32], SWE-agent is initialized in a directory with a single file containing a buggy code snippet and example test(s) if available. It is then asked to edit the code and verify its fixes. The configuration file is identical to the one used for SWE-bench, with the exception of a language-specific demonstration. For this task, localization and navigating a large codebase are not necessary; the main focus is on generating the correct edit. SWE-agent achieves the best performance on the HumanEvalFix benchmark for three of the languages we evaluate on, as shown in Table 2. Figure 25 also suggests that the large majority of task instances are solved within the first 10 turns.

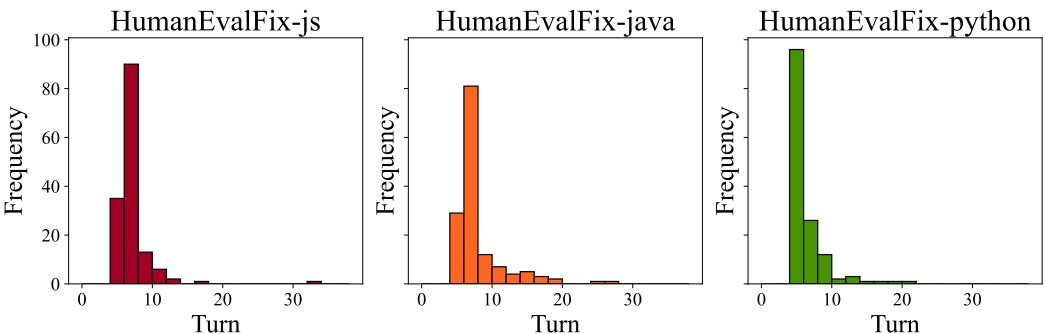

Figure 25: Similar to Figure 14, we show the distribution of the number of turns for trajectories corersponding to solved task instances from the HumanEvalFix dataset.

## B.8 Dataset Information

In the following Table 12, we provide descriptions of the two datasets that we use for evaluation: SWE-bench [20] and HumanEvalFix [32]. Both datasets have been released under permissive software licenses that allow for evaluation use, and can be used in proprietary systems.

Table 12: Information about each of the datasets that we evaluate SWE-agent on.

| Dataset | Released | License | Splits | Count | Languages | GitHub Repo |
|---------|----------|---------|--------|-------|-----------|-------------|
| SWE-bench | 10/10/2023 | MIT | Test | 2294 | Python | princeton-nlp/ |
| | | | Lite | 300 | | SWE-bench |
| | | | Dev | 225 | | |
| HumanEvalFix | 07/23/2023 | MIT | Test | 164 | Python, JS, Go Java, C++, Rust | bigcode-project/ octopack |

## B.9 Miscellaneous

In this section, we include additional minor analyses around agent behavior and their generations.

**Agents are better at localizing files than BM25.** The interactive setting also enables agents to identify the correct file(s) to edit more often compared to the RAG baselines in Jimenez et al. [20]. To measure this, we calculate the F1 score between the set of [edited, removed] files by the agent's prediction versus the gold patch. SWE-agent w/ GPT-4 Turbo achieves an F1 score of 59.05%, while BM25 w/ Claude 3 Opus produces an F1 score of just 45.47%.

**Most resolved task instances are intentionally submitted.** There are four ways a task episode ends.

- "Submit" refers to a task episode that ends when the agent generates the `submit` command.
- "Exit Cost (Submit)" refers to the scenario where the episode ends because the cost limit was hit, and the changes so far are gathered and submitted as an edit.
- "Exit Cost (No Submit)" refers to when the cost limit was hit and no `edit`'s were made, so there was nothing to submit. In this scenario, the instance is guaranteed to be unresolved.
- "Early Exit" refers to when the task episode terminates because an agent issued too many malformed responses in a row. Any changes are submitted as an edit.

Table 13 shows the counts for the number of trajectories that ended on these four different outcomes, categorized across the agent, SWE-bench split, and whether or not that task instance was resolved. For SWE-agent with GPT-4 Turbo, the majority of "All" task instances are `submit`. For the trajectories corresponding to"All" task instances by SWE-agent with Claude 3 Opus, slightly less than 50% of task instances are submitted, while the slight majority are auto-submitted when the cost limit is hit.

Table 13: This table showcases the counts for the four ways ("Submit", "Exit Cost (Submit)", "Exit Cost (No Submit)", "Early Exit") a task episode could conclude.

| Model | Split | Outcome | Submit | Exit Cost (Submit) | Exit Cost (No Submit) | Early Exit |
|---|---|---|---|---|---|---|
| SWE-agent w/ GPT-4 Turbo | Full | Resolved | 266 | 20 | 0 | 0 |
| | | All | 1589 | 630 | 48 | 1 |
| | Lite | Resolved | 50 | 4 | 0 | 0 |
| | | All | 203 | 95 | 2 | 0 |
| SWE-agent w/ Claude 3 Opus | Full | Resolved | 206 | 35 | 0 | 0 |
| | | All | 882 | 1048 | 73 | 1 |
| | Lite | Resolved | 32 | 3 | 0 | 0 |
| | | All | 133 | 156 | 11 | 0 |

However, these trends do not hold for "Resolved" task instances. For SWE-agent with both models, the large majority of these task instances are submit. Reiterating the conclusion in Section 5.2 and prior visualizations in Section B.3, we see here again that resolved task instances often imply that the agent is able to produce and verify an edit within the allotted number of turns. The SWE-agent ACI is also effective at eliciting well-formed thoughts and actions from agents. Across all runs, there are only two "Early Exit" occurrences, where the episode terminated because the agent generated too many malformed responses in a row.

Finally, Table 13 also upholds an expected trend. Task instances that finish with a submit action are more likely to be resolved than those that are cutoff by cost. For instance, for SWE-agent with GPT-4 Turbo on full SWE-bench, 14.3% of task instances that end with a submit are resolved, which is much higher than 3.1% for those finishing on exit_cost.

# C  Prompts

In this section, we go through the prompt templates that make up the agent's history, discussing them in the order of presentation to SWE-agent. Per template, we describe its purpose, walk through its content, and note any additional motivations that influenced how we wrote the template. The companion figures of template content are all drawn from our default configuration, using SWE-agent w/ GPT-4.

The template content can and should be adapted slightly to fit the agent's intended use case. The purpose of this section is to describe our thought process for how we designed each template for these tasks to serve as reference for future work. Across templates, we find that providing tips which tell agents to not make specific mistakes, avoid common pitfalls, and use helpful execution signals are effective for eliciting more successful problem solving.

**Prompt Workflow.** We present Figure 26 which shows the order in which different prompt templates are invoked. This flow of prompts reflects the logic that generates trajectories similar to the one that is visualized in Figure 9.

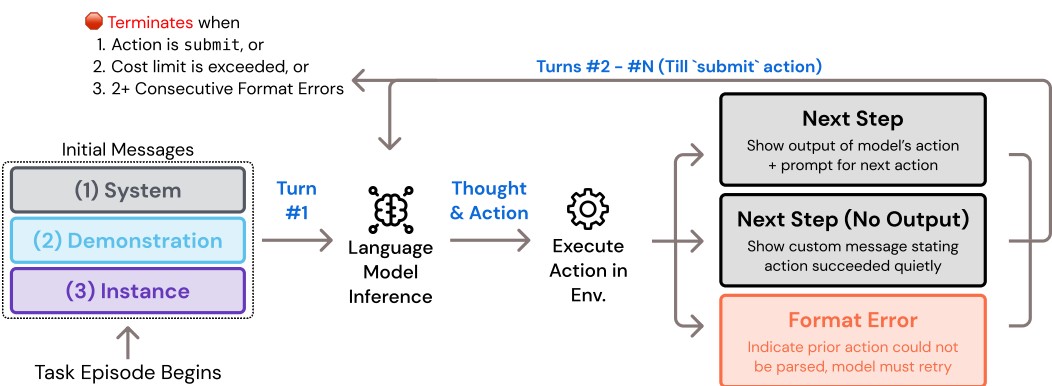

Figure 26: The flow of prompt templates throughout a single SWE-agent task instance episode. The system, demonstration, and issue templates are shown all together at the beginning of the task episode, followed by turn-specific prompts that are shown depending on whether the agent response is well-formatted and whether the action has standard output.

**System Template.** The system template describes the interactive task setting, the commands at the agent's disposal, and the expected response format. It is the first message for any episode, does not change in content across task instances, and is not removed or collapsed at any point from the message history. The agent is told of the general task setting, which is a command line that comes with a special file viewer interface. After this, the agent is presented the command documentation, which shows a usage example and docstring for every custom command, mirroring the content of Figure 27. As discussed before, from manual observation, we find that agents need a lot of support to make effective use of the `edit` command.

---

**System Prompt**

**SETTING:** You are an autonomous programmer, and you're working directly in the command line with a special interface.

The special interface consists of a file editor that shows you 100 lines of a file at a time. In addition to typical bash commands, you can also use the following commands to help you navigate and edit files.

**COMMANDS:** {documentation}

Please note that THE EDIT COMMAND REQUIRES PROPER INDENTATION. If you'd like to add the line '      print(x)' you must fully write that out, with all those spaces before the code! Indentation is important and code that is not indented correctly will fail and require fixing before it can be run.

**RESPONSE FORMAT:**
Your shell prompt is formatted as follows:
(Open file: <path>) <cwd> $
You need to format your output using two fields; discussion and command. Your output should always include *one* discussion and *one* command field EXACTLY as in the following example:

---

DISCUSSION
First I'll start by using ls to see what files are in the current directory. Then maybe we can look at some
relevant files to see what they look like.
```
ls -a
```

You should only include a *SINGLE* command in the command section and then wait for a response from the
shell before continuing with more discussion and commands. Everything you include in the DISCUSSION
section will be saved for future reference. If you'd like to issue two commands at once, PLEASE DO NOT
DO THAT! Please instead first submit just the first command, and then after receiving a response you'll be
able to issue the second command. You're free to use any other bash commands you want (e.g. find, grep,
cat, ls, cd) in addition to the special commands listed above. However, the environment does NOT support
interactive session commands (e.g. python, vim), so please do not invoke them.

Figure 27: The system prompt for SWE-agent describes the environment. The `documentation`
field is populated with brief description of all enabled commands, similar to Table 4.

An agent will occasionally generate an `edit` with either the wrong level of indentation or incorrectly
specified line numbers. Because of this, we include a note telling the agent to pay attention to
proper indentation. Finally, the system prompt describes what the agent's response should look
like, communicated with an example (e.g. JSON format, XML delimiters) followed by a paragraph
reinforcing the importance of issuing a *single* thought/action pair per turn. Because of the constraints
imposed by Docker containers, we include one last point about the command line environment not
supporting any interactive session commands, such as `vi` or `python`. The system template does not
introduce any task instance specific information.

**Demonstration Template.** If provided, the demonstration template immediately follows the system
template as the second message showing the agent a trajectory which resulted in the successful
resolution of a task instance from the development set. As confirmed by the ablation in Table 3,
including a demonstration slightly helps agents with understanding proper command usage and
reduces the rate of errant responses. Rather than being written out entirely like the system template,
the demonstration template is put together based on two fields as shown in the configuration file
discussed in Section A.3. First, the `demonstrations` argument points at 1+ `.traj` file(s), each
containing a multi-turn, SWE-agent style history of interactions.

Second, the `demonstration_template` and `put_demos_in_history` arguments control
how the demonstration is represented in the message history. If the `put_demos_in_history`
argument is set `True`, every turn of the demonstration trajectory is used as a separate message in
the history. The alternative is to write out the entire demonstration as a single message, wrapped in
a natural language instruction indicating it is a demonstration (this is the method that is displayed
in Figure 28). The effectiveness of each method varies by model. The first method of adding a
demonstration as separate messages can be preferable because it gives an agent the impression that it
has already solved a task instance and is working on the next one. However, for a different model, it
can be confusing to have the contexts from two tasks in a single trajectory, which is why the second
method of explicitly telling the model that it's being given a demonstration can be better.

**Demonstration**

```
Here is a demonstration of how to correctly accomplish this task.
It is included to show you how to correctly use the interface.
You do not need to follow exactly what is done in the demonstration.
--- DEMONSTRATION ---
{{INSTANCE PROMPT}}

The code has been updated to use the `round` function, which should
↪  fix the rounding issue. Before submitting the changes, it would
↪  be prudent to run the reproduce.py code again to ensure that this
↪  change has the desired effect. I see that "Current directory" is
↪  still `/marshmallow-code__marshmallow`, so we can just run
↪  `reproduce.py` directly.
```
```

```
python reproduce.py
```
345

(Open file: /marshmallow-code__marshmallow/src/marshmallow/fields.py)
(Current directory: /marshmallow-code__marshmallow)
bash-$
The output has changed from 344 to 345, which suggests that the
↪  rounding issue has been fixed. Let's remove the reproduce.py file
↪  since it is no longer needed.
```
rm reproduce.py
```
Your command ran successfully and did not produce any output.
(Open file: /marshmallow-code__marshmallow/src/marshmallow/fields.py)
(Current directory: /marshmallow-code__marshmallow)
bash-$
rm doesn't have any output when it deletes successfully, so that must
↪  have worked. Now that we have fixed the issue, we should submit
↪  our changes to the code base. Let's submit the changes using the
↪  `submit` command.
```
submit
```
--- END OF DEMONSTRATION ---
```

Figure 28: A simplified demonstration template showing how demonstrations are provided to the model as a single message. Here we show only the final 3 turns in the demonstration for brevity.

We are unsure if demonstrations actually help agents understand the nuances of domain specific problem solving. Because of the diversity of software engineering issues, we think the role the demonstration plays is primarily to help the agent learn to issue properly formatted commands. Prior work has demonstrated that fine tuning may have the potential to imbue agents with a certain degree of expertise around how to adaptively solve task instances that may vary in terms of what strategy is most successful.

**Instance Template.** The instance template introduces the agent to the task instance. The problem statement is shown, followed by a brief set of instructions that reiterate important points from the system template. These points are the one thought/action per-turn requirement, mentioning the lack of support for interactive shell commands, and a reminder of the importance of editing indentation. Finally, a notably effective part of the instance template is the inclusion of tips which serve as an additional guidelines for how to operate successfully in the bash environment, shown in Figure 29. These tips were developed manually and iteratively; after running SWE-agent with a particular configuration on the development set, we manually looked at the trajectories for failure modes. The tips were born out of these failures, and through repeated inspection, we found that such tips did reduce the frequency of errant problem solving strategies that they are meant to address. While our manual approach to writing tips certainly does not scale, representing feedback for common mistakes as tips is surprisingly effective. Developing better methods for this process of identifying failure modes and writing natural language instructions that describe the correct alternative behavior could be an avenue to better performance for future SWE-agent based systems. Finally, at the end of the message, the agent is presented with a command line prompt indicating that the task has begun and that the agent should issue its first command.

**Instance Message**

```
We're currently solving the following issue within our repository.
Here's the issue text:
ISSUE:
{issue}
```

```
INSTRUCTIONS:
Now, you're going to solve this issue on your own. Your terminal
↪  session has started and you're in the repository's root
↪  directory. You can use any bash commands or the special interface
↪  to help you. Edit all the files you need to and run any checks or
↪  tests that you want.  Remember, YOU CAN ONLY ENTER ONE COMMAND AT
↪  A TIME. You should always wait for feedback after every command.
↪  When you're satisfied with all of the changes you've made, you
↪  can submit your changes to the code base by simply running the
↪  submit command. Note however that you cannot use any interactive
↪  session commands (e.g. python, vim) in this environment, but you
↪  can write scripts and run them. E.g. you can write a python
↪  script and then run it with `python <script_name>.py`.

NOTE ABOUT THE EDIT COMMAND: Indentation really matters! When editing
↪  a file, make sure to insert appropriate indentation before each
↪  line!

IMPORTANT TIPS:
1. Always start by trying to replicate the bug that the issues
↪  discusses. If the issue includes code for reproducing the bug, we
↪  recommend that you re-implement that in your environment, and run
↪  it to make sure you can reproduce the bug. Then start trying to
↪  fix it. When you think you've fixed the bug, re-run the bug
↪  reproduction script to make sure that the bug has indeed been
↪  fixed.

2. If you run a command and it doesn't work, try running a different
↪  command. A command that did not work once will not work the
↪  second time unless you modify it!

3. If you open a file and need to get to an area around a specific
↪  line that is not in the first 100 lines, say line 583, don't just
↪  use the scroll_down command multiple times. Instead, use the goto
↪  583 command. It's much quicker.

4. If the bug reproduction script requires inputting/reading a
↪  specific file, such as buggy-input.png, and you'd like to
↪  understand how to input that file, conduct a search in the
↪  existing repo code, to see whether someone else has already done
↪  that. Do this by running the command: find_file "buggy-input.png"
↪  If that doesn't work, use the linux 'find' command.

5. Always make sure to look at the currently open file and the
↪  current working directory (which appears right after the
↪  currently open file). The currently open file might be in a
↪  different directory than the working directory! Note that some
↪  commands, such as 'create', open files, so they might change the
↪  current  open file.

6. When editing files, it is easy to accidentally specify a wrong
↪  line number or to write code with incorrect indentation. Always
↪  check the code after you issue an edit to make sure that it
↪  reflects what you wanted to accomplish. If it didn't, issue
↪  another command to fix it.

(Open file: {open_file})
(Current directory: {working_dir})
bash-$
```

Figure 29: The instance template. This message shows the task instance's problem statement (referenced by the {issue} field), shows additional task instance-specific information, and provides a set of tips suggesting recommended problem solving approaches and pitfalls to look out for.

**Next Step Template.** Assuming an agent's response is well formed and contains an action, there are two simple templates used to present the corresponding output of the agent's action, as shown in Figure 30. If an agent's action produces some standard output, the agent is simply shown this output with a command line prompt to indicate that the agent should respond with the next action.

> **Next Step Template**
>
> ```
> {OBSERVATION}
> (Open file: /path/to/open/file.py)
> (Current directory: /path/to/cwd)
> bash-$
> ```

Figure 30: The environment's "next step" template. This is emitted after each observation to inform the model of the current state of the shell and programs.

However, if an agent's action runs silently and produces no output (e.g. `rm abc.py`, `touch abc.py`), we found that simple showing no output along with a prompt can be confusing for agents to interpret, and it will often run additional, unnecessary commands to determine the effect of the prior action. To guard against this situation, the agent is informed verbosely that the command ran successfully and did not produce output. While the System, Demonstration, and Instances template are only used a single time, the next step template is used repeatedly. In the SWE-agent configuration described in this work, the next step templates are fairly simple, as they essentially just add the command line prompt to the end of the execution standard output. We have not explored other variations to this style.

**Collapsed Observation Template.** As shown in Figure 9 and discussed in Section 2, old observations are *collapsed*; meaning that the structure and order of the agent's interaction history is preserved, but the content of old observations are replaced with a one-line placeholder. This summary simply states that the observation is omitted with the number of lines that were removed, as shown in Figure 31.

> **Environment Response (collapsed) Template**
>
> ```
> Old output omitted (101 lines)
> ```

Figure 31: A *collapsed* environment response. The content of old observations are replaced with this template.

The purpose of collapsing old observations is twofold. First, it simply reduces the number of tokens needing to be processed at each interaction step. Second, by removing old observations' content, it can also reduce the presence of outdated or duplicate information, such as directory and file contents or command output. We think that both of these purposes can serve to improve agent performance.

**Error Message Template.** An agent's response is not always well formed. As discussed, an agent is generally expected to respond with a single thought and action per turn. In our work, we experimented with asking the agent to generate one of three different styles of responses for communicating one thought and one action (same type of response across any single episode).

1. A string where the action is wrapped as a markdown-style code block (```).
2. A JSON style dictionary with "thought" and "action" keys.
3. An XML style generation with "<thought>" and "<action>" delimiters.

On occasion, an agent may generate a response that doesn't conform to the requested format. If this occurs, we show an error message to the agent indicating that the prior message was malformed and to issue another response that does not make the same mistake, as presented in Figure 32. If a model generates 3 malformed responses in a row, the episode will terminate early.

Figure 32: The environment's error message. This is emitted if a model generation doesn't conform to the thought-action format suggested.

Another context management trick is that if models generate a malformed response, but then subsequently respond with a valid one, the message history is modified such that the action and response correspond to the malformed generation is removed. This kind of de-noising reduces unnecessary context and helps prevent future malformed generations. Each well-formatted response becomes an additional in-context demonstration of how to interact with the environment correctly; this "momentum" of correct responses is effective for helping agents continue to issue correct actions at later turns in trajectories when there is a lot of information in the message history.

# D  Qualitative Analysis

We choose 2 successful and unsuccessful task instances from SWE-bench Lite to perform a full qualitative analysis, based on the trajectories generated by the default configuration using GPT-4 Turbo.

**Analysis for psf/requests-2317**

**Issue Summary.** The agent was tasked with resolving a problem in the `requests` library where the `method` variable, a bytes object (e.g., `b'GET'`), was being incorrectly converted to a literal string (`"b'GET'"`) by the `builtin_str` function. This caused the `requests` library to raise a `404 Not Found` response due to the incorrect method string. The issue was observed in Python 3.4 and not in Python 2.6, indicating a version-specific difference in handling bytes.

**Interaction Summary**

1. **Locate the Target File**: The agent began by finding the `sessions.py` file in the `requests` library directory. It correctly identified that the relevant file was in the `requests` directory rather than the build directory.

2. **Search for the Issue**: It opened `sessions.py` and used the `search_file` command to locate the `builtin_str` function call. It found the relevant line (line 428) where the `method` variable was being converted.

3. **Edit the Code**: The agent edited the `sessions.py` file to check if the `method` was a bytes object and, if so, decoded it to ASCII before applying the `builtin_str` function.

4. **Test Changes**: To verify the fix, the agent created a new file, `reproduce_issue.py`, which attempted to send an HTTP request using a binary method string. The script printed the response status code to check if the request was handled correctly. It ran the `reproduce_issue.py` script, which executed successfully and received a `200 OK` response, indicating that the issue was resolved.

5. **Clean Up and Submit**: Finally, the agent deleted the reproduction script and submitted the changes to the code base.

**Evaluation.** Upon evaluation, the agent's proposed changes successfully pass the instance's unit tests. When comparing the agent's generation to the 'gold patch', we notice that the gold patch uses the `to_native_string` function, defined in [requests/utils.py#L655](requests/utils.py#L655), which behaves identically to the agent's proposed changes. Ultimately, while the agent successfully resolves the issue, it fails to use existing utility functions, which could lead to less maintainable or portable code in production.

Figure 33: Trajectory for instance `psf__requests-2317`.

---

| **Issue** | **psf__requests-2317** |
| --- | --- |

We're currently solving the following issue within our repository. Here's the issue text:
ISSUE:
method = builtin_str(method) problem
In requests/sessions.py is a command:
method = builtin_str(method)
Converts method from
b'GET'
to
"b'GET'"
Which is the literal string, no longer a binary string. When requests tries to use the method "b'GET'", it gets a 404 Not Found response.
I am using python3.4 and python-neutronclient (2.3.9) with requests (2.4.3). neutronclient is broken because it uses this "args = utils.safe_encode_list(args)" command which converts all the values to binary string, including method.
I'm not sure if this is a bug with neutronclient or a bug with requests, but I'm starting here. Seems if requests handled the method value being a binary string, we wouldn't have any problem.
Also, I tried in python2.6 and this bug doesn't exist there. Some difference between 2.6 and 3.4 makes this not work right.

INSTRUCTIONS: (31 lines)

```
(Open file: n/a)
(Current directory: /psf__requests)
bash-\$
```

**SWE-agent**                                                    psf__requests-2317

To address the issue, we first need to replicate the behavior described. The issue mentions that the 'builtin_str' function in 'requests/sessions.py' is converting a binary string method like 'b'GET'' into the literal string '"b'GET'"', which is incorrect. We should start by locating the 'sessions.py' file within the 'requests' library to examine the code around the 'builtin_str' function call.
Let's begin by finding the 'sessions.py' file in the current repository.
────────────────────────────────────────────────────────────────────

```
find_file "sessions.py"
```

**Observation**                                                  psf__requests-2317

```
Found 2 matches for "sessions.py" in /psf__requests:
/psf__requests/build/lib/requests/sessions.py
/psf__requests/requests/sessions.py

(Open file: n/a)
(Current directory: /psf__requests)
bash-$
```

**SWE-agent**                                                    psf__requests-2317

We have found two matches for 'sessions.py'. The one in the 'build/lib/requests' directory is likely a build artifact, and the one we're interested in is probably '/psf__requests/requests/sessions.py'. Let's open this file to inspect the code around the 'builtin_str' function call.
────────────────────────────────────────────────────────────────────

```
open requests/sessions.py
```

**Observation**                                                  psf__requests-2317

```
[File: /psf__requests/requests/sessions.py (671 lines total)]
1:# -*- coding: utf-8 -*-
2:
3:"""
4:requests.session
5:~~~~~~~~~~~~~~~~~
6:
7:This module provides a Session object to manage and persist
↪   settings across
8:requests (cookies, auth, proxies).
9:
10:"""
11:import os
12:from collections import Mapping
13:from datetime import datetime
14:
15:from .auth import _basic_auth_str
16:from .compat import cookielib, OrderedDict, urljoin, urlparse,
↪   builtin_str
```

```
17:from .cookies import (
18:    cookiejar_from_dict, extract_cookies_to_jar,
↪  RequestsCookieJar, merge_cookies)
19:from .models import Request, PreparedRequest,
↪  DEFAULT_REDIRECT_LIMIT
20:from .hooks import default_hooks, dispatch_hook
21:from .utils import to_key_val_list, default_headers,
↪  to_native_string
22:from .exceptions import (
23:    TooManyRedirects, InvalidSchema, ChunkedEncodingError,
↪  ContentDecodingError)
24:from .structures import CaseInsensitiveDict
25:
26:from .adapters import HTTPAdapter
27:
28:from .utils import (
29:    requote_uri, get_environ_proxies, get_netrc_auth,
↪  should_bypass_proxies,
30:    get_auth_from_url
31:)
32:
33:from .status_codes import codes
34:
35:# formerly defined here, reexposed here for backward
↪  compatibility
36:from .models import REDIRECT_STATI
37:
38:
39:def merge_setting(request_setting, session_setting,
↪  dict_class=OrderedDict):
40:    """
41:    Determines appropriate setting for a given request, taking
↪  into account the
42:    explicit setting on that request, and the setting in the
↪  session. If a
43:    setting is a dictionary, they will be merged together using
↪  `dict_class`
44:    """
45:
46:    if session_setting is None:
47:        return request_setting
48:
49:    if request_setting is None:
50:        return session_setting
51:
52:    # Bypass if not a dictionary (e.g. verify)
53:    if not (
54:            isinstance(session_setting, Mapping) and
55:            isinstance(request_setting, Mapping)
56:    ):
57:        return request_setting
58:
59:    merged_setting =
↪  dict_class(to_key_val_list(session_setting))
60:    merged_setting.update(to_key_val_list(request_setting))
61:
62:    # Remove keys that are set to None.
63:    for (k, v) in request_setting.items():
64:        if v is None:
65:            del merged_setting[k]
66:
67:    merged_setting = dict((k, v) for (k, v) in
↪  merged_setting.items() if v is not None)
```

```
68:
69:    return merged_setting
70:
71:
72:def merge_hooks(request_hooks, session_hooks,
↪  dict_class=OrderedDict):
73:    """
74:    Properly merges both requests and session hooks.
75:
76:    This is necessary because when request_hooks == {'response':
↪  []}, the
77:    merge breaks Session hooks entirely.
78:    """
79:    if session_hooks is None or session_hooks.get('response') ==
↪  []:
80:        return request_hooks
81:
82:    if request_hooks is None or request_hooks.get('response') ==
↪  []:
83:        return session_hooks
84:
85:    return merge_setting(request_hooks, session_hooks,
↪  dict_class)
86:
87:
88:class SessionRedirectMixin(object):
89:    def resolve_redirects(self, resp, req, stream=False,
↪  timeout=None,
90:                          verify=True, cert=None, proxies=None):
91:        """Receives a Response. Returns a generator of
↪  Responses."""
92:
93:        i = 0
94:        hist = [] # keep track of history
95:
96:        while resp.is_redirect:
97:            prepared_request = req.copy()
98:
99:            if i > 0:
100:                # Update history and keep track of redirects.
(571 more lines below)

(Open file: n/a)
(Current directory: /psf__requests)
bash-$
```

The 'sessions.py' file is now open, but the 'builtin_str' function call is not visible in the first 100 lines of the file. Since we don't have the exact line number where the 'builtin_str' function is called, we'll need to search within the file for it.
Let's use the 'search_file' command to find the usage of 'builtin_str' within 'sessions.py'.

```
search_file "builtin_str"
```

```
Found 2 matches for "builtin_str" in
↪  /psf__requests/requests/sessions.py:
Line 16:from .compat import cookielib, OrderedDict, urljoin,
↪  urlparse, builtin_str
Line 428:       method = builtin_str(method)
End of matches for "builtin_str" in
↪  /psf__requests/requests/sessions.py

(Open file: /psf__requests/requests/sessions.py)
(Current directory: /psf__requests)
bash-$
```

We have found two occurrences of 'builtin_str' in 'sessions.py'. The second occurrence on line 428 is where the method is being converted to a string. We should inspect this line to understand the context and see how we can address the issue.

Let's navigate to line 428 in 'sessions.py' to see the relevant code.

```
goto 428
```

```
[File: /psf__requests/requests/sessions.py (671 lines total)]
(412 more lines above)
413:          data before giving up, as a float, or a (`connect
↪  timeout, read
414:          timeout <user/advanced.html#timeouts>`_) tuple.
415:      :type timeout: float or tuple
416:      :param allow_redirects: (optional) Set to True by
↪  default.
417:      :type allow_redirects: bool
418:      :param proxies: (optional) Dictionary mapping protocol
↪  to the URL of
419:          the proxy.
420:      :param stream: (optional) whether to immediately
↪  download the response
421:          content. Defaults to ``False``.
422:      :param verify: (optional) if ``True``, the SSL cert will
↪  be verified.
423:          A CA_BUNDLE path can also be provided.
424:      :param cert: (optional) if String, path to ssl client
↪  cert file (.pem).
425:          If Tuple, ('cert', 'key') pair.
426:      """
427:
428:      method = builtin_str(method)
429:
430:      # Create the Request.
431:      req = Request(
432:          method = method.upper(),
433:          url = url,
434:          headers = headers,
435:          files = files,
436:          data = data or {},
437:          json = json,
438:          params = params or {},
```

```
439:              auth = auth,
440:              cookies = cookies,
441:              hooks = hooks,
442:          )
443:          prep = self.prepare_request(req)
444:
445:          proxies = proxies or {}
446:
447:          settings = self.merge_environment_settings(
448:              prep.url, proxies, stream, verify, cert
449:          )
450:
451:          # Send the request.
452:          send_kwargs = {
453:              'timeout': timeout,
454:              'allow_redirects': allow_redirects,
455:          }
456:          send_kwargs.update(settings)
457:          resp = self.send(prep, **send_kwargs)
458:
459:          return resp
460:
461:      def get(self, url, **kwargs):
462:          """Sends a GET request. Returns :class:`Response`
↪  object.
463:
464:          :param url: URL for the new :class:`Request` object.
465:          :param \*\*kwargs: Optional arguments that ``request``
↪  takes.
466:          """
467:
468:          kwargs.setdefault('allow_redirects', True)
469:          return self.request('GET', url, **kwargs)
470:
471:      def options(self, url, **kwargs):
472:          """Sends a OPTIONS request. Returns :class:`Response`
↪  object.
473:
474:          :param url: URL for the new :class:`Request` object.
475:          :param \*\*kwargs: Optional arguments that ``request``
↪  takes.
476:          """
477:
478:          kwargs.setdefault('allow_redirects', True)
479:          return self.request('OPTIONS', url, **kwargs)
480:
481:      def head(self, url, **kwargs):
482:          """Sends a HEAD request. Returns :class:`Response`
↪  object.
483:
484:          :param url: URL for the new :class:`Request` object.
485:          :param \*\*kwargs: Optional arguments that ``request``
↪  takes.
486:          """
487:
488:          kwargs.setdefault('allow_redirects', False)
489:          return self.request('HEAD', url, **kwargs)
490:
491:      def post(self, url, data=None, json=None, **kwargs):
492:          """Sends a POST request. Returns :class:`Response`
↪  object.
493:
494:          :param url: URL for the new :class:`Request` object.
```

```
495:         :param data: (optional) Dictionary, bytes, or file-like
↪   object to send in the body of the :class:`Request`.
496:         :param json: (optional) json to send in the body of the
↪   :class:`Request`.
497:         :param \*\*kwargs: Optional arguments that ``request``
↪   takes.
498:         """
499:
500:         return self.request('POST', url, data=data, json=json,
↪   **kwargs)
501:
502:     def put(self, url, data=None, **kwargs):
503:         """Sends a PUT request. Returns :class:`Response`
↪   object.
504:
505:         :param url: URL for the new :class:`Request` object.
506:         :param data: (optional) Dictionary, bytes, or file-like
↪   object to send in the body of the :class:`Request`.
507:         :param \*\*kwargs: Optional arguments that ``request``
↪   takes.
508:         """
509:
510:         return self.request('PUT', url, data=data, **kwargs)
511:
512:     def patch(self, url, data=None, **kwargs):
(159 more lines below)

(Open file: /psf__requests/requests/sessions.py)
(Current directory: /psf__requests)
bash-$
```

We are now looking at the section of the 'sessions.py' file where the 'builtin_str' function is used to convert the 'method' variable. According to the issue, this conversion is causing a problem when the 'method' is a binary string because it's being converted to a literal string that includes the binary prefix 'b'...'`.

To address this issue, we need to ensure that the 'method' variable is correctly converted to a string without the binary prefix. This might involve checking the type of the 'method' variable and handling binary strings differently from regular strings.

Let's edit the line to add a check that ensures binary strings are decoded to regular strings before being passed to 'builtin_str'.

```
edit 428:428
        # Ensure method is a string, decode if it's a binary string
        if isinstance(method, bytes):
            method = method.decode('ascii')
        method = builtin_str(method)
end_of_edit
```

```
[File: /psf__requests/requests/sessions.py (674 lines total)]
(377 more lines above)
378:     def request(self, method, url,
379:         params=None,
380:         data=None,
381:         headers=None,
382:         cookies=None,
```

```
383:         files=None,
384:         auth=None,
385:         timeout=None,
386:         allow_redirects=True,
387:         proxies=None,
388:         hooks=None,
389:         stream=None,
390:         verify=None,
391:         cert=None,
392:         json=None):
393:         """Constructs a :class:`Request <Request>`, prepares it
↪  and sends it.
394:         Returns :class:`Response <Response>` object.
395:
396:         :param method: method for the new :class:`Request`
↪  object.
397:         :param url: URL for the new :class:`Request` object.
398:         :param params: (optional) Dictionary or bytes to be
↪  sent in the query
399:             string for the :class:`Request`.
400:         :param data: (optional) Dictionary or bytes to send in
↪  the body of the
401:             :class:`Request`.
402:         :param json: (optional) json to send in the body of the
403:             :class:`Request`.
404:         :param headers: (optional) Dictionary of HTTP Headers
↪  to send with the
405:             :class:`Request`.
406:         :param cookies: (optional) Dict or CookieJar object to
↪  send with the
407:             :class:`Request`.
408:         :param files: (optional) Dictionary of ``'filename':
↪  file-like-objects``
409:             for multipart encoding upload.
410:         :param auth: (optional) Auth tuple or callable to
↪  enable
411:             Basic/Digest/Custom HTTP Auth.
412:         :param timeout: (optional) How long to wait for the
↪  server to send
413:             data before giving up, as a float, or a (`connect
↪  timeout, read
414:             timeout <user/advanced.html#timeouts>`_) tuple.
415:         :type timeout: float or tuple
416:         :param allow_redirects: (optional) Set to True by
↪  default.
417:         :type allow_redirects: bool
418:         :param proxies: (optional) Dictionary mapping protocol
↪  to the URL of
419:             the proxy.
420:         :param stream: (optional) whether to immediately
↪  download the response
421:             content. Defaults to ``False``.
422:         :param verify: (optional) if ``True``, the SSL cert will
↪  be verified.
423:             A CA_BUNDLE path can also be provided.
424:         :param cert: (optional) if String, path to ssl client
↪  cert file (.pem).
425:             If Tuple, ('cert', 'key') pair.
426:         """
427:
428:         # Ensure method is a string, decode if it's a binary
↪  string
429:         if isinstance(method, bytes):
```

```
430:            method = method.decode('ascii')
431:         method = builtin_str(method)
432:
433:         # Create the Request.
434:         req = Request(
435:             method = method.upper(),
436:             url = url,
437:             headers = headers,
438:             files = files,
439:             data = data or {},
440:             json = json,
441:             params = params or {},
442:             auth = auth,
443:             cookies = cookies,
444:             hooks = hooks,
445:         )
446:         prep = self.prepare_request(req)
447:
448:         proxies = proxies or {}
449:
450:         settings = self.merge_environment_settings(
451:             prep.url, proxies, stream, verify, cert
452:         )
453:
454:         # Send the request.
455:         send_kwargs = {
456:             'timeout': timeout,
457:             'allow_redirects': allow_redirects,
458:         }
459:         send_kwargs.update(settings)
460:         resp = self.send(prep, **send_kwargs)
461:
462:         return resp
463:
464:     def get(self, url, **kwargs):
465:         """Sends a GET request. Returns :class:`Response`
↪  object.
466:
467:         :param url: URL for the new :class:`Request` object.
468:         :param \*\*kwargs: Optional arguments that ``request``
↪  takes.
469:         """
470:
471:         kwargs.setdefault('allow_redirects', True)
472:         return self.request('GET', url, **kwargs)
473:
474:     def options(self, url, **kwargs):
475:         """Sends a OPTIONS request. Returns :class:`Response`
↪  object.
476:
477:         :param url: URL for the new :class:`Request` object.
(197 more lines below)
File updated. Please review the changes and make sure they are
↪  correct (correct indentation, no duplicate lines, etc). Edit
↪  the file again if necessary.

(Open file: /psf__requests/requests/sessions.py)
(Current directory: /psf__requests)
bash-$
```

**SWE-agent**                                                    **psf__requests-2317**

The edit has been made to decode binary strings before converting them to a string with 'builtin_str'. This should prevent the issue where a binary string like 'b'GET'' is incorrectly converted to the literal string "'b'GET''".
To verify that this change resolves the issue, we should create a script that replicates the problem as described. We'll write a script that uses the 'requests' library to make a request with a binary method and check if it now handles the method correctly.
Let's create a new file called 'reproduce_issue.py' to test the fix.

───────────────────────────────────────────────────────

```
create reproduce_issue.py
```

---

**Observation**                                                 **psf__requests-2317**

```
[File: /psf__requests/reproduce_issue.py (1 lines total)]
1:

(Open file: /psf__requests/requests/sessions.py)
(Current directory: /psf__requests)
bash-$
```

---

**SWE-agent**                                                    **psf__requests-2317**

Now that we have created the 'reproduce_issue.py' file, let's write a script that attempts to send a request with a binary method string, which should now be handled correctly due to our fix in 'sessions.py'.
We'll write a simple script that imports the 'requests' library, sets a binary method, and tries to make a request to a test URL.

───────────────────────────────────────────────────────

```
edit 1:1
import requests

# Define a binary method string
method = b'GET'

# Test URL (example)
url = 'https://httpbin.org/get'

# Attempt to send a request with a binary method
response = requests.request(method, url)

# Print the response status code
print('Response status code:', response.status_code)

# Print a success message if the script completes
print("Script completed successfully, no errors.")
end_of_edit
```

---

**Observation**                                                 **psf__requests-2317**

```
[File: /psf__requests/reproduce_issue.py (16 lines total)]
1:import requests
2:
3:# Define a binary method string
4:method = b'GET'
5:
6:# Test URL (example)
7:url = 'https://httpbin.org/get'
```

```
8:
9:# Attempt to send a request with a binary method
10:response = requests.request(method, url)
11:
12:# Print the response status code
13:print('Response status code:', response.status_code)
14:
15:# Print a success message if the script completes
16:print("Script completed successfully, no errors.")
File updated. Please review the changes and make sure they are
↪  correct (correct indentation, no duplicate lines, etc). Edit
↪  the file again if necessary.

(Open file: /psf__requests/reproduce_issue.py)
(Current directory: /psf__requests)
bash-$
```

---

**SWE-agent**                                                         **psf__requests-2317**

The script has been written and should now attempt to send a request using a binary method string. This will help us verify if the issue has been resolved by the changes we made to 'sessions.py'.
Let's run the script to see if it executes successfully without encountering the issue described.

```
python reproduce_issue.py
```

---

**Observation**                                                       **psf__requests-2317**

```
Response status code: 200
Script completed successfully, no errors.

(Open file: /psf__requests/reproduce_issue.py)
(Current directory: /psf__requests)
bash-$
```

---

**SWE-agent**                                                         **psf__requests-2317**

The script ran successfully and received a 200 OK response, which indicates that the issue with handling binary method strings has been resolved. The 'requests' library is now correctly processing the binary string 'b'GET'' as a valid HTTP method.
Since the fix appears to be working, we can clean up by removing the 'reproduce_issue.py' script, as it is no longer needed. After that, we can submit our changes to the code base.
Let's remove the 'reproduce_issue.py' file.

```
rm reproduce_issue.py
```

---

**Observation**                                                       **psf__requests-2317**

```
(Open file: /psf__requests/reproduce_issue.py)
(Current directory: /psf__requests)
bash-$
```

**SWE-agent**                                                    **psf__requests-2317**

The 'reproduce_issue.py' file has been successfully removed. Now that we have verified the fix and cleaned up the test script, we can proceed to submit our changes to the code base.
Let's submit the changes using the 'submit' command.

```
submit
```

---

**Observation**                                                 **psf__requests-2317**

```
diff --git a/requests/sessions.py b/requests/sessions.py
index d701ff2e..4829747d 100644
--- a/requests/sessions.py
+++ b/requests/sessions.py
@@ -425,6 +425,9 @@ class Session(SessionRedirectMixin):
             If Tuple, ('cert', 'key') pair.
         """

+        # Ensure method is a string, decode if it's a binary
↪  string
+        if isinstance(method, bytes):
+            method = method.decode('ascii')
         method = builtin_str(method)

         # Create the Request.

(Open file: /psf__requests/reproduce_issue.py)
(Current directory: /psf__requests)
bash-\$
```

---

**Gold Patch**                                                  **psf__requests-2317**

```
diff --git a/requests/sessions.py b/requests/sessions.py
--- a/requests/sessions.py
+++ b/requests/sessions.py
@@ -13,7 +13,7 @@
 from datetime import datetime

 from .auth import _basic_auth_str
-from .compat import cookielib, OrderedDict, urljoin, urlparse,
↪  builtin_str
+from .compat import cookielib, OrderedDict, urljoin, urlparse
 from .cookies import (
     cookiejar_from_dict, extract_cookies_to_jar,
      ↪  RequestsCookieJar, merge_cookies)
 from .models import Request, PreparedRequest,
  ↪  DEFAULT_REDIRECT_LIMIT
@@ -425,7 +425,7 @@ def request(self, method, url,
             If Tuple, ('cert', 'key') pair.
         """

-        method = builtin_str(method)
+        method = to_native_string(method)

         # Create the Request.
         req = Request(
```

**Analysis for `pylint-dev/pylint-5859`**

**Issue Summary.** The issue involves the `–notes` option in `pylint`, which fails to report fixme warnings (W0511) for note tags that are entirely punctuation. Specifically, when running `pylint` with the `–notes="YES,???"` option, only the "YES" tag is recognized, and the "???" tag is ignored. The expected behavior is to recognize the punctuation-only tag normally.

**Interaction Summary**

1. **Reproducing the Issue**:
   - The agent started by creating a `test.py` file and added comments to replicate the issue described. This was followed by running `pylint` to confirm the issue, which only reported the "YES" note and not the "???" note.

2. **Searching for Relevant Code**:
   - The agent attempted to search for the handling of the `–notes` option using the `search_dir` command. After some initial confusion with the command options, it successfully identified the `pylint/checkers/misc.py` file as containing the relevant code.

3. **Editing the Code**:
   - The agent located the regular expression responsible for matching note tags within the `open` method of the `EncodingChecker` class. It modified the regular expression to handle punctuation-only note tags by replacing the word boundary `\b` with a lookahead assertion `(?\W|$)`.

4. **Test Changes with Reproduction Code**:
   - After making the modification, the agent re-ran `pylint` on the `test.py` file. The output confirmed that both note tags ("YES" and "???") were now correctly identified.
   - **Clean Up and Submit**:
   - Finally, the agent removed the `test.py` file and submitted the changes.

**Evaluation** The agent effectively replicated the issue, identified and navigated to the relevant code, made a targeted modification to the regular expression, and verified the fix through testing. While the agent's solution successfully passes the necessary tests, there are slight differences in behavior between the agent's solution and the gold patch. In particular, the agent replaced the `\b` word boundary in the regular expression to handle note tags with the regex pattern `(\W|$)` to mark the tag boundary. Comparing this to the gold patch solution which uses the pattern `(:|\s|\Z)` in the same way, it may be that the agent's solution causes other unintended consequences that aren't captured by the existing tests.

Figure 34: Trajectory for instance `pylint-dev__pylint-5859`.

---

| **Issue** | **pylint-dev__pylint-5859** |
| --- | --- |

We're currently solving the following issue within our repository. Here's the issue text:
ISSUE:
"–notes" option ignores note tags that are entirely punctuation
**Bug description**
If a note tag specified with the '–notes' option is entirely punctuation, pylint won't report a fixme warning (W0511).

```
YES: yes
???: no
```

`pylint test.py –notes="YES,???"` will return a fixme warning (W0511) for the first line, but not the second.
**Configuration**
ini

```
Default
```

## Command used

```
shell
```

---

pylint test.py –notes="YES,???"

### Pylint output

shell

```
************* Module test
test.py:1:1: W0511: YES: yes (fixme)
```

## Expected behavior

```
************* Module test
test.py:1:1: W0511: YES: yes (fixme)
test.py:2:1: W0511: ???: no (fixme)
```

## Pylint version

shell

pylint 2.12.2
astroid 2.9.0
Python 3.10.2 (main, Feb 2 2022, 05:51:25) [Clang 13.0.0 (clang-1300.0.29.3)]

```
### OS / Environment

macOS 11.6.1

### Additional dependencies

_No response_

INSTRUCTIONS: (31 lines)
```

(Open file: n/a)
(Current directory: /pylint)
bash-$

---

To begin addressing the issue, we should first try to replicate the bug as described. We can do this by creating a new Python script that includes the comments with the note tags "YES" and "???" and then running pylint with the '–notes' option. Let's start by creating a new file called 'test.py' and adding the relevant comments to it.

---

```
create test.py
```

---

```
[File: /pylint-dev__pylint/test.py (1 lines total)]
1:

(Open file: n/a)
(Current directory: /pylint-dev__pylint)
bash-$
```

**SWE-agent**                                           **pylint-dev__pylint-5859**

Now let's add the example comments from the issue into 'test.py'.
_______________________________________________________________________

```
edit 1:1
# YES: yes
# ???: no
end_of_edit
```

---

**Observation**                                         **pylint-dev__pylint-5859**

```
[File: /pylint-dev__pylint/test.py (2 lines total)]
1:# YES: yes
2:# ???: no
File updated. Please review the changes and make sure they are
↪  correct (correct indentation, no duplicate lines, etc). Edit
↪  the file again if necessary.

(Open file: /pylint-dev__pylint/test.py)
(Current directory: /pylint-dev__pylint)
bash-$
```

---

**SWE-agent**                                           **pylint-dev__pylint-5859**

The 'test.py' file has been updated with the example comments. Next, we should run pylint with the '–notes' option to see if we can replicate the issue where the second line with punctuation-only tags is ignored.
_______________________________________________________________________

```
pylint test.py --notes="YES,???"
```

---

**Observation**                                         **pylint-dev__pylint-5859**

```
************* Module test
test.py:1:1: W0511: YES: yes (fixme)

(Open file: /pylint-dev__pylint/test.py)
(Current directory: /pylint-dev__pylint)
bash-$
```

---

**SWE-agent**                                           **pylint-dev__pylint-5859**

The output confirms the issue: pylint only reports a fixme warning for the first line with the "YES" tag and ignores the second line with the "???" tag. This matches the behavior described in the issue.
The next step is to locate the part of the pylint code that handles the '–notes' option and determine why it's ignoring note tags that are entirely punctuation. We should start by finding the file that contains the relevant code. Since we don't know the exact file name, we can search the directory for files that might contain the string '–notes' to find where the option is handled.
_______________________________________________________________________

```
search_dir "--notes"
```

**Observation**          **pylint-dev__pylint-5859**

```
grep: unrecognized option '--notes'
Usage: grep [OPTION]... PATTERNS [FILE]...
Try 'grep --help' for more information.
grep: unrecognized option '--notes'
Usage: grep [OPTION]... PATTERNS [FILE]...
Try 'grep --help' for more information.
No matches found for "--notes" in /pylint-dev__pylint

(Open file: /pylint-dev__pylint/test.py)
(Current directory: /pylint-dev__pylint)
bash-$
```

**SWE-agent**          **pylint-dev__pylint-5859**

It seems that the 'search_dir' command did not work as expected because '–notes' was interpreted as an option for 'grep' rather than a search term. To avoid this issue, we can enclose the search term in quotes to ensure it's treated as a literal string. Let's try searching again using the correct format for the search term.

———————————————————————————————————————————

```
search_dir "--notes"
```

**Observation**          **pylint-dev__pylint-5859**

```
grep: unrecognized option '--notes'
Usage: grep [OPTION]... PATTERNS [FILE]...
Try 'grep --help' for more information.
grep: unrecognized option '--notes'
Usage: grep [OPTION]... PATTERNS [FILE]...
Try 'grep --help' for more information.
No matches found for "--notes" in /pylint-dev__pylint

(Open file: /pylint-dev__pylint/test.py)
(Current directory: /pylint-dev__pylint)
bash-$
```

**SWE-agent**          **pylint-dev__pylint-5859**

It appears that the issue persists because the search term is still being interpreted as an option. To circumvent this, we can use a different search term that is less likely to be confused with a command-line option. Let's try searching for just "notes" instead, which should still lead us to the relevant part of the code.

———————————————————————————————————————————

```
search_dir "notes"
```

**Observation**          **pylint-dev__pylint-5859**

```
Found 24 matches for "notes" in /pylint-dev__pylint:
/pylint-dev__pylint/ChangeLog (2 matches)
/pylint-dev__pylint/doc/development_guide/testing.rst (1 matches)
/pylint-dev__pylint/doc/whatsnew/2.5.rst (1 matches)
/pylint-dev__pylint/examples/pylintrc (2 matches)
/pylint-dev__pylint/pylint/checkers/misc.py (9 matches)
/pylint-dev__pylint/pylintrc (2 matches)
```

```
/pylint-dev__pylint/script/bump_changelog.py (1 matches)
/pylint-dev__pylint/tests/checkers/unittest_misc.py (2 matches)
/pylint-dev__pylint/tests/functional/f/fixme.py (1 matches)
/pylint-dev__pylint/tests/functional/f/fixme.rc (2 matches)
/pylint-dev__pylint/tests/functional/f/fixme.txt (1 matches)
End of matches for "notes" in /pylint-dev__pylint

(Open file: /pylint-dev__pylint/test.py)
(Current directory: /pylint-dev__pylint)
bash-$
```

---

**SWE-agent**                                    **pylint-dev__pylint-5859**

The search results indicate that there are several matches for "notes" in the repository, but the most likely candidate for the code we need to examine is in 'pylint/checkers/misc.py', as it contains the most matches. This file is probably where the miscellaneous checkers, including the fixme warning (W0511), are implemented.
Let's open the 'misc.py' file to investigate further.

---

```
open pylint/checkers/misc.py
```

---

**Observation**                                  **pylint-dev__pylint-5859**

```
[File: /pylint-dev__pylint/pylint/checkers/misc.py (203 lines
↪  total)]
1:# Copyright (c) 2006, 2009-2013 LOGILAB S.A. (Paris, FRANCE)
↪  <contact@logilab.fr>
2:# Copyright (c) 2012-2014 Google, Inc.
3:# Copyright (c) 2014-2020 Claudiu Popa <pcmanticore@gmail.com>
4:# Copyright (c) 2014 Brett Cannon <brett@python.org>
5:# Copyright (c) 2014 Alexandru Coman <fcoman@bitdefender.com>
6:# Copyright (c) 2014 Arun Persaud <arun@nubati.net>
7:# Copyright (c) 2015 Ionel Cristian Maries <contact@ionelmc.ro>
8:# Copyright (c) 2016 Łukasz Rogalski <rogalski.91@gmail.com>
9:# Copyright (c) 2016 glegoux <gilles.legoux@gmail.com>
10:# Copyright (c) 2017-2020 hippo91 <guillaume.peillex@gmail.com>
11:# Copyright (c) 2017 Mikhail Fesenko <proggga@gmail.com>
12:# Copyright (c) 2018 Rogalski, Lukasz
↪  <lukasz.rogalski@intel.com>
13:# Copyright (c) 2018 Lucas Cimon <lucas.cimon@gmail.com>
14:# Copyright (c) 2018 Ville Skyttä <ville.skytta@iki.fi>
15:# Copyright (c) 2019-2021 Pierre Sassoulas
↪  <pierre.sassoulas@gmail.com>
16:# Copyright (c) 2020 wtracy <afishionado@gmail.com>
17:# Copyright (c) 2020 Anthony Sottile <asottile@umich.edu>
18:# Copyright (c) 2020 Benny <benny.mueller91@gmail.com>
19:# Copyright (c) 2021 Daniël van Noord
↪  <13665637+DanielNoord@users.noreply.github.com>
20:# Copyright (c) 2021 Nick Drozd <nicholasdrozd@gmail.com>
21:# Copyright (c) 2021 Marc Mueller
↪  <30130371+cdce8p@users.noreply.github.com>
22:# Copyright (c) 2021 Konstantina Saketou
↪  <56515303+ksaketou@users.noreply.github.com>
23:
24:# Licensed under the GPL:
↪  https://www.gnu.org/licenses/old-licenses/gpl-2.0.html
25:# For details: https://github.com/PyCQA/pylint/blob/main/LICENSE
26:
27:
```

```
28:"""Check source code is ascii only or has an encoding
↪  declaration (PEP 263)."""
29:
30:import re
31:import tokenize
32:from typing import TYPE_CHECKING, List, Optional
33:
34:from astroid import nodes
35:
36:from pylint.checkers import BaseChecker
37:from pylint.interfaces import IRawChecker, ITokenChecker
38:from pylint.typing import ManagedMessage
39:from pylint.utils.pragma_parser import OPTION_PO,
↪  PragmaParserError, parse_pragma
40:
41:if TYPE_CHECKING:
42:    from pylint.lint import PyLinter
43:
44:
45:class ByIdManagedMessagesChecker(BaseChecker):
46:
47:    """Checks for messages that are enabled or disabled by id
↪  instead of symbol."""
48:
49:    __implements__ = IRawChecker
50:    name = "miscellaneous"
51:    msgs = {
52:        "I0023": (
53:            "%s",
54:            "use-symbolic-message-instead",
55:            "Used when a message is enabled or disabled by id.",
56:        )
57:    }
58:    options = ()
59:
60:    def _clear_by_id_managed_msgs(self) -> None:
61:        self.linter._by_id_managed_msgs.clear()
62:
63:    def _get_by_id_managed_msgs(self) -> List[ManagedMessage]:
64:        return self.linter._by_id_managed_msgs
65:
66:    def process_module(self, node: nodes.Module) -> None:
67:        """Inspect the source file to find messages activated or
↪  deactivated by id."""
68:        managed_msgs = self._get_by_id_managed_msgs()
69:        for (mod_name, msgid, symbol, lineno, is_disabled) in
↪  managed_msgs:
70:            if mod_name == node.name:
71:                verb = "disable" if is_disabled else "enable"
72:                txt = f"'{msgid}' is cryptic: use '# pylint:
↪  {verb}={symbol}' instead"
73:                self.add_message("use-symbolic-message-instead",
↪  line=lineno, args=txt)
74:        self._clear_by_id_managed_msgs()
75:
76:
77:class EncodingChecker(BaseChecker):
78:
79:    """Checks for:
80:    * warning notes in the code like FIXME, XXX
81:    * encoding issues.
82:    """
83:
```

```
84:    __implements__ = (IRawChecker, ITokenChecker)
85:
86:    # configuration section name
87:    name = "miscellaneous"
88:    msgs = {
89:        "W0511": (
90:            "%s",
91:            "fixme",
92:            "Used when a warning note as FIXME or XXX is
↪  detected.",
93:        )
94:    }
95:
96:    options = (
97:        (
98:            "notes",
99:            {
100:                "type": "csv",
(103 more lines below)

(Open file: /pylint-dev__pylint/test.py)
(Current directory: /pylint-dev__pylint)
bash-$
```

We have opened the 'misc.py' file which likely contains the code for the fixme warning (W0511) related to the '–notes' option. Since the file is 203 lines long and we are currently viewing the first 100 lines, we need to scroll down or use the 'goto' command to find the relevant section that handles the '–notes' option.
Let's scroll down to see more of the file.

---

```
scroll_down
```

**Observation**                                  **pylint-dev__pylint-5859**

```
[File: /pylint-dev__pylint/pylint/checkers/misc.py (203 lines
↪  total)]
(98 more lines above)
99:            {
100:                "type": "csv",
101:                "metavar": "<comma separated values>",
102:                "default": ("FIXME", "XXX", "TODO"),
103:                "help": (
104:                    "List of note tags to take in
↪  consideration, "
105:                    "separated by a comma."
106:                ),
107:            },
108:        ),
109:        (
110:            "notes-rgx",
111:            {
112:                "type": "string",
113:                "metavar": "<regexp>",
114:                "help": "Regular expression of note tags to
↪  take in consideration.",
115:            },
116:        ),
```

```
117:     )
118:
119:     def open(self):
120:         super().open()
121:
122:         notes = "|".join(re.escape(note) for note in
↪    self.config.notes)
123:         if self.config.notes_rgx:
124:             regex_string =
↪    rf"#\s*({notes}|{self.config.notes_rgx})\b"
125:         else:
126:             regex_string = rf"#\s*({notes})\b"
127:
128:         self._fixme_pattern = re.compile(regex_string, re.I)
129:
130:     def _check_encoding(
131:         self, lineno: int, line: bytes, file_encoding: str
132:     ) -> Optional[str]:
133:         try:
134:             return line.decode(file_encoding)
135:         except UnicodeDecodeError:
136:             pass
137:         except LookupError:
138:             if (
139:                 line.startswith(b"#")
140:                 and "coding" in str(line)
141:                 and file_encoding in str(line)
142:             ):
143:                 msg = f"Cannot decode using encoding
↪    '{file_encoding}', bad encoding"
144:                 self.add_message("syntax-error", line=lineno,
↪    args=msg)
145:         return None
146:
147:     def process_module(self, node: nodes.Module) -> None:
148:         """Inspect the source file to find encoding problem."""
149:         encoding = node.file_encoding if node.file_encoding
↪    else "ascii"
150:
151:         with node.stream() as stream:
152:             for lineno, line in enumerate(stream):
153:                 self._check_encoding(lineno + 1, line,
↪    encoding)
154:
155:     def process_tokens(self, tokens):
156:         """Inspect the source to find fixme problems."""
157:         if not self.config.notes:
158:             return
159:         comments = (
160:             token_info for token_info in tokens if
↪    token_info.type == tokenize.COMMENT
161:         )
162:         for comment in comments:
163:             comment_text = comment.string[1:].lstrip()  # trim
↪    '#' and whitespaces
164:
165:             # handle pylint disable clauses
166:             disable_option_match =
↪    OPTION_PO.search(comment_text)
167:             if disable_option_match:
168:                 try:
169:                     values = []
170:                     try:
```

```
171:                              for pragma_repr in (
172:                                  p_rep
173:                                  for p_rep in
↪   parse_pragma(disable_option_match.group(2))
174:                                  if p_rep.action == "disable"
175:                              ):
176:                                  values.extend(pragma_repr.messages)
177:                      except PragmaParserError:
178:                          # Printing useful information dealing
↪   with this error is done in the lint package
179:                          pass
180:                      if set(values) & set(self.config.notes):
181:                          continue
182:                  except ValueError:
183:                      self.add_message(
184:                          "bad-inline-option",
185:
↪   args=disable_option_match.group(1).strip(),
186:                          line=comment.start[0],
187:                      )
188:                      continue
189:
190:              # emit warnings if necessary
191:              match = self._fixme_pattern.search("#" +
↪   comment_text.lower())
192:              if match:
193:                  self.add_message(
194:                      "fixme",
195:                      col_offset=comment.start[1] + 1,
196:                      args=comment_text,
197:                      line=comment.start[0],
198:                  )
(5 more lines below)

(Open file: /pylint-dev__pylint/pylint/checkers/misc.py)
(Current directory: /pylint-dev__pylint)
bash-$
```

We have found the relevant section of the 'misc.py' file that handles the '–notes' option. The regular expression pattern used to match note tags is constructed in lines 122-128. It seems that the issue might be related to how the regular expression is constructed or how it is used to match comments in the source code.

The regular expression is built from the 'notes' configuration, which is a list of comma-separated values, and an optional 'notes-rgx' regular expression. The pattern is then compiled and used in the 'process_tokens' method to search for matching comments.

The issue might be that the regular expression pattern does not account for note tags that consist entirely of punctuation characters. To fix this, we may need to adjust the regular expression pattern to ensure it matches these tags correctly.

Before we make any changes, let's examine the regular expression pattern construction in more detail. We should look at the 'open' method where the pattern is compiled, specifically lines 122-128.

_______________________________________________________________

```
goto 122
```

```
[File: /pylint-dev__pylint/pylint/checkers/misc.py (203 lines
↪ total)]
(103 more lines above)
104:                    "List of note tags to take in
↪ consideration, "
105:                    "separated by a comma."
106:                ),
107:            },
108:        ),
109:        (
110:            "notes-rgx",
111:            {
112:                "type": "string",
113:                "metavar": "<regexp>",
114:                "help": "Regular expression of note tags to
↪ take in consideration.",
115:            },
116:        ),
117:    )
118:
119:    def open(self):
120:        super().open()
121:
122:        notes = "|".join(re.escape(note) for note in
↪ self.config.notes)
123:        if self.config.notes_rgx:
124:            regex_string =
↪ rf"#\s*({notes}|{self.config.notes_rgx})\b"
125:        else:
126:            regex_string = rf"#\s*({notes})\b"
127:
128:        self._fixme_pattern = re.compile(regex_string, re.I)
129:
130:    def _check_encoding(
131:        self, lineno: int, line: bytes, file_encoding: str
132:    ) -> Optional[str]:
133:        try:
134:            return line.decode(file_encoding)
135:        except UnicodeDecodeError:
136:            pass
137:        except LookupError:
138:            if (
139:                line.startswith(b"#")
140:                and "coding" in str(line)
141:                and file_encoding in str(line)
142:            ):
143:                msg = f"Cannot decode using encoding
↪ '{file_encoding}', bad encoding"
144:                self.add_message("syntax-error", line=lineno,
↪ args=msg)
145:        return None
146:
147:    def process_module(self, node: nodes.Module) -> None:
148:        """Inspect the source file to find encoding problem."""
149:        encoding = node.file_encoding if node.file_encoding
↪ else "ascii"
150:
151:        with node.stream() as stream:
152:            for lineno, line in enumerate(stream):
153:                self._check_encoding(lineno + 1, line,
↪ encoding)
```

```
154:
155:    def process_tokens(self, tokens):
156:        """Inspect the source to find fixme problems."""
157:        if not self.config.notes:
158:            return
159:        comments = (
160:            token_info for token_info in tokens if
→   token_info.type == tokenize.COMMENT
161:        )
162:        for comment in comments:
163:            comment_text = comment.string[1:].lstrip()  # trim
→   '#' and whitespaces
164:
165:            # handle pylint disable clauses
166:            disable_option_match =
→   OPTION_PO.search(comment_text)
167:            if disable_option_match:
168:                try:
169:                    values = []
170:                    try:
171:                        for pragma_repr in (
172:                            p_rep
173:                            for p_rep in
→   parse_pragma(disable_option_match.group(2))
174:                            if p_rep.action == "disable"
175:                        ):
176:                            values.extend(pragma_repr.messages)
177:                    except PragmaParserError:
178:                        # Printing useful information dealing
→   with this error is done in the lint package
179:                        pass
180:                    if set(values) & set(self.config.notes):
181:                        continue
182:                except ValueError:
183:                    self.add_message(
184:                        "bad-inline-option",
185:
→   args=disable_option_match.group(1).strip(),
186:                        line=comment.start[0],
187:                    )
188:                    continue
189:
190:            # emit warnings if necessary
191:            match = self._fixme_pattern.search("#" +
→   comment_text.lower())
192:            if match:
193:                self.add_message(
194:                    "fixme",
195:                    col_offset=comment.start[1] + 1,
196:                    args=comment_text,
197:                    line=comment.start[0],
198:                )
199:
200:
201:def register(linter: "PyLinter") -> None:
202:    linter.register_checker(EncodingChecker(linter))
203:    linter.register_checker(ByIdManagedMessagesChecker(linter))

(Open file: /pylint-dev__pylint/pylint/checkers/misc.py)
(Current directory: /pylint-dev__pylint)
bash-$
```

We are now looking at the section of the 'misc.py' file where the regular expression for the '–notes' option is constructed. The regular expression is built from the 'notes' configuration option, which is a list of note tags. These tags are escaped using 're.escape' and joined with a pipe '|' to form an alternation pattern. This pattern is then used to create a regular expression that matches comments starting with one of the note tags followed by a word boundary '\b'.

The issue might be with the word boundary '\b', which does not match if the tag is followed by punctuation. Since the tag "???" is entirely punctuation, the word boundary may not be recognized correctly, causing the tag not to match.

To fix this issue, we can modify the regular expression to ensure that it matches note tags that are entirely punctuation. We can do this by removing the word boundary and allowing the tag to be followed by any non-word character or the end of the string.

Let's edit the regular expression construction to fix the issue.

---

```
edit 124:126
            regex_string =
            ↪  rf"#\s*({notes}|{self.config.notes_rgx})(?=\W|$)"
        else:
            regex_string = rf"#\s*({notes})(?=\W|$)"
end_of_edit
```

---

```
[File: /pylint-dev__pylint/pylint/checkers/misc.py (203 lines
↪  total)]
(73 more lines above)
74:         self._clear_by_id_managed_msgs()
75:
76:
77:class EncodingChecker(BaseChecker):
78:
79:     """Checks for:
80:     * warning notes in the code like FIXME, XXX
81:     * encoding issues.
82:     """
83:
84:     __implements__ = (IRawChecker, ITokenChecker)
85:
86:     # configuration section name
87:     name = "miscellaneous"
88:     msgs = {
89:         "W0511": (
90:             "%s",
91:             "fixme",
92:             "Used when a warning note as FIXME or XXX is
↪  detected.",
93:         )
94:     }
95:
96:     options = (
97:         (
98:             "notes",
99:             {
100:                 "type": "csv",
101:                 "metavar": "<comma separated values>",
102:                 "default": ("FIXME", "XXX", "TODO"),
103:                 "help": (
104:                     "List of note tags to take in
↪  consideration, "
```

```
105:                    "separated by a comma."
106:                ),
107:            },
108:        ),
109:        (
110:            "notes-rgx",
111:            {
112:                "type": "string",
113:                "metavar": "<regexp>",
114:                "help": "Regular expression of note tags to
↪  take in consideration.",
115:            },
116:        ),
117:    )
118:
119:    def open(self):
120:        super().open()
121:
122:        notes = "|".join(re.escape(note) for note in
↪  self.config.notes)
123:        if self.config.notes_rgx:
124:            regex_string =
↪  rf"#\s*({notes}|{self.config.notes_rgx})(?=\W|$)"
125:        else:
126:            regex_string = rf"#\s*({notes})(?=\W|$)"
127:
128:        self._fixme_pattern = re.compile(regex_string, re.I)
129:
130:    def _check_encoding(
131:        self, lineno: int, line: bytes, file_encoding: str
132:    ) -> Optional[str]:
133:        try:
134:            return line.decode(file_encoding)
135:        except UnicodeDecodeError:
136:            pass
137:        except LookupError:
138:            if (
139:                line.startswith(b"#")
140:                and "coding" in str(line)
141:                and file_encoding in str(line)
142:            ):
143:                msg = f"Cannot decode using encoding
↪  '{file_encoding}', bad encoding"
144:                self.add_message("syntax-error", line=lineno,
↪  args=msg)
145:        return None
146:
147:    def process_module(self, node: nodes.Module) -> None:
148:        """Inspect the source file to find encoding problem."""
149:        encoding = node.file_encoding if node.file_encoding
↪  else "ascii"
150:
151:        with node.stream() as stream:
152:            for lineno, line in enumerate(stream):
153:                self._check_encoding(lineno + 1, line,
↪  encoding)
154:
155:    def process_tokens(self, tokens):
156:        """Inspect the source to find fixme problems."""
157:        if not self.config.notes:
158:            return
159:        comments = (
```

```
160:             token_info for token_info in tokens if
↪  token_info.type == tokenize.COMMENT
161:         )
162:         for comment in comments:
163:             comment_text = comment.string[1:].lstrip()  # trim
↪  '#' and whitespaces
164:
165:             # handle pylint disable clauses
166:             disable_option_match =
↪  OPTION_PO.search(comment_text)
167:             if disable_option_match:
168:                 try:
169:                     values = []
170:                     try:
171:                         for pragma_repr in (
172:                             p_rep
173:                             for p_rep in
↪  parse_pragma(disable_option_match.group(2))
(30 more lines below)
File updated. Please review the changes and make sure they are
↪  correct (correct indentation, no duplicate lines, etc). Edit
↪  the file again if necessary.

(Open file: /pylint-dev__pylint/pylint/checkers/misc.py)
(Current directory: /pylint-dev__pylint)
bash-$
```

---

**SWE-agent**                                          **pylint-dev__pylint-5859**

The edit to the regular expression has been made successfully. We replaced the word boundary '\b'
with a lookahead assertion '(?=\W|$)' that matches either a non-word character or the end of the string
after the note tag. This should allow the regular expression to match note tags that consist entirely of
punctuation.
Before submitting the changes, we should test to ensure that the fix works as expected. We can do this
by running pylint on the 'test.py' file again with the '--notes' option to see if it now correctly identifies
the "???" note tag.
Let's run pylint on 'test.py' with the updated code.

---

```
pylint test.py --notes="YES,???"
```

---

**Observation**                                        **pylint-dev__pylint-5859**

```
************* Module test
test.py:1:1: W0511: YES: yes (fixme)
test.py:2:1: W0511: ???: no (fixme)

(Open file: /pylint-dev__pylint/pylint/checkers/misc.py)
(Current directory: /pylint-dev__pylint)
bash-$
```

---

**SWE-agent**                                          **pylint-dev__pylint-5859**

The pylint output now shows that it correctly identifies both the "YES" and "???" note tags, which
means the fix has resolved the issue. With the bug fixed, we can proceed to clean up any test files and
then submit our changes to the code base.
First, let's remove the 'test.py' file since it was only needed for reproducing the issue.

---

```
rm test.py
```

```
(Open file: /pylint-dev__pylint/pylint/checkers/misc.py)
(Current directory: /pylint-dev__pylint)
bash-$
```

**SWE-agent**                                      **pylint-dev__pylint-5859**

The 'test.py' file has been successfully removed. Now that we have fixed the issue and cleaned up the test files, we can submit our changes to the code base. Let's submit the changes using the 'submit' command.

---

```
submit
```

```
diff --git a/pylint/checkers/misc.py b/pylint/checkers/misc.py
index 69149e61a..28978365f 100644
--- a/pylint/checkers/misc.py
+++ b/pylint/checkers/misc.py
@@ -121,9 +121,9 @@ class EncodingChecker(BaseChecker):

        notes = "|".join(re.escape(note) for note in
        ↪  self.config.notes)
        if self.config.notes_rgx:
-          regex_string =
↪  rf"#\s*({notes}|{self.config.notes_rgx})\b"
+          regex_string =
↪  rf"#\s*({notes}|{self.config.notes_rgx})(?=\W|$)"
        else:
-          regex_string = rf"#\s*({notes})\b"
+          regex_string = rf"#\s*({notes})(?=\W|$)"

        self._fixme_pattern = re.compile(regex_string, re.I)
```

**Gold Patch**                                     **pylint-dev__pylint-5859**

```
diff --git a/pylint/checkers/misc.py b/pylint/checkers/misc.py
--- a/pylint/checkers/misc.py
+++ b/pylint/checkers/misc.py
@@ -121,9 +121,9 @@ def open(self):

        notes = "|".join(re.escape(note) for note in
        ↪  self.config.notes)
        if self.config.notes_rgx:
-          regex_string =
↪  rf"#\s*({notes}|{self.config.notes_rgx})\b"
+          regex_string =
↪  rf"#\s*({notes}|{self.config.notes_rgx})(?=(:|\s|\Z))"
        else:
-          regex_string = rf"#\s*({notes})\b"
```

```
 +                    regex_string = rf"#\s*({notes})(?=(:|\s|\Z))"

            self._fixme_pattern = re.compile(regex_string, re.I)
```

**Analysis for `sympy/sympy-21614`**

**Issue Summary.** The issue is reporting an `UnedinedKind` type returned when printing a `sympy.Derivative`'s `kind` attribute. They expect something like a `DerivativeKind` instead of `UndefinedKind`.

**Interaction Summary**

1. **Reproduce the Issue**: The agent began by creating a Python script (`reproduce_issue.py`) to reproduce the issue, using the reproduction code provided in the issue report. It ran the script and successfully reproduced the issue.
2. **Searching for Derivative Code**: The agent begins to search for code where the `Derivative` class is defined. It makes several attempts with different strategies to find the relevant code. After a few unsuccessful approaches, looking for particular filenames, it searches simply for "class Derivative", which provides a single search result. It continues to read the `Derivative` class definition, which spans many hundreds of lines.
3. **Exhausting Budget**: It continuously scrolls through the `function.py` file where `Derivative` is defined until is exhausts its budget and terminates the episode without submitting.

**Evaluation** The agent makes a few mistakes using the search function effectively. The agent should have made more efficient searches by using more commonsensical patterns (e.g. searching for "class Derivative") early on, instead of more speculative search attempts. In this particular setting, the gold patch shows that the updated code involves overriding an inherited property `kind` from the **sympy.basic** object. This fix would require a more thorough understanding of the codebase and inheritence structure of `Derivative` to be effective. An agent attempting to resolve this issue would need to have more targeted and efficient search approaches to improve its ability to identify this problem.

Figure 35: Trajectory for instance `sympy__sympy-21614`.

---

**Issue**                                                          **sympy__sympy-21614**

We're currently solving the following issue within our repository. Here's the issue text:
ISSUE:
Wrong Derivative kind attribute
I'm playing around with the 'kind' attribute.
The following is correct:

```
from sympy import Integral, Derivative
from sympy import MatrixSymbol
from sympy.abc import x
A = MatrixSymbol('A', 2, 2)
i = Integral(A, x)
i.kind
# MatrixKind(NumberKind)
```

This one is wrong:

```
d = Derivative(A, x)
d.kind
# UndefinedKind
```

INSTRUCTIONS: (31 lines)

```
(Open file: n/a)
(Current directory: /sympy__sympy)
bash-$
```

**SWE-agent**                                    **sympy__sympy-21614**

First, I'll create a new Python script to reproduce the issue as described. This script will attempt to create a 'Derivative' of a 'MatrixSymbol' and print out the 'kind' attribute to verify if it is indeed 'UndefinedKind'.

```
create reproduce_issue.py
```

---

**Observation**                                  **sympy__sympy-21614**

```
[File: /sympy__sympy/reproduce_issue.py (1 lines total)]
1:

(Open file: n/a)
(Current directory: /sympy__sympy)
bash-$
```

---

**SWE-agent**                                    **sympy__sympy-21614**

Now let's paste in the example code from the issue that demonstrates the incorrect behavior for the 'kind' attribute of a 'Derivative' of a 'MatrixSymbol'.

```
edit 1:1
from sympy import Integral, Derivative
from sympy import MatrixSymbol
from sympy.abc import x
A = MatrixSymbol('A', 2, 2)
d = Derivative(A, x)
print(d.kind)
# Expected: MatrixKind(NumberKind), but getting UndefinedKind
end_of_edit
```

---

**Observation**                                  **sympy__sympy-21614**

```
[File: /sympy__sympy/reproduce_issue.py (7 lines total)]
1:from sympy import Integral, Derivative
2:from sympy import MatrixSymbol
3:from sympy.abc import x
4:A = MatrixSymbol('A', 2, 2)
5:d = Derivative(A, x)
6:print(d.kind)
7:# Expected: MatrixKind(NumberKind), but getting UndefinedKind
File updated. Please review the changes and make sure they are
↪  correct (correct indentation, no duplicate lines, etc). Edit
↪  the file again if necessary.

(Open file: /sympy__sympy/reproduce_issue.py)
(Current directory: /sympy__sympy)
bash-$
```

---

**SWE-agent**                                    **sympy__sympy-21614**

The reproduction script has been created and updated with the code that demonstrates the issue. Now, let's run the script to confirm that the 'kind' attribute of the 'Derivative' is indeed 'UndefinedKind'.

---

```
python reproduce_issue.py
```

```
UndefinedKind

(Open file: /sympy__sympy/reproduce_issue.py)
(Current directory: /sympy__sympy)
bash-$
```

**SWE-agent**                                                         **sympy__sympy-21614**

The output confirms that the 'kind' attribute of the 'Derivative' is 'UndefinedKind', which is not the expected behavior. The next step is to locate the part of the code that handles the 'kind' attribute for the 'Derivative' class and make the necessary corrections.
Since the 'Derivative' class is likely to be defined in a file related to calculus or differentiation, I will search for the file that contains the 'Derivative' class definition.

```
find_file "derivative.py"
```

**Observation**                                                       **sympy__sympy-21614**

```
No matches found for "derivative.py" in /sympy__sympy

(Open file: /sympy__sympy/reproduce_issue.py)
(Current directory: /sympy__sympy)
bash-$
```

**SWE-agent**                                                         **sympy__sympy-21614**

It seems that the file name I searched for does not match the actual file name where the 'Derivative' class is defined. The 'Derivative' class might be defined in a file with a different name, possibly something more general related to symbolic calculus in SymPy.
Let's try a broader search for files that might contain the 'Derivative' class definition. Since the class name is 'Derivative', it's likely that the file name contains "calculus" or a similar keyword. I'll use the 'search_dir' command to look for the term "Derivative" within the Python files in the current directory.

```
search_dir "Derivative" .
```

**Observation**                                                       **sympy__sympy-21614**

```
More than 182 files matched for "Derivative" in /sympy__sympy.
↪  Please narrow your search.

(Open file: /sympy__sympy/reproduce_issue.py)
(Current directory: /sympy__sympy)
bash-$
```

**SWE-agent**                                                         **sympy__sympy-21614**

The search for "Derivative" returned too many results, so I need to narrow it down. Since the 'Derivative' class is a core part of SymPy, it's likely to be in a core module. A common convention is to have such

core classes in a file named after the module, so I will search for a file that might be named 'core.py' or similar.

---

```
find_file "core.py"
```

---

**Observation**                                    **sympy__sympy-21614**

```
Found 5 matches for "core.py" in /sympy__sympy:
/sympy__sympy/sympy/core/core.py
/sympy__sympy/sympy/multipledispatch/core.py
/sympy__sympy/sympy/unify/core.py
/sympy__sympy/sympy/strategies/branch/core.py
/sympy__sympy/sympy/strategies/core.py

(Open file: /sympy__sympy/reproduce_issue.py)
(Current directory: /sympy__sympy)
bash-$
```

---

**SWE-agent**                                       **sympy__sympy-21614**

The search has returned several 'core.py' files, but the most likely candidate for containing the 'Derivative' class definition is within the main 'sympy/core' directory. Let's open the 'core.py' file in the 'sympy/core' directory to investigate further.

---

```
open sympy/core/core.py
```

---

**Observation**                                     **sympy__sympy-21614**

```
[File: /sympy__sympy/sympy/core/core.py (104 lines total)]
1:""" The core's core. """
2:
3:# used for canonical ordering of symbolic sequences
4:# via __cmp__ method:
5:# FIXME this is *so* irrelevant and outdated!
6:ordering_of_classes = [
7:    # singleton numbers
8:    'Zero', 'One', 'Half', 'Infinity', 'NaN', 'NegativeOne',
↪   'NegativeInfinity',
9:    # numbers
10:    'Integer', 'Rational', 'Float',
11:    # singleton symbols
12:    'Exp1', 'Pi', 'ImaginaryUnit',
13:    # symbols
14:    'Symbol', 'Wild', 'Temporary',
15:    # arithmetic operations
16:    'Pow', 'Mul', 'Add',
17:    # function values
18:    'Derivative', 'Integral',
19:    # defined singleton functions
20:    'Abs', 'Sign', 'Sqrt',
21:    'Floor', 'Ceiling',
22:    'Re', 'Im', 'Arg',
23:    'Conjugate',
24:    'Exp', 'Log',
25:    'Sin', 'Cos', 'Tan', 'Cot', 'ASin', 'ACos', 'ATan', 'ACot',
26:    'Sinh', 'Cosh', 'Tanh', 'Coth', 'ASinh', 'ACosh', 'ATanh',
↪   'ACoth',
```

```
27:    'RisingFactorial', 'FallingFactorial',
28:    'factorial', 'binomial',
29:    'Gamma', 'LowerGamma', 'UpperGamma', 'PolyGamma',
30:    'Erf',
31:    # special polynomials
32:    'Chebyshev', 'Chebyshev2',
33:    # undefined functions
34:    'Function', 'WildFunction',
35:    # anonymous functions
36:    'Lambda',
37:    # Landau O symbol
38:    'Order',
39:    # relational operations
40:    'Equality', 'Unequality', 'StrictGreaterThan',
↪  'StrictLessThan',
41:    'GreaterThan', 'LessThan',
42:]
43:
44:
45:class Registry:
46:    """
47:    Base class for registry objects.
48:
49:    Registries map a name to an object using attribute notation.
↪  Registry
50:    classes behave singletonically: all their instances share
↪  the same state,
51:    which is stored in the class object.
52:
53:    All subclasses should set `__slots__ = ()`.
54:    """
55:    __slots__ = ()
56:
57:    def __setattr__(self, name, obj):
58:        setattr(self.__class__, name, obj)
59:
60:    def __delattr__(self, name):
61:        delattr(self.__class__, name)
62:
63:#A set containing all sympy class objects
64:all_classes = set()
65:
66:
67:class BasicMeta(type):
68:
69:    def __init__(cls, *args, **kws):
70:        all_classes.add(cls)
71:        cls.__sympy__ = property(lambda self: True)
72:
73:    def __cmp__(cls, other):
74:        # If the other object is not a Basic subclass, then we
↪  are not equal to
75:        # it.
76:        if not isinstance(other, BasicMeta):
77:            return -1
78:        n1 = cls.__name__
79:        n2 = other.__name__
80:        if n1 == n2:
81:            return 0
82:
83:        UNKNOWN = len(ordering_of_classes) + 1
84:        try:
85:            i1 = ordering_of_classes.index(n1)
```

```
86:        except ValueError:
87:            i1 = UNKNOWN
88:        try:
89:            i2 = ordering_of_classes.index(n2)
90:        except ValueError:
91:            i2 = UNKNOWN
92:        if i1 == UNKNOWN and i2 == UNKNOWN:
93:            return (n1 > n2) - (n1 < n2)
94:        return (i1 > i2) - (i1 < i2)
95:
96:    def __lt__(cls, other):
97:        if cls.__cmp__(other) == -1:
98:            return True
99:        return False
100:
(4 more lines below)

(Open file: /sympy__sympy/reproduce_issue.py)
(Current directory: /sympy__sympy)
bash-$
```

---

**SWE-agent**                                            **sympy__sympy-21614**

The 'core.py' file in the 'sympy/core' directory does not seem to contain the implementation details of the 'Derivative' class. It's more likely that the implementation is in another file within the 'sympy/core' directory. Since the 'Derivative' class is a fundamental part of symbolic calculus, it might be in a file named something like 'function.py' or 'operations.py'.

Let's try to find a file that specifically contains the implementation of the 'Derivative' class within the 'sympy/core' directory.

---

```
find_file "function.py" sympy/core
```

---

**Observation**                                         **sympy__sympy-21614**

```
Found 1 matches for "function.py" in /sympy__sympy/sympy/core:
/sympy__sympy/sympy/core/function.py

(Open file: /sympy__sympy/sympy/core/core.py)
(Current directory: /sympy__sympy)
bash-$
```

---

**SWE-agent**                                            **sympy__sympy-21614**

Great, we have found the 'function.py' file in the 'sympy/core' directory, which is likely to contain the implementation of the 'Derivative' class. Let's open this file to investigate and make the necessary changes.

---

```
open sympy/core/function.py
```

---

**Observation**                                         **sympy__sympy-21614**

```
[File: /sympy__sympy/sympy/core/function.py (3389 lines total)]
1:"""
2:There are three types of functions implemented in SymPy:
3:
```

```
 4:    1) defined functions (in the sense that they can be
↪  evaluated) like
 5:       exp or sin; they have a name and a body:
 6:           f = exp
 7:    2) undefined function which have a name but no body.
↪  Undefined
 8:       functions can be defined using a Function class as
↪  follows:
 9:           f = Function('f')
10:       (the result will be a Function instance)
11:    3) anonymous function (or lambda function) which have a body
↪  (defined
12:       with dummy variables) but have no name:
13:           f = Lambda(x, exp(x)*x)
14:           f = Lambda((x, y), exp(x)*y)
15:    The fourth type of functions are composites, like (sin +
↪  cos)(x); these work in
16:    SymPy core, but are not yet part of SymPy.
17:
18:    Examples
19:    ========
20:
21:    >>> import sympy
22:    >>> f = sympy.Function("f")
23:    >>> from sympy.abc import x
24:    >>> f(x)
25:    f(x)
26:    >>> print(sympy.srepr(f(x).func))
27:    Function('f')
28:    >>> f(x).args
29:    (x,)
30:
31:"""
32:
33:from typing import Any, Dict as tDict, Optional, Set as tSet,
↪  Tuple as tTuple, Union
34:
35:from .add import Add
36:from .assumptions import ManagedProperties
37:from .basic import Basic, _atomic
38:from .cache import cacheit
39:from .compatibility import iterable, is_sequence, as_int,
↪  ordered, Iterable
40:from .decorators import _sympifyit
41:from .expr import Expr, AtomicExpr
42:from .numbers import Rational, Float
43:from .operations import LatticeOp
44:from .rules import Transform
45:from .singleton import S
46:from .sympify import sympify
47:
48:from sympy.core.containers import Tuple, Dict
49:from sympy.core.parameters import global_parameters
50:from sympy.core.logic import fuzzy_and, fuzzy_or, fuzzy_not,
↪  FuzzyBool
51:from sympy.utilities import default_sort_key
52:from sympy.utilities.exceptions import SymPyDeprecationWarning
53:from sympy.utilities.iterables import has_dups, sift
54:from sympy.utilities.misc import filldedent
55:
56:import mpmath
57:import mpmath.libmp as mlib
58:
```

```
59:import inspect
60:from collections import Counter
61:
62:def _coeff_isneg(a):
63:    """Return True if the leading Number is negative.
64:
65:    Examples
66:    ========
67:
68:    >>> from sympy.core.function import _coeff_isneg
69:    >>> from sympy import S, Symbol, oo, pi
70:    >>> _coeff_isneg(-3*pi)
71:    True
72:    >>> _coeff_isneg(S(3))
73:    False
74:    >>> _coeff_isneg(-oo)
75:    True
76:    >>> _coeff_isneg(Symbol('n', negative=True)) # coeff is 1
77:    False
78:
79:    For matrix expressions:
80:
81:    >>> from sympy import MatrixSymbol, sqrt
82:    >>> A = MatrixSymbol("A", 3, 3)
83:    >>> _coeff_isneg(-sqrt(2)*A)
84:    True
85:    >>> _coeff_isneg(sqrt(2)*A)
86:    False
87:    """
88:
89:    if a.is_MatMul:
90:        a = a.args[0]
91:    if a.is_Mul:
92:        a = a.args[0]
93:    return a.is_Number and a.is_extended_negative
94:
95:
96:class PoleError(Exception):
97:    pass
98:
99:
100:class ArgumentIndexError(ValueError):
(3289 more lines below)

(Open file: /sympy__sympy/sympy/core/core.py)
(Current directory: /sympy__sympy)
bash-$
```

## Observation                                              sympy__sympy-21614

```
Found 1 matches for "class Derivative" in
↪  /sympy__sympy/sympy/core/function.py:
Line 1053:class Derivative(Expr):
End of matches for "class Derivative" in
↪  /sympy__sympy/sympy/core/function.py

(Open file: /sympy__sympy/sympy/core/function.py)
(Current directory: /sympy__sympy)
bash-$
```

## SWE-agent                                                sympy__sympy-21614

We have found that the 'Derivative' class is defined on line 1053 of 'function.py'. Let's navigate to that line to inspect the implementation and determine how to correct the 'kind' attribute issue.

---

```
goto 1053
```

## Observation                                              sympy__sympy-21614

```
[File: /sympy__sympy/sympy/core/function.py (3389 lines total)]
(1037 more lines above)
1038:                   nargs = (as_int(nargs),)
1039:               nargs = FiniteSet(*nargs)
1040:           cls.nargs = nargs
1041:
1042:    def matches(self, expr, repl_dict={}, old=False):
1043:           if not isinstance(expr, (AppliedUndef, Function)):
1044:               return None
1045:           if len(expr.args) not in self.nargs:
1046:               return None
1047:
1048:           repl_dict = repl_dict.copy()
1049:           repl_dict[self] = expr
1050:           return repl_dict
1051:
1052:
1053:class Derivative(Expr):
1054:     """
1055:     Carries out differentiation of the given expression with
↪  respect to symbols.
1056:
1057:     Examples
1058:     ========
1059:
1060:     >>> from sympy import Derivative, Function, symbols, Subs
1061:     >>> from sympy.abc import x, y
1062:     >>> f, g = symbols('f g', cls=Function)
1063:
1064:     >>> Derivative(x**2, x, evaluate=True)
1065:     2*x
1066:
1067:     Denesting of derivatives retains the ordering of
↪  variables:
1068:
1069:         >>> Derivative(Derivative(f(x, y), y), x)
1070:         Derivative(f(x, y), y, x)
1071:
```

```
1072:    Contiguously identical symbols are merged into a tuple
↪  giving
1073:    the symbol and the count:
1074:
1075:        >>> Derivative(f(x), x, x, y, x)
1076:        Derivative(f(x), (x, 2), y, x)
1077:
1078:    If the derivative cannot be performed, and evaluate is
↪  True, the
1079:    order of the variables of differentiation will be made
↪  canonical:
1080:
1081:        >>> Derivative(f(x, y), y, x, evaluate=True)
1082:        Derivative(f(x, y), x, y)
1083:
1084:    Derivatives with respect to undefined functions can be
↪  calculated:
1085:
1086:        >>> Derivative(f(x)**2, f(x), evaluate=True)
1087:        2*f(x)
1088:
1089:    Such derivatives will show up when the chain rule is used
↪  to
1090:    evalulate a derivative:
1091:
1092:        >>> f(g(x)).diff(x)
1093:        Derivative(f(g(x)), g(x))*Derivative(g(x), x)
1094:
1095:    Substitution is used to represent derivatives of functions
↪  with
1096:    arguments that are not symbols or functions:
1097:
1098:        >>> f(2*x + 3).diff(x) == 2*Subs(f(y).diff(y), y, 2*x +
↪  3)
1099:        True
1100:
1101:    Notes
1102:    =====
1103:
1104:    Simplification of high-order derivatives:
1105:
1106:    Because there can be a significant amount of
↪  simplification that can be
1107:    done when multiple differentiations are performed, results
↪  will be
1108:    automatically simplified in a fairly conservative fashion
↪  unless the
1109:    keyword ``simplify`` is set to False.
1110:
1111:        >>> from sympy import sqrt, diff, Function, symbols
1112:        >>> from sympy.abc import x, y, z
1113:        >>> f, g = symbols('f,g', cls=Function)
1114:
1115:        >>> e = sqrt((x + 1)**2 + x)
1116:        >>> diff(e, (x, 5), simplify=False).count_ops()
1117:        136
1118:        >>> diff(e, (x, 5)).count_ops()
1119:        30
1120:
1121:    Ordering of variables:
1122:
1123:    If evaluate is set to True and the expression cannot be
↪  evaluated, the
```

```
1124:    list of differentiation symbols will be sorted, that is,
↪  the expression is
1125:    assumed to have continuous derivatives up to the order
↪  asked.
1126:
1127:    Derivative wrt non-Symbols:
1128:
1129:    For the most part, one may not differentiate wrt
↪  non-symbols.
1130:    For example, we do not allow differentiation wrt `x*y`
↪  because
1131:    there are multiple ways of structurally defining where x*y
↪  appears
1132:    in an expression: a very strict definition would make
1133:    (x*y*z).diff(x*y) == 0. Derivatives wrt defined functions
↪  (like
1134:    cos(x)) are not allowed, either:
1135:
1136:        >>> (x*y*z).diff(x*y)
1137:        Traceback (most recent call last):
(2252 more lines below)

(Open file: /sympy__sympy/sympy/core/function.py)
(Current directory: /sympy__sympy)
bash-$
```

---

**SWE-agent**                                        sympy__sympy-21614

We are now at the beginning of the 'Derivative' class definition. To address the issue with the 'kind'
attribute, we need to find where the 'kind' property is defined or handled within this class. Since the
file is large, I will scroll down to find the relevant section where the 'kind' property is likely to be
implemented.

─────────────────────────────────────────────────────────────────

```
scroll_down
```

---

**Observation**                                      sympy__sympy-21614

```
[File: /sympy__sympy/sympy/core/function.py (3389 lines total)]
(1135 more lines above)
1136:        >>> (x*y*z).diff(x*y)
1137:        Traceback (most recent call last):
1138:        ...
1139:        ValueError: Can't calculate derivative wrt x*y.
1140:
1141:    To make it easier to work with variational calculus,
↪  however,
1142:    derivatives wrt AppliedUndef and Derivatives are allowed.
1143:    For example, in the Euler-Lagrange method one may write
1144:    F(t, u, v) where u = f(t) and v = f'(t). These variables
↪  can be
1145:    written explicitly as functions of time::
1146:
1147:        >>> from sympy.abc import t
1148:        >>> F = Function('F')
1149:        >>> U = f(t)
1150:        >>> V = U.diff(t)
1151:
1152:    The derivative wrt f(t) can be obtained directly:
1153:
```

```
1154:        >>> direct = F(t, U, V).diff(U)
1155:
1156:    When differentiation wrt a non-Symbol is attempted, the
↪  non-Symbol
1157:    is temporarily converted to a Symbol while the
↪  differentiation
1158:    is performed and the same answer is obtained:
1159:
1160:        >>> indirect = F(t, U, V).subs(U, x).diff(x).subs(x,
↪  U)
1161:        >>> assert direct == indirect
1162:
1163:    The implication of this non-symbol replacement is that all
1164:    functions are treated as independent of other functions
↪  and the
1165:    symbols are independent of the functions that contain
↪  them::
1166:
1167:        >>> x.diff(f(x))
1168:        0
1169:        >>> g(x).diff(f(x))
1170:        0
1171:
1172:    It also means that derivatives are assumed to depend only
1173:    on the variables of differentiation, not on anything
↪  contained
1174:    within the expression being differentiated::
1175:
1176:        >>> F = f(x)
1177:        >>> Fx = F.diff(x)
1178:        >>> Fx.diff(F)  # derivative depends on x, not F
1179:        0
1180:        >>> Fxx = Fx.diff(x)
1181:        >>> Fxx.diff(Fx)  # derivative depends on x, not Fx
1182:        0
1183:
1184:    The last example can be made explicit by showing the
↪  replacement
1185:    of Fx in Fxx with y:
1186:
1187:        >>> Fxx.subs(Fx, y)
1188:        Derivative(y, x)
1189:
1190:        Since that in itself will evaluate to zero,
↪  differentiating
1191:        wrt Fx will also be zero:
1192:
1193:        >>> _.doit()
1194:        0
1195:
1196:    Replacing undefined functions with concrete expressions
1197:
1198:    One must be careful to replace undefined functions with
↪  expressions
1199:    that contain variables consistent with the function
↪  definition and
1200:    the variables of differentiation or else insconsistent
↪  result will
1201:    be obtained. Consider the following example:
1202:
1203:    >>> eq = f(x)*g(y)
1204:    >>> eq.subs(f(x), x*y).diff(x, y).doit()
1205:    y*Derivative(g(y), y) + g(y)
```

```
1206:    >>> eq.diff(x, y).subs(f(x), x*y).doit()
1207:    y*Derivative(g(y), y)
1208:
1209:    The results differ because `f(x)` was replaced with an
↪  expression
1210:    that involved both variables of differentiation. In the
↪  abstract
1211:    case, differentiation of `f(x)` by `y` is 0; in the
↪  concrete case,
1212:    the presence of `y` made that derivative nonvanishing and
↪  produced
1213:    the extra `g(y)` term.
1214:
1215:    Defining differentiation for an object
1216:
1217:    An object must define ._eval_derivative(symbol) method
↪  that returns
1218:    the differentiation result. This function only needs to
↪  consider the
1219:    non-trivial case where expr contains symbol and it should
↪  call the diff()
1220:    method internally (not _eval_derivative); Derivative
↪  should be the only
1221:    one to call _eval_derivative.
1222:
1223:    Any class can allow derivatives to be taken with respect
↪  to
1224:    itself (while indicating its scalar nature). See the
1225:    docstring of Expr._diff_wrt.
1226:
1227:    See Also
1228:    ========
1229:    _sort_variable_count
1230:    """
1231:
1232:    is_Derivative = True
1233:
1234:    @property
1235:    def _diff_wrt(self):
(2154 more lines below)

(Open file: /sympy__sympy/sympy/core/function.py)
(Current directory: /sympy__sympy)
bash-$
```

---

**SWE-agent**                                    **sympy__sympy-21614**

We have scrolled down through the 'function.py' file, but we have not yet found the implementation of
the 'kind' property for the 'Derivative' class. We need to continue scrolling to find the relevant section.

---

```
scroll_down
```

---

**Observation**                                 **sympy__sympy-21614**

```
[File: /sympy__sympy/sympy/core/function.py (3389 lines total)]
(1233 more lines above)
[File contents omitted for brevity]
(2056 more lines below)
```

```
(Open file: /sympy__sympy/sympy/core/function.py)
(Current directory: /sympy__sympy)
bash-$
```

---

**SWE-agent**                                          **sympy__sympy-21614**

We have scrolled further down the 'function.py' file, but we still haven't found the implementation of
the 'kind' property for the 'Derivative' class. We need to continue scrolling to find the relevant section
where the 'kind' property is likely to be implemented or overridden.

___________________________________________________________________

```
scroll_down
```

---

**Observation**                                        **sympy__sympy-21614**

```
[File: /sympy__sympy/sympy/core/function.py (3389 lines total)]
(1331 more lines above)
[File contents omitted for brevity]
(1958 more lines below)

(Open file: /sympy__sympy/sympy/core/function.py)
(Current directory: /sympy__sympy)
bash-$
```

---

**SWE-agent**                                          **sympy__sympy-21614**

We have scrolled further down the 'function.py' file, but we still haven't found the implementation of
the 'kind' property for the 'Derivative' class. We need to continue scrolling to find the relevant section
where the 'kind' property is likely to be implemented or overridden.

___________________________________________________________________

```
scroll_down
```

---

**Observation**                                        **sympy__sympy-21614**

```
[File: /sympy__sympy/sympy/core/function.py (3389 lines total)]
(1429 more lines above)
[File contents omitted for brevity]
(1860 more lines below)

(Open file: /sympy__sympy/sympy/core/function.py)
(Current directory: /sympy__sympy)
bash-$
```

---

**SWE-agent**                                          **sympy__sympy-21614**

We have scrolled further down the 'function.py' file, but we still haven't found the implementation of
the 'kind' property for the 'Derivative' class. We need to continue scrolling to find the relevant section
where the 'kind' property is likely to be implemented or overridden.

___________________________________________________________________

```
scroll_down
```

```
[File: /sympy__sympy/sympy/core/function.py (3389 lines total)]
(1527 more lines above)
[File contents omitted for brevity]
(1762 more lines below)

(Open file: /sympy__sympy/sympy/core/function.py)
(Current directory: /sympy__sympy)
bash-$
```

**SWE-agent**                                                    **sympy__sympy-21614**

We have scrolled further down the 'function.py' file, but we still haven't found the implementation of the 'kind' property for the 'Derivative' class. We need to continue scrolling to find the relevant section where the 'kind' property is likely to be implemented or overridden.

---

```
scroll_down
```

```
[File: /sympy__sympy/sympy/core/function.py (3389 lines total)]
(1625 more lines above)
[File contents omitted for brevity]
(1664 more lines below)

(Open file: /sympy__sympy/sympy/core/function.py)
(Current directory: /sympy__sympy)
bash-$
```

**SWE-agent**                                                    **sympy__sympy-21614**

We have scrolled further down the 'function.py' file, but we still haven't found the implementation of the 'kind' property for the 'Derivative' class. We need to continue scrolling to find the relevant section where the 'kind' property is likely to be implemented or overridden.

---

```
scroll_down
```

```
[File: /sympy__sympy/sympy/core/function.py (3389 lines total)]
(1723 more lines above)
[File contents omitted for brevity]
(1566 more lines below)

(Open file: /sympy__sympy/sympy/core/function.py)
(Current directory: /sympy__sympy)
bash-$
```

**SWE-agent**                                                    **sympy__sympy-21614**

We have scrolled further down the 'function.py' file, but we still haven't found the implementation of the 'kind' property for the 'Derivative' class. We need to continue scrolling to find the relevant section where the 'kind' property is likely to be implemented or overridden.

```
scroll_down
```

**SWE-agent**                                                      **sympy__sympy-21614**

We have scrolled further down the 'function.py' file, but we still haven't found the implementation of the 'kind' property for the 'Derivative' class. We need to continue scrolling to find the relevant section where the 'kind' property is likely to be implemented or overridden.

```
scroll_down
```

**Observation**                                                    **sympy__sympy-21614**

```
[File: /sympy__sympy/sympy/core/function.py (3389 lines total)]
(1919 more lines above)
[File contents omitted for brevity]
(1370 more lines below)

(Open file: /sympy__sympy/sympy/core/function.py)
(Current directory: /sympy__sympy)
bash-$
```

**SWE-agent**                                                      **sympy__sympy-21614**

We have scrolled further down the 'function.py' file, but we still haven't found the implementation of the 'kind' property for the 'Derivative' class. We need to continue scrolling to find the relevant section where the 'kind' property is likely to be implemented or overridden.

```
scroll_down
```

**Observation**                                                    **sympy__sympy-21614**

```
[File: /sympy__sympy/sympy/core/function.py (3389 lines total)]
(2017 more lines above)
[File contents omitted for brevity]
(1272 more lines below)

(Open file: /sympy__sympy/sympy/core/function.py)
(Current directory: /sympy__sympy)
bash-$
```

We have scrolled further down the 'function.py' file, but we still haven't found the implementation of the 'kind' property for the 'Derivative' class. We need to continue scrolling to find the relevant section where the 'kind' property is likely to be implemented or overridden.

```
scroll_down
```

Observation                                                sympy__sympy-21614

```
[File: /sympy__sympy/sympy/core/function.py (3389 lines total)]
(2115 more lines above)
[File contents omitted for brevity]
(1174 more lines below)

(Open file: /sympy__sympy/sympy/core/function.py)
(Current directory: /sympy__sympy)
bash-$
```

SWE-agent                                                  sympy__sympy-21614

We have scrolled further down the 'function.py' file, but we still haven't found the implementation of the 'kind' property for the 'Derivative' class. We need to continue scrolling to find the relevant section where the 'kind' property is likely to be implemented or overridden.

```
scroll_down
```

Observation                                                sympy__sympy-21614

```
[File: /sympy__sympy/sympy/core/function.py (3389 lines total)]
(2213 more lines above)
[File contents omitted for brevity]
(1076 more lines below)

(Open file: /sympy__sympy/sympy/core/function.py)
(Current directory: /sympy__sympy)
bash-$
```

SWE-agent                                                  sympy__sympy-21614

We have scrolled further down the 'function.py' file, but we still haven't found the implementation of the 'kind' property for the 'Derivative' class. We need to continue scrolling to find the relevant section where the 'kind' property is likely to be implemented or overridden.

```
scroll_down
```

Observation                                                sympy__sympy-21614

```
[File: /sympy__sympy/sympy/core/function.py (3389 lines total)]
(2311 more lines above)
[File contents omitted for brevity]
(978 more lines below)
```

```
(Open file: /sympy__sympy/sympy/core/function.py)
(Current directory: /sympy__sympy)
bash-$
```

### SWE-agent                                    sympy__sympy-21614

We have scrolled further down the 'function.py' file, but we still haven't found the implementation of the 'kind' property for the 'Derivative' class. We need to continue scrolling to find the relevant section where the 'kind' property is likely to be implemented or overridden.

―――――――――――――――――――――――――――――――――――――――――――――

```
scroll_down
```

### Observation                                  sympy__sympy-21614

```
[File: /sympy__sympy/sympy/core/function.py (3389 lines total)]
(2409 more lines above)
[File contents omitted for brevity]
(880 more lines below)

(Open file: /sympy__sympy/sympy/core/function.py)
(Current directory: /sympy__sympy)
bash-$
```

### SWE-agent                                    sympy__sympy-21614

We have scrolled further down the 'function.py' file, and we have now found the 'diff' function, which is related to differentiation. However, we still need to find the implementation of the 'kind' property for the 'Derivative' class. Since we are in the section related to differentiation, we should be close to the relevant code.
Let's continue scrolling to find the 'Derivative' class implementation.

―――――――――――――――――――――――――――――――――――――――――――――

```
scroll_down
```

### Observation                                  sympy__sympy-21614

```
[File: /sympy__sympy/sympy/core/function.py (3389 lines total)]
(2507 more lines above)
[File contents omitted for brevity]
(782 more lines below)

(Open file: /sympy__sympy/sympy/core/function.py)
(Current directory: /sympy__sympy)
bash-$
```

### SWE-agent                                    sympy__sympy-21614

We have scrolled further down the 'function.py' file, and we have now found the 'diff' function, which is related to differentiation. However, we still need to find the implementation of the 'kind' property for the 'Derivative' class. Since we are in the section related to differentiation, we should be close to the relevant code.
Let's continue scrolling to find the 'Derivative' class implementation.

―――――――――――――――――――――――――――――――――――――――――――――

```
scroll_down
```

**Observation**                  **sympy__sympy-21614**

```
[File: /sympy__sympy/sympy/core/function.py (3389 lines total)]
(2605 more lines above)
[File contents omitted for brevity]
(684 more lines below)

(Open file: /sympy__sympy/sympy/core/function.py)
(Current directory: /sympy__sympy)
bash-$
```

**SWE-agent**                  **sympy__sympy-21614**

We have scrolled further down the 'function.py' file, and we have now found the 'diff' function, which is related to differentiation. However, we still need to find the implementation of the 'kind' property for the 'Derivative' class. Since we are in the section related to differentiation, we should be close to the relevant code.
Let's continue scrolling to find the 'Derivative' class implementation.

―――――――――――――――――――――――――――――――――――――

```
scroll_down
```

**Observation**                  **sympy__sympy-21614**

```
[File: /sympy__sympy/sympy/core/function.py (3389 lines total)]
(2703 more lines above)
[File contents omitted for brevity]
(586 more lines below)

(Open file: /sympy__sympy/sympy/core/function.py)
(Current directory: /sympy__sympy)
bash-$
```

**SWE-agent**                  **sympy__sympy-21614**

We have scrolled further down the 'function.py' file, and we have now found the 'expand' function, which is related to expression expansion. However, we still need to find the implementation of the 'kind' property for the 'Derivative' class. Since we are in the section related to expression manipulation, we should be close to the relevant code.
Let's continue scrolling to find the 'Derivative' class implementation.

―――――――――――――――――――――――――――――――――――――

```
scroll_down
```

**Observation**                  **sympy__sympy-21614**

```
[File: /sympy__sympy/sympy/core/function.py (3389 lines total)]
(2801 more lines above)
[File contents omitted for brevity]
(488 more lines below)

(Open file: /sympy__sympy/sympy/core/function.py)
(Current directory: /sympy__sympy)
bash-$
```

**SWE-agent**          **sympy__sympy-21614**

We have scrolled further down the 'function.py' file, and we have now found various expansion functions and wrappers, but we still need to find the implementation of the 'kind' property for the 'Derivative' class. Since we are in the section related to expression manipulation, we should be close to the relevant code.

Let's continue scrolling to find the 'Derivative' class implementation.

---

```
scroll_down
```

---

**Observation**          **sympy__sympy-21614**

```
[File: /sympy__sympy/sympy/core/function.py (3389 lines total)]
(2899 more lines above)
[File contents omitted for brevity]
(390 more lines below)

(Open file: /sympy__sympy/sympy/core/function.py)
(Current directory: /sympy__sympy)
bash-$
```

---

**SWE-agent**          **sympy__sympy-21614**

We have scrolled further down the 'function.py' file, and we have now found various expansion functions and wrappers, but we still need to find the implementation of the 'kind' property for the 'Derivative' class. Since we are in the section related to expression manipulation, we should be close to the relevant code.

Let's continue scrolling to find the 'Derivative' class implementation.

---

```
scroll_down
```

---

**Observation**          **sympy__sympy-21614**

```
[File: /sympy__sympy/sympy/core/function.py (3389 lines total)]
(2997 more lines above)
[File contents omitted for brevity]
(292 more lines below)

(Open file: /sympy__sympy/sympy/core/function.py)
(Current directory: /sympy__sympy)
bash-$
```

---

**SWE-agent**          **sympy__sympy-21614**

We have scrolled further down the 'function.py' file, and we have now found various expansion functions and wrappers, but we still need to find the implementation of the 'kind' property for the 'Derivative' class. Since we are in the section related to expression manipulation, we should be close to the relevant code.

Let's continue scrolling to find the 'Derivative' class implementation.

---

```
scroll_down
```

**SWE-agent**           sympy__sympy-21614

Exit due to cost limit

_______________________________________________________

```
exit_cost
```

**Observation**           sympy__sympy-21614

```
Exited
```

**Gold Patch**           sympy__sympy-21614

```
diff --git a/sympy/core/function.py b/sympy/core/function.py
--- a/sympy/core/function.py
+++ b/sympy/core/function.py
@@ -1707,6 +1707,10 @@ def free_symbols(self):
             ret.update(count.free_symbols)
         return ret

+    @property
+    def kind(self):
+        return self.args[0].kind
+
     def _eval_subs(self, old, new):
         # The substitution (old, new) cannot be done inside
         # Derivative(expr, vars) for a variety of reasons
```

**Analysis for django/django-14411**

**Issue Summary.** The issue involves the `ReadOnlyPasswordHashWidget` in the Django admin interface. The label element for this widget has a 'for' attribute that points to a non-labelable element. Since the `ReadOnlyPasswordHashWidget` renders text rather than an input element, the label should not have a 'for' attribute.

**Interaction Summary**

1. **Locate the Relevant File**: The agent began by searching for the `ReadOnlyPasswordHashWidget` definition using the `find_file` command, but found no results. It then used the `search_dir` command to find references to the widget within the repository and located relevant matches in several files. The agent chose to first examine the `helpers.py` file, but found it only contained a reference to the widget and did not include the label rendering code.

2. **Explore Related Files**: Next, the agent opened the `forms.py` file in the `django/contrib/auth` directory, which defined the `ReadOnlyPasswordHashWidget` and its associated

`ReadOnlyPasswordHashField`. It identified that the widget used a template
(`read_only_password_hash.html`) for rendering, but the template did not include
relevant information relating to the Widget's `for` attribute.

3. **Edit the Class Definition**: Returning to `forms.py`, the agent found the
   `UserChangeForm` which used the `ReadOnlyPasswordHashField`. It reasoned
   that Django's form rendering system likely added the 'for' attribute automatically.
   To address the issue, the agent attempted to write a `label_tag` method for the
   `ReadOnlyPasswordHashField` to exclude the 'for' attribute. The agent made several
   edits to this effect, importing the necessary utilities from `django.utils.html` and
   adding a custom `label_tag` method.

4. **Submit Without Testing**: After the desired changes were made, the agent proceeds to
   submit without testing as it assumes that a Django server cannot be run and tested in this
   environment.

**Evaluation** The agent correctly identified the need to customize the label rendering for the
`ReadOnlyPasswordHashField` class. However, its solution failed to solve the issue since
it did not understand how labels were defined for a `Widget` object. In actuality, the model needed
to override the `id_for_label` property defined in the `Widget` class to change this behavior
properly. Similar functionality can be seen in other parts of the codebase.

Figure 36: Trajectory for instance `django__django-14411`.

---

**Issue**                                  **django__django-14411**

We're currently solving the following issue within our repository. Here's the issue text:
ISSUE:
Label for ReadOnlyPasswordHashWidget points to non-labelable element.
Description
(last modified by David Sanders)
In the admin, the label element for the ReadOnlyPasswordHashWidget widget has a 'for' attribute
which points to a non-labelable element, since the widget just renders text, not an input. There's no
labelable element for the widget, so the label shouldn't have a 'for' attribute.
INSTRUCTIONS: (31 lines)

```
(Open file: n/a)
(Current directory: /django__django)
bash-$
```

---

**SWE-agent**                                 **django__django-14411**

To address the issue, we need to locate the file that contains the 'ReadOnlyPasswordHashWidget'
widget's rendering code. Since this is likely part of the Django admin, we should start by searching for
the file that defines 'ReadOnlyPasswordHashWidget'. Let's use the 'find_file' command to locate this
file within the repository.

```
find_file "ReadOnlyPasswordHashWidget"
```

---

**Observation**                                  **django__django-14411**

```
No matches found for "ReadOnlyPasswordHashWidget" in
↪  /django__django

(Open file: n/a)
(Current directory: /django__django)
bash-$
```

**SWE-agent**                                        **django__django-14411**

It seems that the 'find_file' command did not yield any results for "ReadOnlyPasswordHashWidget".
This could be because the file name does not exactly match the widget name. Since the widget is likely
to be part of the Django admin, it might be located in a file related to forms or widgets in the admin app.
Let's try a broader search using the 'search_dir' command to look for the term "ReadOnlyPassword-
HashWidget" within the files of the repository. This should help us find the file where the widget is
defined or used.

```
search_dir "ReadOnlyPasswordHashWidget"
```

---

**Observation**                                          **django__django-14411**

```
Found 9 matches for "ReadOnlyPasswordHashWidget" in
↪  /django__django:
/django__django/django/contrib/admin/helpers.py (1 matches)
/django__django/django/contrib/auth/forms.py (2 matches)
/django__django/docs/releases/1.11.1.txt (1 matches)
/django__django/tests/auth_tests/test_forms.py (4 matches)
/django__django/tests/auth_tests/test_views.py (1 matches)
End of matches for "ReadOnlyPasswordHashWidget" in /django__django

(Open file: n/a)
(Current directory: /django__django)
bash-$
```

---

**SWE-agent**                                          **django__django-14411**

The search results indicate that there are matches for "ReadOnlyPasswordHashWidget" in several files,
but the most likely places where the rendering code would be defined are in 'helpers.py' and 'forms.py'
within the 'django/contrib/admin' and 'django/contrib/auth' directories, respectively.
Since the issue is related to the admin interface, let's first check the 'helpers.py' file in the 'django/con-
trib/admin' directory. We'll open this file and look for the relevant code.

```
open django/contrib/admin/helpers.py
```

---

**Observation**                                          **django__django-14411**

```
[File: /django__django/django/contrib/admin/helpers.py (424 lines
↪  total)]
1:import json
2:
3:from django import forms
4:from django.contrib.admin.utils import (
5:    display_for_field, flatten_fieldsets, help_text_for_field,
↪  label_for_field,
6:    lookup_field, quote,
7:)
8:from django.core.exceptions import ObjectDoesNotExist
9:from django.db.models.fields.related import (
10:    ForeignObjectRel, ManyToManyRel, OneToOneField,
11:)
12:from django.forms.utils import flatatt
13:from django.template.defaultfilters import capfirst,
↪  linebreaksbr
14:from django.urls import NoReverseMatch, reverse
15:from django.utils.html import conditional_escape, format_html
```

```
16:from django.utils.safestring import mark_safe
17:from django.utils.translation import gettext, gettext_lazy as _
18:
19:ACTION_CHECKBOX_NAME = '_selected_action'
20:
21:
22:class ActionForm(forms.Form):
23:    action = forms.ChoiceField(label=_('Action:'))
24:    select_across = forms.BooleanField(
25:        label='',
26:        required=False,
27:        initial=0,
28:        widget=forms.HiddenInput({'class': 'select-across'}),
29:    )
30:
31:
32:checkbox = forms.CheckboxInput({'class': 'action-select'},
→  lambda value: False)
33:
34:
35:class AdminForm:
36:    def __init__(self, form, fieldsets, prepopulated_fields,
→  readonly_fields=None, model_admin=None):
37:        self.form, self.fieldsets = form, fieldsets
38:        self.prepopulated_fields = [{
39:            'field': form[field_name],
40:            'dependencies': [form[f] for f in dependencies]
41:        } for field_name, dependencies in
→  prepopulated_fields.items()]
42:        self.model_admin = model_admin
43:        if readonly_fields is None:
44:            readonly_fields = ()
45:        self.readonly_fields = readonly_fields
46:
47:    def __iter__(self):
48:        for name, options in self.fieldsets:
49:            yield Fieldset(
50:                self.form, name,
51:                readonly_fields=self.readonly_fields,
52:                model_admin=self.model_admin,
53:                **options
54:            )
55:
56:    @property
57:    def errors(self):
58:        return self.form.errors
59:
60:    @property
61:    def non_field_errors(self):
62:        return self.form.non_field_errors
63:
64:    @property
65:    def media(self):
66:        media = self.form.media
67:        for fs in self:
68:            media = media + fs.media
69:        return media
70:
71:
72:class Fieldset:
73:    def __init__(self, form, name=None, readonly_fields=(),
→  fields=(), classes=(),
74:                    description=None, model_admin=None):
```

```
75:        self.form = form
76:        self.name, self.fields = name, fields
77:        self.classes = ' '.join(classes)
78:        self.description = description
79:        self.model_admin = model_admin
80:        self.readonly_fields = readonly_fields
81:
82:    @property
83:    def media(self):
84:        if 'collapse' in self.classes:
85:            return forms.Media(js=['admin/js/collapse.js'])
86:        return forms.Media()
87:
88:    def __iter__(self):
89:        for field in self.fields:
90:            yield Fieldline(self.form, field,
↪  self.readonly_fields, model_admin=self.model_admin)
91:
92:
93:class Fieldline:
94:    def __init__(self, form, field, readonly_fields=None,
↪  model_admin=None):
95:        self.form = form  # A django.forms.Form instance
96:        if not hasattr(field, "__iter__") or isinstance(field,
↪  str):
97:            self.fields = [field]
98:        else:
99:            self.fields = field
100:        self.has_visible_field = not all(
(324 more lines below)

(Open file: n/a)
(Current directory: /django__django)
bash-$
```

```
search_file "ReadOnlyPasswordHashWidget"
```

```
Found 1 matches for "ReadOnlyPasswordHashWidget" in
↪  /django__django/django/contrib/admin/helpers.py:
Line 215:                # ReadOnlyPasswordHashWidget.
End of matches for "ReadOnlyPasswordHashWidget" in
↪  /django__django/django/contrib/admin/helpers.py

(Open file: /django__django/django/contrib/admin/helpers.py)
(Current directory: /django__django)
bash-$
```

We have found a reference to 'ReadOnlyPasswordHashWidget' on line 215 of the 'helpers.py' file. Let's navigate to that line to examine the context and see how the label is being rendered.

```
goto 215
```

---

```
[File: /django__django/django/contrib/admin/helpers.py (424 lines
↪  total)]
(199 more lines above)
200:            return format_html('<a href="{}">{}</a>', url,
↪  remote_obj)
201:        except NoReverseMatch:
202:            return str(remote_obj)
203:
204:    def contents(self):
205:        from django.contrib.admin.templatetags.admin_list
↪  import _boolean_icon
206:        field, obj, model_admin = self.field['field'],
↪  self.form.instance, self.model_admin
207:        try:
208:            f, attr, value = lookup_field(field, obj,
↪  model_admin)
209:        except (AttributeError, ValueError,
↪  ObjectDoesNotExist):
210:            result_repr = self.empty_value_display
211:        else:
212:            if field in self.form.fields:
213:                widget = self.form[field].field.widget
214:                # This isn't elegant but suffices for
↪  contrib.auth's
215:                # ReadOnlyPasswordHashWidget.
216:                if getattr(widget, 'read_only', False):
217:                    return widget.render(field, value)
218:            if f is None:
219:                if getattr(attr, 'boolean', False):
220:                    result_repr = _boolean_icon(value)
221:                else:
222:                    if hasattr(value, "__html__"):
223:                        result_repr = value
224:                    else:
225:                        result_repr = linebreaksbr(value)
226:            else:
227:                if isinstance(f.remote_field, ManyToManyRel)
↪  and value is not None:
228:                    result_repr = ", ".join(map(str,
↪  value.all()))
229:                elif (
230:                    isinstance(f.remote_field,
↪  (ForeignObjectRel, OneToOneField)) and
231:                    value is not None
232:                ):
233:                    result_repr =
↪  self.get_admin_url(f.remote_field, value)
234:                else:
235:                    result_repr = display_for_field(value, f,
↪  self.empty_value_display)
236:                result_repr = linebreaksbr(result_repr)
```

```
237:            return conditional_escape(result_repr)
238:
239:
240:class InlineAdminFormSet:
241:    """
242:    A wrapper around an inline formset for use in the admin
↪ system.
243:    """
244:    def __init__(self, inline, formset, fieldsets,
↪ prepopulated_fields=None,
245:                 readonly_fields=None, model_admin=None,
↪ has_add_permission=True,
246:                 has_change_permission=True,
↪ has_delete_permission=True,
247:                 has_view_permission=True):
248:        self.opts = inline
249:        self.formset = formset
250:        self.fieldsets = fieldsets
251:        self.model_admin = model_admin
252:        if readonly_fields is None:
253:            readonly_fields = ()
254:        self.readonly_fields = readonly_fields
255:        if prepopulated_fields is None:
256:            prepopulated_fields = {}
257:        self.prepopulated_fields = prepopulated_fields
258:        self.classes = ' '.join(inline.classes) if
↪ inline.classes else ''
259:        self.has_add_permission = has_add_permission
260:        self.has_change_permission = has_change_permission
261:        self.has_delete_permission = has_delete_permission
262:        self.has_view_permission = has_view_permission
263:
264:    def __iter__(self):
265:        if self.has_change_permission:
266:            readonly_fields_for_editing = self.readonly_fields
267:        else:
268:            readonly_fields_for_editing = self.readonly_fields +
↪ flatten_fieldsets(self.fieldsets)
269:
270:        for form, original in zip(self.formset.initial_forms,
↪ self.formset.get_queryset()):
271:            view_on_site_url =
↪ self.opts.get_view_on_site_url(original)
272:            yield InlineAdminForm(
273:                self.formset, form, self.fieldsets,
↪ self.prepopulated_fields,
274:                original, readonly_fields_for_editing,
↪ model_admin=self.opts,
275:                view_on_site_url=view_on_site_url,
276:            )
277:        for form in self.formset.extra_forms:
278:            yield InlineAdminForm(
279:                self.formset, form, self.fieldsets,
↪ self.prepopulated_fields,
280:                None, self.readonly_fields,
↪ model_admin=self.opts,
281:            )
282:        if self.has_add_permission:
283:            yield InlineAdminForm(
284:                self.formset, self.formset.empty_form,
285:                self.fieldsets, self.prepopulated_fields, None,
286:                self.readonly_fields, model_admin=self.opts,
287:            )
```

```
288:
289:    def fields(self):
290:        fk = getattr(self.formset, "fk", None)
291:        empty_form = self.formset.empty_form
292:        meta_labels = empty_form._meta.labels or {}
293:        meta_help_texts = empty_form._meta.help_texts or {}
294:        for i, field_name in
↪ enumerate(flatten_fieldsets(self.fieldsets)):
295:            if fk and fk.name == field_name:
296:                continue
297:            if not self.has_change_permission or field_name in
↪ self.readonly_fields:
298:                yield {
299:                    'name': field_name,
(125 more lines below)

(Open file: /django__django/django/contrib/admin/helpers.py)
(Current directory: /django__django)
bash-$
```

We have navigated to the relevant section of the 'helpers.py' file where the 'ReadOnlyPasswordHash-Widget' is mentioned. The comment on line 215 suggests that there is a special case handling for the 'ReadOnlyPasswordHashWidget'. The code on line 216 checks if the widget has a 'read_only' attribute and, if so, renders the widget directly.

To address the issue, we need to ensure that the label for the 'ReadOnlyPasswordHashWidget' does not have a 'for' attribute, as it points to a non-labelable element. We should look for the part of the code where the label is being rendered with the 'for' attribute and make the necessary changes.

Since the code snippet we are looking at does not show the rendering of the label, we need to find where the label is being rendered. It might be in the template files or somewhere else in the code. Let's search for the term "label" in the current file to find where labels are being rendered.

```
search_file "label"
```

```
Found 23 matches for "label" in
↪ /django__django/django/contrib/admin/helpers.py:
Line 5:    display_for_field, flatten_fieldsets,
↪ help_text_for_field, label_for_field,
Line 23:    action = forms.ChoiceField(label=_('Action:'))
Line 25:        label='',
Line 131:    def label_tag(self):
Line 133:        contents = conditional_escape(self.field.label)
Line 142:        # checkboxes should not have a label suffix as the
↪ checkbox appears
Line 143:        # to the left of the label.
Line 144:        return self.field.label_tag(
Line 146:            label_suffix='' if self.is_checkbox else None,
Line 163:        if form._meta.labels and class_name in
↪ form._meta.labels:
Line 164:            label = form._meta.labels[class_name]
Line 166:            label = label_for_field(field,
↪ form._meta.model, model_admin, form=form)
Line 175:        'label': label,
Line 186:    def label_tag(self):
Line 190:        label = self.field['label']
```

```
Line 191:            return format_html('<label{}>{}{}</label>',
↪  flatatt(attrs), capfirst(label), self.form.label_suffix)
Line 195:               remote_field.model._meta.app_label,
Line 292:         meta_labels = empty_form._meta.labels or {}
Line 300:                   'label': meta_labels.get(field_name)
↪  or label_for_field(
Line 312:              label = form_field.label
Line 313:              if label is None:
Line 314:                  label = label_for_field(field_name,
↪  self.opts.model, self.opts, form=empty_form)
Line 317:                  'label': label,
End of matches for "label" in
↪  /django__django/django/contrib/admin/helpers.py

(Open file: /django__django/django/contrib/admin/helpers.py)
(Current directory: /django__django)
bash-$
```

---

**SWE-agent**                                         **django__django-14411**

The search results show several instances where labels are being handled in the 'helpers.py' file. The
most promising lines for our issue seem to be around line 191, where a label tag is being formatted with
a 'for' attribute. Let's navigate to that line to examine the context and see if this is where we need to
make our changes.

---

```
goto 191
```

---

**Observation**                                       **django__django-14411**

```
[File: /django__django/django/contrib/admin/helpers.py (424 lines
↪  total)]
(175 more lines above)
176:            'help_text': help_text,
177:            'field': field,
178:        }
179:        self.form = form
180:        self.model_admin = model_admin
181:        self.is_first = is_first
182:        self.is_checkbox = False
183:        self.is_readonly = True
184:        self.empty_value_display =
↪  model_admin.get_empty_value_display()
185:
186:    def label_tag(self):
187:        attrs = {}
188:        if not self.is_first:
189:            attrs["class"] = "inline"
190:        label = self.field['label']
191:        return format_html('<label{}>{}{}</label>',
↪  flatatt(attrs), capfirst(label), self.form.label_suffix)
192:
193:    def get_admin_url(self, remote_field, remote_obj):
194:        url_name = 'admin:%s_%s_change' % (
195:            remote_field.model._meta.app_label,
196:            remote_field.model._meta.model_name,
197:        )
198:        try:
199:            url = reverse(url_name,
↪  args=[quote(remote_obj.pk)])
```

```
200:            return format_html('<a href="{}">{}</a>', url,
↪   remote_obj)
201:        except NoReverseMatch:
202:            return str(remote_obj)
203:
204:    def contents(self):
205:        from django.contrib.admin.templatetags.admin_list
↪   import _boolean_icon
206:        field, obj, model_admin = self.field['field'],
↪   self.form.instance, self.model_admin
207:        try:
208:            f, attr, value = lookup_field(field, obj,
↪   model_admin)
209:        except (AttributeError, ValueError,
↪   ObjectDoesNotExist):
210:            result_repr = self.empty_value_display
211:        else:
212:            if field in self.form.fields:
213:                widget = self.form[field].field.widget
214:                # This isn't elegant but suffices for
↪   contrib.auth's
215:                # ReadOnlyPasswordHashWidget.
216:                if getattr(widget, 'read_only', False):
217:                    return widget.render(field, value)
218:            if f is None:
219:                if getattr(attr, 'boolean', False):
220:                    result_repr = _boolean_icon(value)
221:                else:
222:                    if hasattr(value, "__html__"):
223:                        result_repr = value
224:                    else:
225:                        result_repr = linebreaksbr(value)
226:            else:
227:                if isinstance(f.remote_field, ManyToManyRel)
↪   and value is not None:
228:                    result_repr = ", ".join(map(str,
↪   value.all()))
229:                elif (
230:                    isinstance(f.remote_field,
↪   (ForeignObjectRel, OneToOneField)) and
231:                    value is not None
232:                ):
233:                    result_repr =
↪   self.get_admin_url(f.remote_field, value)
234:                else:
235:                    result_repr = display_for_field(value, f,
↪   self.empty_value_display)
236:                result_repr = linebreaksbr(result_repr)
237:        return conditional_escape(result_repr)
238:
239:
240:class InlineAdminFormSet:
241:    """
242:    A wrapper around an inline formset for use in the admin
↪   system.
243:    """
244:    def __init__(self, inline, formset, fieldsets,
↪   prepopulated_fields=None,
245:                 readonly_fields=None, model_admin=None,
↪   has_add_permission=True,
246:                 has_change_permission=True,
↪   has_delete_permission=True,
247:                 has_view_permission=True):
```

```
248:        self.opts = inline
249:        self.formset = formset
250:        self.fieldsets = fieldsets
251:        self.model_admin = model_admin
252:        if readonly_fields is None:
253:            readonly_fields = ()
254:        self.readonly_fields = readonly_fields
255:        if prepopulated_fields is None:
256:            prepopulated_fields = {}
257:        self.prepopulated_fields = prepopulated_fields
258:        self.classes = ' '.join(inline.classes) if
↪  inline.classes else ''
259:        self.has_add_permission = has_add_permission
260:        self.has_change_permission = has_change_permission
261:        self.has_delete_permission = has_delete_permission
262:        self.has_view_permission = has_view_permission
263:
264:    def __iter__(self):
265:        if self.has_change_permission:
266:            readonly_fields_for_editing = self.readonly_fields
267:        else:
268:            readonly_fields_for_editing = self.readonly_fields +
↪  flatten_fieldsets(self.fieldsets)
269:
270:        for form, original in zip(self.formset.initial_forms,
↪  self.formset.get_queryset()):
271:            view_on_site_url =
↪  self.opts.get_view_on_site_url(original)
272:            yield InlineAdminForm(
273:                self.formset, form, self.fieldsets,
↪  self.prepopulated_fields,
274:                original, readonly_fields_for_editing,
↪  model_admin=self.opts,
275:                view_on_site_url=view_on_site_url,
(149 more lines below)

(Open file: /django__django/django/contrib/admin/helpers.py)
(Current directory: /django__django)
bash-$
```

---

**SWE-agent**                                                    **django__django-14411**

We are now looking at the section of the 'helpers.py' file where a label tag is being generated. However, this label tag does not include a 'for' attribute, which means this is not the part of the code responsible for the issue described.

The issue states that the label for the 'ReadOnlyPasswordHashWidget' has a 'for' attribute pointing to a non-labelable element. Since the 'ReadOnlyPasswordHashWidget' is likely a custom widget used in the Django admin, the rendering of its label might be handled in a template or in the widget's own Python code.

Given that the 'ReadOnlyPasswordHashWidget' is part of the 'django.contrib.auth' app, we should look into the 'forms.py' file in the 'django/contrib/auth' directory, as it might contain the definition of the widget and the rendering of its label.

Let's open the 'forms.py' file in the 'django/contrib/auth' directory to investigate further.

---

```
open django/contrib/auth/forms.py
```

```
[File: /django__django/django/contrib/auth/forms.py (440 lines
↪  total)]
1:import unicodedata
2:
3:from django import forms
4:from django.contrib.auth import (
5:    authenticate, get_user_model, password_validation,
6:)
7:from django.contrib.auth.hashers import (
8:    UNUSABLE_PASSWORD_PREFIX, identify_hasher,
9:)
10:from django.contrib.auth.models import User
11:from django.contrib.auth.tokens import default_token_generator
12:from django.contrib.sites.shortcuts import get_current_site
13:from django.core.exceptions import ValidationError
14:from django.core.mail import EmailMultiAlternatives
15:from django.template import loader
16:from django.utils.encoding import force_bytes
17:from django.utils.http import urlsafe_base64_encode
18:from django.utils.text import capfirst
19:from django.utils.translation import gettext, gettext_lazy as _
20:
21:UserModel = get_user_model()
22:
23:
24:def _unicode_ci_compare(s1, s2):
25:    """
26:    Perform case-insensitive comparison of two identifiers,
↪  using the
27:    recommended algorithm from Unicode Technical Report 36,
↪  section
28:    2.11.2(B)(2).
29:    """
30:    return unicodedata.normalize('NFKC', s1).casefold() ==
↪  unicodedata.normalize('NFKC', s2).casefold()
31:
32:
33:class ReadOnlyPasswordHashWidget(forms.Widget):
34:    template_name = 'auth/widgets/read_only_password_hash.html'
35:    read_only = True
36:
37:    def get_context(self, name, value, attrs):
38:        context = super().get_context(name, value, attrs)
39:        summary = []
40:        if not value or
↪  value.startswith(UNUSABLE_PASSWORD_PREFIX):
41:            summary.append({'label': gettext("No password
↪  set.")})
42:        else:
43:            try:
44:                hasher = identify_hasher(value)
45:            except ValueError:
46:                summary.append({'label': gettext("Invalid
↪  password format or unknown hashing algorithm.")})
47:            else:
48:                for key, value_ in
↪  hasher.safe_summary(value).items():
49:                    summary.append({'label': gettext(key),
↪  'value': value_})
50:        context['summary'] = summary
51:        return context
```

```python
52:
53:
54:class ReadOnlyPasswordHashField(forms.Field):
55:    widget = ReadOnlyPasswordHashWidget
56:
57:    def __init__(self, *args, **kwargs):
58:        kwargs.setdefault("required", False)
59:        kwargs.setdefault('disabled', True)
60:        super().__init__(*args, **kwargs)
61:
62:
63:class UsernameField(forms.CharField):
64:    def to_python(self, value):
65:        return unicodedata.normalize('NFKC',
↪  super().to_python(value))
66:
67:    def widget_attrs(self, widget):
68:        return {
69:            **super().widget_attrs(widget),
70:            'autocapitalize': 'none',
71:            'autocomplete': 'username',
72:        }
73:
74:
75:class UserCreationForm(forms.ModelForm):
76:    """
77:    A form that creates a user, with no privileges, from the
↪  given username and
78:    password.
79:    """
80:    error_messages = {
81:        'password_mismatch': _('The two password fields didn't
↪  match.'),
82:    }
83:    password1 = forms.CharField(
84:        label=_("Password"),
85:        strip=False,
86:        widget=forms.PasswordInput(attrs={'autocomplete':
↪  'new-password'}),
87:        help_text=password_validation.password_validators_help_⌋
↪  text_html(),
88:    )
89:    password2 = forms.CharField(
90:        label=_("Password confirmation"),
91:        widget=forms.PasswordInput(attrs={'autocomplete':
↪  'new-password'}),
92:        strip=False,
93:        help_text=_("Enter the same password as before, for
↪  verification."),
94:    )
95:
96:    class Meta:
97:        model = User
98:        fields = ("username",)
99:        field_classes = {'username': UsernameField}
100:
(340 more lines below)

(Open file: /django__django/django/contrib/admin/helpers.py)
(Current directory: /django__django)
bash-$
```

We have opened the 'forms.py' file in the 'django/contrib/auth' directory and found the definition of the 'ReadOnlyPasswordHashWidget' starting at line 33. The widget uses a template named 'read_only_password_hash.html' for rendering, as specified on line 34.

Since the issue is related to the label's 'for' attribute in the rendered HTML, we should look at the template file 'read_only_password_hash.html' to see how the label is being rendered. Let's use the 'find_file' command to locate this template file within the repository.

---

```
find_file "read_only_password_hash.html"
```

```
Found 1 matches for "read_only_password_hash.html" in
↪   /django__django:
/django__django/django/contrib/auth/templates/auth/widgets/read_on⌋
↪   ly_password_hash.html

(Open file: /django__django/django/contrib/auth/forms.py)
(Current directory: /django__django)
bash-$
```

We have located the template file 'read_only_password_hash.html' in the 'django/contrib/auth/templates/auth/widgets' directory. Let's open this file to inspect the HTML and see how the label is being rendered.

---

```
open django/contrib/auth/templates/auth/widgets/read_only_password⌋
↪   _hash.html
```

```
[File: /django__django/django/contrib/auth/templates/auth/widgets/⌋
↪   read_only_password_hash.html (5 lines
↪   total)]
1:<div{% include 'django/forms/widgets/attrs.html' %}>
2:{% for entry in summary %}
3:{{ entry.label }}{% if entry.value %}: {{
↪   entry.value }}{% endif %}
4:{% endfor %}
5:</div>

(Open file: /django__django/django/contrib/auth/forms.py)
(Current directory: /django__django)
bash-$
```

The template file 'read_only_password_hash.html' does not show a 'label' element with a 'for' attribute, which means the issue might be elsewhere. The issue description mentions that the label element for the 'ReadOnlyPasswordHashWidget' has a 'for' attribute pointing to a non-labelable element. Since we don't see a 'label' element here, the 'for' attribute is likely added in the Python code that generates the HTML for the widget or in a different part of the template.

Given that the 'ReadOnlyPasswordHashWidget' is a custom widget, the rendering of the label with the 'for' attribute might be handled in the Python code where the widget is used in a form, rather than in the widget's template itself.

Let's go back to the 'forms.py' file and look for the usage of 'ReadOnlyPasswordHashWidget' in a form where a label might be rendered with a 'for' attribute. We should look for a form that includes 'ReadOnlyPasswordHashWidget' and inspect how the label is being generated.

---

```
open django/contrib/auth/forms.py
```

**Observation**                                                    **django__django-14411**

```
[File: /django__django/django/contrib/auth/forms.py (440 lines
↪  total)]
1:import unicodedata
2:
3:from django import forms
4:from django.contrib.auth import (
5:    authenticate, get_user_model, password_validation,
6:)
7:from django.contrib.auth.hashers import (
8:    UNUSABLE_PASSWORD_PREFIX, identify_hasher,
9:)
10:from django.contrib.auth.models import User
11:from django.contrib.auth.tokens import default_token_generator
12:from django.contrib.sites.shortcuts import get_current_site
13:from django.core.exceptions import ValidationError
14:from django.core.mail import EmailMultiAlternatives
15:from django.template import loader
16:from django.utils.encoding import force_bytes
17:from django.utils.http import urlsafe_base64_encode
18:from django.utils.text import capfirst
19:from django.utils.translation import gettext, gettext_lazy as _
20:
21:UserModel = get_user_model()
22:
23:
24:def _unicode_ci_compare(s1, s2):
25:    """
26:    Perform case-insensitive comparison of two identifiers,
↪  using the
27:    recommended algorithm from Unicode Technical Report 36,
↪  section
28:    2.11.2(B)(2).
29:    """
30:    return unicodedata.normalize('NFKC', s1).casefold() ==
↪  unicodedata.normalize('NFKC', s2).casefold()
31:
32:
33:class ReadOnlyPasswordHashWidget(forms.Widget):
34:    template_name = 'auth/widgets/read_only_password_hash.html'
35:    read_only = True
36:
37:    def get_context(self, name, value, attrs):
38:        context = super().get_context(name, value, attrs)
39:        summary = []
40:        if not value or
↪  value.startswith(UNUSABLE_PASSWORD_PREFIX):
41:            summary.append({'label': gettext("No password
↪  set.")})
42:        else:
```

```
43:            try:
44:                hasher = identify_hasher(value)
45:            except ValueError:
46:                summary.append({'label': gettext("Invalid
↪  password format or unknown hashing algorithm.")})
47:            else:
48:                for key, value_ in
↪  hasher.safe_summary(value).items():
49:                    summary.append({'label': gettext(key),
↪  'value': value_})
50:        context['summary'] = summary
51:        return context
52:
53:
54:class ReadOnlyPasswordHashField(forms.Field):
55:    widget = ReadOnlyPasswordHashWidget
56:
57:    def __init__(self, *args, **kwargs):
58:        kwargs.setdefault("required", False)
59:        kwargs.setdefault('disabled', True)
60:        super().__init__(*args, **kwargs)
61:
62:
63:class UsernameField(forms.CharField):
64:    def to_python(self, value):
65:        return unicodedata.normalize('NFKC',
↪  super().to_python(value))
66:
67:    def widget_attrs(self, widget):
68:        return {
69:            **super().widget_attrs(widget),
70:            'autocapitalize': 'none',
71:            'autocomplete': 'username',
72:        }
73:
74:
75:class UserCreationForm(forms.ModelForm):
76:    """
77:    A form that creates a user, with no privileges, from the
↪  given username and
78:    password.
79:    """
80:    error_messages = {
81:        'password_mismatch': _('The two password fields didn't
↪  match.'),
82:    }
83:    password1 = forms.CharField(
84:        label=_("Password"),
85:        strip=False,
86:        widget=forms.PasswordInput(attrs={'autocomplete':
↪  'new-password'}),
87:        help_text=password_validation.password_validators_help_⌋
↪  text_html(),
88:    )
89:    password2 = forms.CharField(
90:        label=_("Password confirmation"),
91:        widget=forms.PasswordInput(attrs={'autocomplete':
↪  'new-password'}),
92:        strip=False,
93:        help_text=_("Enter the same password as before, for
↪  verification."),
94:    )
95:
```

```
96:    class Meta:
97:        model = User
98:        fields = ("username",)
99:        field_classes = {'username': UsernameField}
100:
(340 more lines below)

(Open file: /django__django/django/contrib/auth/templates/auth/wid⌐
↪  gets/read_only_password_hash.html)
(Current directory: /django__django)
bash-$
```

---

### SWE-agent                                              django__django-14411

We are back in the 'forms.py' file where the 'ReadOnlyPasswordHashWidget' is defined. Since the
issue is with the label's 'for' attribute, we need to find where the 'ReadOnlyPasswordHashField' is used
in a form and how the label is being rendered there.
The 'ReadOnlyPasswordHashField' is a custom field that uses the 'ReadOnlyPasswordHashWidget',
and it's likely used in the admin forms for the User model. We should search for the usage of
'ReadOnlyPasswordHashField' within this file to find where it's included in a form.
Let's use the 'search_file' command to search for "ReadOnlyPasswordHashField" in the current file.

---

```
search_file "ReadOnlyPasswordHashField"
```

---

### Observation                                            django__django-14411

```
Found 2 matches for "ReadOnlyPasswordHashField" in
↪  /django__django/django/contrib/auth/forms.py:
Line 54:class ReadOnlyPasswordHashField(forms.Field):
Line 136:    password = ReadOnlyPasswordHashField(
End of matches for "ReadOnlyPasswordHashField" in
↪  /django__django/django/contrib/auth/forms.py

(Open file: /django__django/django/contrib/auth/forms.py)
(Current directory: /django__django)
bash-$
```

---

### SWE-agent                                              django__django-14411

We have found two references to 'ReadOnlyPasswordHashField' in the 'forms.py' file. The first
reference on line 54 is the class definition itself, and the second reference on line 136 is likely where
the field is used in a form.
Let's navigate to line 136 to see the context in which 'ReadOnlyPasswordHashField' is used and how
the label is being rendered.

---

```
goto 136
```

---

### Observation                                            django__django-14411

```
[File: /django__django/django/contrib/auth/forms.py (440 lines
↪  total)]
(120 more lines above)
121:        if password:
122:            try:
```

```
123:                    password_validation.validate_password(password,
→  self.instance)
124:            except ValidationError as error:
125:                self.add_error('password2', error)
126:
127:    def save(self, commit=True):
128:        user = super().save(commit=False)
129:        user.set_password(self.cleaned_data["password1"])
130:        if commit:
131:            user.save()
132:        return user
133:
134:
135:class UserChangeForm(forms.ModelForm):
136:    password = ReadOnlyPasswordHashField(
137:        label=_("Password"),
138:        help_text=_(
139:            'Raw passwords are not stored, so there is no way
→  to see this '
140:            'user's password, but you can change the password
→  using '
141:            '<a href="{}">this form</a>.'
142:        ),
143:    )
144:
145:    class Meta:
146:        model = User
147:        fields = '__all__'
148:        field_classes = {'username': UsernameField}
149:
150:    def __init__(self, *args, **kwargs):
151:        super().__init__(*args, **kwargs)
152:        password = self.fields.get('password')
153:        if password:
154:            password.help_text =
→  password.help_text.format('../password/')
155:        user_permissions = self.fields.get('user_permissions')
156:        if user_permissions:
157:            user_permissions.queryset =
→  user_permissions.queryset.select_related('content_type')
158:
159:
160:class AuthenticationForm(forms.Form):
161:    """
162:    Base class for authenticating users. Extend this to get a
→  form that accepts
163:    username/password logins.
164:    """
165:    username =
→  UsernameField(widget=forms.TextInput(attrs={'autofocus':
→  True}))
166:    password = forms.CharField(
167:        label=_("Password"),
168:        strip=False,
169:        widget=forms.PasswordInput(attrs={'autocomplete':
→  'current-password'}),
170:    )
171:
172:    error_messages = {
173:        'invalid_login': _(
174:            "Please enter a correct %(username)s and password.
→  Note that both "
175:            "fields may be case-sensitive."
```

```python
176:        ),
177:        'inactive': _("This account is inactive."),
178:    }
179:
180:    def __init__(self, request=None, *args, **kwargs):
181:        """
182:        The 'request' parameter is set for custom auth use by
↪  subclasses.
183:        The form data comes in via the standard 'data' kwarg.
184:        """
185:        self.request = request
186:        self.user_cache = None
187:        super().__init__(*args, **kwargs)
188:
189:        # Set the max length and label for the "username"
↪  field.
190:        self.username_field =
↪  UserModel._meta.get_field(UserModel.USERNAME_FIELD)
191:        username_max_length = self.username_field.max_length or
↪  254
192:        self.fields['username'].max_length =
↪  username_max_length
193:        self.fields['username'].widget.attrs['maxlength'] =
↪  username_max_length
194:        if self.fields['username'].label is None:
195:            self.fields['username'].label =
↪  capfirst(self.username_field.verbose_name)
196:
197:    def clean(self):
198:        username = self.cleaned_data.get('username')
199:        password = self.cleaned_data.get('password')
200:
201:        if username is not None and password:
202:            self.user_cache = authenticate(self.request,
↪  username=username, password=password)
203:            if self.user_cache is None:
204:                raise self.get_invalid_login_error()
205:            else:
206:                self.confirm_login_allowed(self.user_cache)
207:
208:        return self.cleaned_data
209:
210:    def confirm_login_allowed(self, user):
211:        """
212:        Controls whether the given User may log in. This is a
↪  policy setting,
213:        independent of end-user authentication. This default
↪  behavior is to
214:        allow login by active users, and reject login by
↪  inactive users.
215:
216:        If the given user cannot log in, this method should
↪  raise a
217:        ``ValidationError``.
218:
219:        If the given user may log in, this method should return
↪  None.
220:        """
(220 more lines below)

(Open file: /django__django/django/contrib/auth/forms.py)
(Current directory: /django__django)
bash-$
```

We are now looking at the 'UserChangeForm' class where the 'ReadOnlyPasswordHashField' is used. The field is instantiated with a label on line 136, and the help text includes an HTML anchor tag for changing the password.

The issue reported is that the label element for the 'ReadOnlyPasswordHashWidget' has a 'for' attribute which points to a non-labelable element. However, in the 'UserChangeForm', the label is defined in the Python code and not in the HTML template. This suggests that the 'for' attribute is likely added automatically by Django's form rendering.

To resolve the issue, we need to ensure that when the form is rendered, the label for the 'ReadOnly-PasswordHashField' does not include a 'for' attribute. Since the rendering of form fields to HTML is handled by Django's form rendering system, we may need to customize the rendering for this specific field to exclude the 'for' attribute.

One way to do this is to override the 'label_tag' method for the 'ReadOnlyPasswordHashField' to ensure it does not render a 'for' attribute. Let's edit the 'ReadOnlyPasswordHashField' class to include this customization.

We will add a custom 'label_tag' method to the 'ReadOnlyPasswordHashField' class that returns a label without the 'for' attribute.

---

```
edit 54:60
class ReadOnlyPasswordHashField(forms.Field):
    widget = ReadOnlyPasswordHashWidget

    def __init__(self, *args, **kwargs):
        kwargs.setdefault("required", False)
        kwargs.setdefault('disabled', True)
        super().__init__(*args, **kwargs)

    def label_tag(self, contents=None, attrs=None,
    ↪  label_suffix=None):
        contents = contents or self.label
        if label_suffix is not None and label_suffix not in
        ↪  contents:
            contents += label_suffix
        widget = self.widget
        id_ = widget.attrs.get('id') or ''
        return format_html('<label{}>{}</label>',
        ↪  format_html_join('', ' {}="{}"', sorted(attrs.items())
        ↪  if attrs else ()), contents)
end_of_edit
```

```
Your proposed edit has introduced new syntax error(s). Please
↪  understand the fixes and retry your edit commmand.

ERRORS:
- F821 undefined name 'format_html'
- F821 undefined name 'format_html_join'

This is how your edit would have looked if applied
-------------------------------------------------
[File: /django__django/django/contrib/auth/forms.py (448 lines
↪  total)]
(47 more lines above)
48:                for key, value_ in
↪  hasher.safe_summary(value).items():
49:                    summary.append({'label': gettext(key),
↪  'value': value_})
50:        context['summary'] = summary
```

```
51:        return context
52:
53:
54:class ReadOnlyPasswordHashField(forms.Field):
55:    widget = ReadOnlyPasswordHashWidget
56:
57:    def __init__(self, *args, **kwargs):
58:        kwargs.setdefault("required", False)
59:        kwargs.setdefault('disabled', True)
60:        super().__init__(*args, **kwargs)
61:
62:    def label_tag(self, contents=None, attrs=None,
↪  label_suffix=None):
63:        contents = contents or self.label
64:        if label_suffix is not None and label_suffix not in
↪  contents:
65:            contents += label_suffix
66:        widget = self.widget
67:        id_ = widget.attrs.get('id') or ''
68:        return format_html('<label{}>{}</label>',
↪  format_html_join('', ' {}="{}"', sorted(attrs.items()) if attrs
↪  else ()), contents)
69:
70:
71:class UsernameField(forms.CharField):
72:    def to_python(self, value):
(376 more lines below)
--------------------------------------------------

This is the original code before your edit
--------------------------------------------------
[File: /django__django/django/contrib/auth/forms.py (440 lines
↪  total)]
(48 more lines above)
49:                    summary.append({'label': gettext(key),
↪  'value': value_})
50:        context['summary'] = summary
51:        return context
52:
53:
54:class ReadOnlyPasswordHashField(forms.Field):
55:    widget = ReadOnlyPasswordHashWidget
56:
57:    def __init__(self, *args, **kwargs):
58:        kwargs.setdefault("required", False)
59:        kwargs.setdefault('disabled', True)
60:        super().__init__(*args, **kwargs)
61:
62:
63:class UsernameField(forms.CharField):
64:    def to_python(self, value):
65:        return unicodedata.normalize('NFKC',
↪  super().to_python(value))
(375 more lines below)
--------------------------------------------------
Your changes have NOT been applied. Please fix your edit command
↪  and try again.
You either need to 1) Specify the correct start/end line arguments
↪  or 2) Correct your edit code.
DO NOT re-run the same failed edit command. Running it again will
↪  lead to the same error.

(Open file: /django__django/django/contrib/auth/forms.py)
```

```
(Current directory: /django__django)
bash-$
```

The error message indicates that the names 'format_html' and 'format_html_join' are undefined. These functions are utility functions provided by Django for safely creating HTML output. To use them, we need to import them from 'django.utils.html'.
Let's correct the edit by adding the necessary imports and retrying the edit command.
────────────────────────────────────────────────────────────────────

```
edit 54:60
from django.utils.html import format_html, format_html_join

class ReadOnlyPasswordHashField(forms.Field):
    widget = ReadOnlyPasswordHashWidget

    def __init__(self, *args, **kwargs):
        kwargs.setdefault("required", False)
        kwargs.setdefault('disabled', True)
        super().__init__(*args, **kwargs)

    def label_tag(self, contents=None, attrs=None,
    ↪  label_suffix=None):
        contents = contents or self.label
        if label_suffix is not None and label_suffix not in
        ↪  contents:
            contents += label_suffix
        return format_html('<label>{}</label>', contents)
end_of_edit
```

```
[File: /django__django/django/contrib/auth/forms.py (448 lines
↪  total)]
(3 more lines above)
4:from django.contrib.auth import (
5:    authenticate, get_user_model, password_validation,
6:)
7:from django.contrib.auth.hashers import (
8:    UNUSABLE_PASSWORD_PREFIX, identify_hasher,
9:)
10:from django.contrib.auth.models import User
11:from django.contrib.auth.tokens import default_token_generator
12:from django.contrib.sites.shortcuts import get_current_site
13:from django.core.exceptions import ValidationError
14:from django.core.mail import EmailMultiAlternatives
15:from django.template import loader
16:from django.utils.encoding import force_bytes
17:from django.utils.http import urlsafe_base64_encode
18:from django.utils.text import capfirst
19:from django.utils.translation import gettext, gettext_lazy as _
20:
21:UserModel = get_user_model()
22:
23:
24:def _unicode_ci_compare(s1, s2):
25:    """
26:    Perform case-insensitive comparison of two identifiers,
↪  using the
```

```
27:    recommended algorithm from Unicode Technical Report 36,
↪  section
28:    2.11.2(B)(2).
29:    """
30:    return unicodedata.normalize('NFKC', s1).casefold() ==
↪  unicodedata.normalize('NFKC', s2).casefold()
31:
32:
33:class ReadOnlyPasswordHashWidget(forms.Widget):
34:    template_name = 'auth/widgets/read_only_password_hash.html'
35:    read_only = True
36:
37:    def get_context(self, name, value, attrs):
38:        context = super().get_context(name, value, attrs)
39:        summary = []
40:        if not value or
↪  value.startswith(UNUSABLE_PASSWORD_PREFIX):
41:            summary.append({'label': gettext("No password
↪  set.")})
42:        else:
43:            try:
44:                hasher = identify_hasher(value)
45:            except ValueError:
46:                summary.append({'label': gettext("Invalid
↪  password format or unknown hashing algorithm.")})
47:            else:
48:                for key, value_ in
↪  hasher.safe_summary(value).items():
49:                    summary.append({'label': gettext(key),
↪  'value': value_})
50:        context['summary'] = summary
51:        return context
52:
53:
54:from django.utils.html import format_html, format_html_join
55:
56:class ReadOnlyPasswordHashField(forms.Field):
57:    widget = ReadOnlyPasswordHashWidget
58:
59:    def __init__(self, *args, **kwargs):
60:        kwargs.setdefault("required", False)
61:        kwargs.setdefault('disabled', True)
62:        super().__init__(*args, **kwargs)
63:
64:    def label_tag(self, contents=None, attrs=None,
↪  label_suffix=None):
65:        contents = contents or self.label
66:        if label_suffix is not None and label_suffix not in
↪  contents:
67:            contents += label_suffix
68:        return format_html('<label>{}</label>', contents)
69:
70:
71:class UsernameField(forms.CharField):
72:    def to_python(self, value):
73:        return unicodedata.normalize('NFKC',
↪  super().to_python(value))
74:
75:    def widget_attrs(self, widget):
76:        return {
77:            **super().widget_attrs(widget),
78:            'autocapitalize': 'none',
79:            'autocomplete': 'username',
```

```
80:        }
81:
82:
83:class UserCreationForm(forms.ModelForm):
84:    """
85:    A form that creates a user, with no privileges, from the
↪  given username and
86:    password.
87:    """
88:    error_messages = {
89:        'password_mismatch': _('The two password fields didn't
↪  match.'),
90:    }
91:    password1 = forms.CharField(
92:        label=_("Password"),
93:        strip=False,
94:        widget=forms.PasswordInput(attrs={'autocomplete':
↪  'new-password'}),
95:        help_text=password_validation.password_validators_help_
↪  text_html(),
96:    )
97:    password2 = forms.CharField(
98:        label=_("Password confirmation"),
99:        widget=forms.PasswordInput(attrs={'autocomplete':
↪  'new-password'}),
100:        strip=False,
101:        help_text=_("Enter the same password as before, for
↪  verification."),
102:    )
103:
(345 more lines below)
File updated. Please review the changes and make sure they are
↪  correct (correct indentation, no duplicate lines, etc). Edit
↪  the file again if necessary.

(Open file: /django__django/django/contrib/auth/forms.py)
(Current directory: /django__django)
bash-$
```

The changes to the 'ReadOnlyPasswordHashField' class have been successfully applied, including the import of 'format_html' and 'format_html_join', and the addition of a custom 'label_tag' method that returns a label without the 'for' attribute.

Before submitting the changes, it would be prudent to test that the changes have the desired effect. However, since we cannot run a Django server or render forms in this environment, we will proceed with submitting the changes.

Let's submit the changes to the code base using the 'submit' command.

---

```
submit
```

```
diff --git a/django/contrib/auth/forms.py
↪  b/django/contrib/auth/forms.py
index 20d8922799..1f249a88dd 100644
--- a/django/contrib/auth/forms.py
+++ b/django/contrib/auth/forms.py
```

```
@@ -51,6 +51,8 @@ class ReadOnlyPasswordHashWidget(forms.Widget):
         return context

+from django.utils.html import format_html, format_html_join
+
 class ReadOnlyPasswordHashField(forms.Field):
     widget = ReadOnlyPasswordHashWidget

@@ -59,6 +61,12 @@ class ReadOnlyPasswordHashField(forms.Field):
         kwargs.setdefault('disabled', True)
         super().__init__(*args, **kwargs)

+    def label_tag(self, contents=None, attrs=None,
↪  label_suffix=None):
+        contents = contents or self.label
+        if label_suffix is not None and label_suffix not in
↪  contents:
+            contents += label_suffix
+        return format_html('<label>{}</label>', contents)
+

 class UsernameField(forms.CharField):
     def to_python(self, value):
```

# E   Statements

## E.1   Ethics & Broader Impacts

Prior works have established how the growing usage of LMs that can code also carries a certain amount of risk. We identify three main risks that could arise when building and using a system like SWE-agent, then discuss how we incorporates measures that mitigate such problems.

First is the security risks that come with executing LM-generated code on device. When evaluating on software engineering tasks that feature an incredibly diverse number of issue descriptions, running code generations on a personal computing device can have negative side effects, such as the unintentional removal of digital assets (e.g., `rm -rf asset/`). To defend against this, we design SWE-agent to use ephemeral containers for both inference and evaluation. SWE-agent's execution environment and the SWE-bench evaluation framework are both carried out in sand-boxed code environments, which is made possible with Docker. Executing code in a Docker container ensures that its effects are mostly isolated from the rest of the system. While not considered as secure as virtualized hardware isolation, the namespace isolation provided by Docker containers is deemed sufficient for code that is not deliberately engineered to exploit recent container escape vulnerabilities. More details are discussion is in §A.2.

Second, if the wider community develops interest for SWE-agent and builds upon it, it is also possible that illegitimate evaluation datasets or infrastructure can be used to inject testing devices with malicious code or instructions to generate malicious code. For instance, an unofficial repository claiming to host an inference/evaluation harness for SWE-agent/bench could include a task instance with an issue description that tells the LM agent to build key logging functionality and store it in a hidden folder. To eliminate confusion and reduce the possibility of such an event, we provide clear guidelines listed on our GitHub repositories, data stores, and websites indicating the official repositories and channels that we actively maintain. We also encourage third parties to incorporate any improvements into our codebase and help with integrating such contributions.

Lastly are the consequences of software engineering agents being deployed in the real world. Prior works have conceptualized and put forth prototypes of agents that can carry out offensive security measures. It is also not difficult to imagine that a system like SWE-agent can be incorporated into pipelines resulting in the production of malicious code. SWE-agent's strong performance on SWE-bench implies that future AI systems will likely be increasingly adept in the aforementioned use cases. Releasing SWE-agent as an open source tool can support research towards designing sound, effective constraints for what software engineering agents are permitted to do. It can also serve as a system that legal experts and policy-making entities can experiment with to shape the future of what AI-driven end to end software engineering could look like.

## E.2   Reproducibility

To help the greater community reproduce the results presented in this paper and build on the SWE-agent platform, we open source all of our resources that were created for this project. The source code for the interactive pipeline, context management logic, command implementations, interface design, and everything else is entirely available in a GitHub repository. We provide extensive text and video documentation describing how to run and modify different parts of the codebase. Practitioners should be able to easily recover our findings by running the agent with simple scripts. We also open source all inference and evaluation artifacts (e.g., trajectories, code generations, evaluation execution traces, analysis notebooks). The results presented in the main and supplementary parts of this paper can be fully rendered from the data. Finally, we also maintain an active online help forum to assist with any reproduction problems or questions about how to build on ACI design and SWE-agent.

## E.3   Limitations & Future Work

The final SWE-agent configuration has a small toolkit, albeit highly effective. With SWE-agent's highly extensible design, we're excited by the prospect of adding more tools, such as web browsing or static analysis, that can leverage more signals from an issue description and codebase to improve the % Resolved performance. Many tools trialed by prior works from software engineering and language model agents, such as static/dynamic analysis, spectrum based fault localization, or test generation via fuzzing could prove useful.

Second, in this work, the ACI development process and case studies are done manually. Many components of SWE-agent were crafted from observations of recurring behavior within a single trajectory or across multiple trajectories. Automating part or all of this process could not only accelerate work built on top of SWE-agent, but also provide greater insights into developing ACI principles for agentic software engineering. Contemporary works have explored automated prompting to improve performance on traditional sequence to sequence tasks, supplanting the need for manual prompt design. Thinking about automating ACI design raises immediately interesting questions around how such systems can scrutinize and iterate upon their own designs. Ensuring such horizon leads to incremental performance improvements across a longer horizon is also a challenging question.

Finally, the scope of SWE-agent is exclusively focused on programmatic tasks like software engineering and code generation. We're curious to see whether the same principles of ACI and our observations of agent behavior are transferable to different domains. Recent work around applying LM agents to a variety of digital work applications have proliferated, such as use cases in education technology, data analysis, and enterprise workflows. We hope that thinking about improving performance of agentic workflows on these domains through the lens of ACI design can be a symbiotic process. For instance, for a task such a shopping on the web, in place of a typical Google-style search tool, could agents benefit from additional information beyond a list of each page's title and snippet? Would the design vary if the nature of the downstream task were to change slightly? For a completely different task, such as navigating an internal company knowledge base to help a recently on-boarded employee, how might the search interface be best adjusted to the agent?

Similar to the progression of the field of User Experience (UX) and Human Computer Interaction (HCI) research, applying ACI to other domains could not only yield improvements in downstream task performance, but also further expand the list of ACI principles. We believe that the fundamental motivations for ACI, the foundational principles we put forth, and our case study of SWE-agent as an instantiation of implementing and improving an ACI can motivate such work.

