# OpenReview forum: "SWE-agent: Agent-Computer Interfaces Enable Automated Software Engineering"
_NeurIPS.cc/2024/Conference — NeurIPS 2024 poster_

### Official Review · Reviewer_3ebt · 2024-07-09

**Soundness:** 2
**Presentation:** 3
**Contribution:** 2
**Rating:** 6
**Confidence:** 5

**Summary:**

Recently, Large Language Models (LLMs) based agents have revealed huge potential in automating tasks. In this paper,

**Strengths:**

This work has presented a powerful AI Agent designed for software engineering, which configured agents with multiple functions like file viewer, file editor and so on. Such a system can significantly improve the capability of agent in programming and doing some software engineering works. Experimental results also demonstrate the capability of SWE-agent in solving tasks in SWE-bench.

**Weaknesses:**

1. This paper is a good engineering work but lack some insights from the perspective of methodology. For example, why we must choose search, navigation, file viewer / editor? Is it possible to add some other operations? It will be better to add some observations or ablation studies to support the motivation of this design, not just some system claims (e.g., Line 85 - 101).
2. Compared with previous works like ReAct, SWE-agent more likes a version that extended with multiple sophisticated functions.
3. Authors claim an agent-computer interface, but actually, it focused on coding or like a software engineering. I don't think that the proposed method can be considered as an agent-computer interface, as computer usually involves more complex functions (e.g., drawing, gaming, web search, and so on). I think such a claim is a little over-claim.

**Questions:**

1. Can you highlight the differences between SWE-agent and MetaGPT? MetaGPT is a multi-agent programming framework designed from software engineering. It also incorporates many unique designs in terms of software engineering.
2. Can the proposed SWE-agent to other benchmarks? It seems like the design of SWE-agent is specialized for SWE-bench.

**Limitations:**

Please see my comments on weaknesses and questions.

---

> ### Author Rebuttal · Authors · 2024-08-07
>
> Thanks for your thorough review of the paper and suggestions - the feedback has been helpful to clarify several details.
>
> **W1: Why these tools?**:
>
>
> > We discuss the design of SWE-agent’s interface in _Section 3_ and our motivation for this design based on principles laid out in _Section 2_.
> The ablation studies discussed in _Table 3_ and _Section 5.1_ largely address how we settled on the final version of this interface.
> These tools are necessary because without them, SWE-agent cannot carry out key steps to solving a software engineering task (e.g. edit code, navigate file system, execute programs).
> Furthermore, we observed that the Shell-only agent typically invokes commands that do exactly these types of operations (e.g. grep, find, cat, sed).
>
> > It is possible to add additional tools. SWE-agent’s codebase is designed to make configuring an interface easy (Technical disc. in _Appendix A.3_ + Ample documentation in SWE-agent codebase). Since its introduction, several projects have built upon SWE-agent, adding features like improved software analysis and editing.
>
> **W2: ReAct vs. SWE-agent**
>
> > While SWE-agent’s fundamental interactive loop is inspired by ReAct, it is _not_ simply ReAct + tool use / programming language, which has been explored by other works (e.g. TroVE [1], InterCode [2], CodeAct [3]).
> While tools are a significant facet of an Agent Computer Interface, we discuss and show the qualitative (_Section [3,5.1]_, _Appendix [A,C]_) and quantitative (_Table [1, 2]_, _Appendix B_) impacts of an agent-friendly ACI, which encompasses far more elements such as context management, i/o format, guardrails, etc.
>
> > Prior works tend to focus on how providing a tool helps performance.
> We show that a tool can be configured (in terms of input/output, documentation, usage, implementation) in many ways, and that these choices have tangible impact on performance.
> As expanded upon in responses [1, 5], SWE-agent is certainly extensible and allows easy addition of more functions.
>
> > [1] https://arxiv.org/pdf/2401.12869
> > [2] https://arxiv.org/abs/2306.14898
> > [3] https://arxiv.org/abs/2402.01030
>
> **W3: ACI is Not General Purpose**
>
> > While SWE-agent was designed with the domain of software engineering in mind, our goal with ACI is to generally convey that developing specialized interfaces for LM agents (ACIs) leads to tangible impact on downstream performance. The structure of our paper reflects this. Our motivations for studying interactive environment design (_Section 1_) inspire our preliminary definition of ACI (_Section 2_). SWE-agent’s design process is a case study on implementing and iterating ACI in practice (_Sections 3, 5_).
>
> > Demonstrating ACI’s efficacy on a large number of agentic tasks (e.g. web browsing, theorem proving), would likely be an effort that would far exceed the breadth of what a single paper can cover. As alluded to in Response 2, we hope and have seen some evidence of how ACI principles have led to the recent flurry of SOTA improvements on SWE-bench.
>
> > There is no concrete work that applies ACI methods to non-code domains, but we hope that the concrete takeaways and SWE-bench performance improvement will inspire such explorations.
>
> **Q1: SWE-agent vs. MetaGPT**
>
> > MetaGPT puts forth a sophisticated network of multiple agents with different roles that follow a Standard Operating Procedure. Their work focuses more on coordinating multiple agents and communication interfaces between agents. MetaGPT also has not been evaluated on SWE-bench or other software engineering or repository-level coding benchmarks (e.g. RepoBench). They primarily run experiments on HumanEval and MBPP.
>
> > We do not study multiagent systems with SWE-agent. Our focus is fundamentally different - we identify pain points in how an LM agent interacts with a computer, then upgrade the interface that sits between (e.g. tools, I/O format, context management) to reduce common pitfalls and improve actions’ usability and feedback. We will add this comparison between MetaGPT and SWE-agent to our related work section.
>
> **Q2: Extensibility to other benchmarks**
>
> > SWE-agent _can be used_ for other coding benchmarks. Our paper presents results for SWE-agent on HumanEvalFix (see _Table 2_), achieving a 87.7% fix rate. SWE-agent also works for traditional code completion benchmarks (e.g. HumanEval, MBPP). However, we did not release such results as (1) performance on these benchmarks is near saturated, and (2) Such benchmarks only test for a minor subset of SWE-agent’s capabilities (e.g. localize error, reproduce bug, navigate file system are _not_ tested), drawing focus away from our work’s main deliverables.
>
> > While SWE-agent cannot be directly applied in its current form to non-coding agentic tasks such as web browsing (e.g. WebShop, WebArena), we believe and hope that future LM agent works can further study and build upon ACI design principles across different domains. This was briefly discussed in Appendix E, and in the latest draft, we have extended this discussion to provide more concrete inspiration for the community.
>
> Thank you again for your feedback! We’ve uploaded a new version of the paper which addresses your concerns. If you have any remaining questions, please let us know.

---

> > ### Comment · Reviewer_3ebt · 2024-08-09
> >
> > Thank you for your response. It has addressed my concerns, and I suggest authors to add these discussion into your paper to highlight your unique insights. I have raised my score.

---

### Official Review · Reviewer_KbN4 · 2024-07-13

**Soundness:** 4
**Presentation:** 3
**Contribution:** 4
**Rating:** 7
**Confidence:** 4

**Summary:**

This paper introduces SWE-agent, a system that enables language models to autonomously perform software engineering tasks by interacting with computers through a specially designed agent-computer interface (ACI). The authors argue that LM agents represent a new category of end users with unique requirements, necessitating interfaces tailored to their strengths and limitations. SWE-agent's ACI is designed to enhance an agent's ability to create and edit code files, navigate repositories, and execute tests and programs.

The paper evaluates SWE-agent on two benchmarks: SWE-bench and HumanEvalFix. On SWE-bench, SWE-agent achieves state-of-the-art performance with a pass@1 rate of 12.5%, significantly outperforming previous non-interactive approaches. On HumanEvalFix, it achieves an impressive 87.7% pass@1 rate. The authors provide detailed analyses of how different ACI design choices impact agent behavior and performance.

**Strengths:**

* Strong empirical results: SWE-agent demonstrates significant improvements over previous approaches on challenging benchmarks, particularly on SWE-bench where it more than triples the previous best published performance.
* Novel and important concept: The introduction of ACIs as a distinct category from human-computer interfaces is an important contribution that could shape future research in AI agent systems.
* Detailed analysis: The paper provides extensive ablation studies and analyses of agent behavior, offering valuable insights into how different interface design choices affect performance.

**Weaknesses:**

* Generalizability: The paper focuses primarily on software engineering tasks. It's unclear how well the ACI design principles would generalize to other domains and how ACI could be implemented more broadly.

**Questions:**

How might the principles of ACI design extend to other domains beyond software engineering? Are there specific challenges you anticipate in adapting this approach to different types of tasks?

**Limitations:**

Limitations are thoroughly discussed

---

> ### Author Rebuttal · Authors · 2024-08-07
>
> Thank you for your thoughtful comments, all of which are very helpful for improving our work! We greatly appreciate your conclusion that ACI is a novel and important concept.
>
> **Q1, W1: Generalizability, Applying ACI to Other Domains**
>
> > Building a good ACI for a specific domain is tough, as we showed in the paper, and so here our focus was on the software engineering domain.
> Our decision to focus on coding is due to
> (1) Its practicality and popularity in current LM agent work (Sections [1,5]).
> (2) Software engineering as a task (e.g. SWE-bench) has a lot of room for improvement compared to other agentic settings, where performance is higher. Before SWE-agent, SWE-bench SOTA was 1.96%, and SWE-agent took that up to 12.5%.
>
> > Adapting an ACI to different domains is a very interesting research direction we hope the community is inspired to pursue.
> For example, when considering a good ACI for web navigation, similar to our design process for SWE-agent’s localization tools, it may be worth considering the effect of redesigning search to be more agent friendly.
> For instance, instead of Google-style search results, could agents benefit from additional information beyond a list of each result’s title and snippet?
> Would the design vary with the specific nature of the downstream digital task (e.g. shopping on the web vs. searching a company’s internal knowledge base)?
>
> > With future works, we hope that the study of ACI might progress similarly to HCI - with more ACI-oriented explorations in different domains, we’ll better understand how to design interactive task environments conducive to downstream performance.
> We have expanded **Appendix E.3** with more content discussing this question.
>
> We greatly appreciate this question for the opportunity to expand on such thoughts! Thanks again for your time and consideration.

---

> > ### Comment · Reviewer_KbN4 · 2024-08-12
> > **Response to authors**
> >
> > Thank you for the response and discussion, I think the work is interesting. I will keep the score.

---

### Official Review · Reviewer_BuFF · 2024-07-14

**Soundness:** 4
**Presentation:** 4
**Contribution:** 3
**Rating:** 8
**Confidence:** 4

**Summary:**

The paper introduces SWE-agent, a system designed to enable language model (LM) agents to autonomously perform software engineering tasks through a custom agent-computer interface (ACI). The study posits that LM agents can benefit from interfaces tailored to their specific needs, similar to how human software engineers use integrated development environments (IDEs). The SWE-agent system demonstrates significant improvements in task performance on the SWE-bench and HumanEvalFix benchmarks compared to non-interactive LMs, achieving state-of-the-art results.

**Strengths:**

**Novel Concept:** The introduction of the ACI tailored specifically for LM agents represents a significant innovation, addressing the unique needs of LMs in software engineering tasks.

**Empirical Results:** The system achieves impressive performance metrics on established benchmarks, indicating the effectiveness of the ACI design.

**Comprehensive Evaluation:** The paper includes a thorough analysis of the system's performance, including ablation studies and comparisons with baseline models.

**Open-Source Contribution:** The authors provide anonymized code and data, contributing to the research community and facilitating reproducibility.

**Weaknesses:**

**Complexity and Generalizability:** The system's reliance on specific design choices and configurations might limit its generalizability to other types of LMs or software engineering tasks.

**Overhead and Efficiency:** While the ACI improves performance, it introduces additional layers of complexity and potential overhead that are not fully quantified in terms of computational resources and efficiency.

**Questions:**

- What are the specific computational and resource overheads introduced by the ACI, and how do they impact overall system efficiency?
- How generalizable are the ACI design principles to other domains beyond software engineering, and what modifications would be necessary?
- Can the ACI be adapted for use with other language models with smaller context windows, and what would be the expected impact on performance?

---

> ### Author Rebuttal · Authors · 2024-08-07
>
> Thank you so much for your interest in our research. We greatly appreciate your feedback and insights.
> We’re especially happy that you see the novelty of the concept of ACI and the potential to impact future work in LM agents.
>
> We’ve tried to address your particular concerns below:
>
> **What’s the overhead introduced by the ACI / SWE-agent?**
>
> > Safely running SWE-agent interactively, requires using a containerized environment, which will considerably increase the CPU resources required compared to most non-interactive systems.
> Additionally, since SWE-agent provides an interactive environment for agents solving problems, each inference call for generating an action cumulatively consumes more GPU usage than typical non-interactive approaches.
>
> > However, as was demonstrated in SWE-bench, even using a near-perfect retrieval system (the oracle retriever in SWE-bench), non-interactive performance may have a lower practical upper-bound compared to interactive systems that can provide real-time execution and testing feedback.
>
> > Thus, at face-value, SWE-agent is a more expensive approach on both the CPU and GPU resources counts, but provided the increase in performance on SWE tasks, it is unclear what computational resources may be necessary for a non-interactive system to achieve comparable performance.
>
> **How generalizable are the ACI design principles?**
>
> > The short version of our design principles for the ACI is that interfaces should be: simple, efficient, concise, and help models recover from mistakes. While the SWE-agent ACI was built with the software development task in mind, the general design principles on the ACI should have a lot of transferability to other tasks and domains.
>
> > In SWE-agent, we show that despite LMs having been trained extensively on bash commands and scripts, they are less effective at using shells directly, compared to our simplified ACI which it had never seen before. Thus, we think that many of the lessons learned from the SWE-agent ACI design process can be applied to other domains and researchers building ACIs for different tasks.
> Based on your feedback, we’ve added some discussion about this in Appendix E.
>
> **Can the ACI be adapted for LMs with smaller context windows?**
>
> > For shorter context LMs, the ideal ACI would likely be slightly different than the default chosen for SWE-agent. At each step, we would need to minimize the number of tokens exchanged. The simplest approach to doing this would involve reducing the size of the fileviewer and performing more context compression as the interaction trajectory grows longer. We might expect a shorter context LM to still achieve non-trivial, albeit lower, performance on SWE-bench Lite, assuming that the LM still has the strong reasoning abilities of frontier models like GPT-4 or Claude.
>
> Thank you again for your valuable feedback.

---

### Official Review · Reviewer_N4zh · 2024-07-15

**Soundness:** 3
**Presentation:** 3
**Contribution:** 3
**Rating:** 6
**Confidence:** 3

**Summary:**

The paper presents SWE-agent, a system designed to enhance language model (LM) agents' performance in software engineering tasks through a specialized agent-computer interface (ACI). The ACI allows LMs to efficiently navigate, edit, and execute code within repositories, significantly improving performance over traditional non-interactive LMs. SWE-agent demonstrates good results on benchmarks such as SWE-bench and HumanEvalFix, showcasing a substantial increase in task resolution rates. The authors highlight the importance of designing interfaces tailored specifically for LMs, drawing parallels to human-computer interaction studies. The paper provides detailed experimental results and ablation studies to validate the efficacy of the ACI.

**Strengths:**

The key idea of designing a specialized interface for LMs to interact with software environments is novel and well-executed. By abstracting and simplifying the interaction process, the authors address a significant gap in current LM capabilities. The experimental validation is thorough, with SWE-agent outperforming existing systems on established benchmarks, showing the practical significance of the approach. The writing is clear, making complex ideas accessible, and the data contribution is valuable, with detailed ablation studies shedding light on the importance of various design choices. Additionally, the open-sourcing of SWE-agent and its dataset contributions provide a useful resource for the research community, encouraging further exploration and validation of the proposed approach.

**Weaknesses:**

1. The process of managing context history within the ACI is not clearly explained, leaving ambiguity about how the system maintains essential information while avoiding unnecessary context.
2. The method for optimizing the ACI configuration through manual inspection and grid search lacks transparency and could be elaborated further.
3. The system's error recovery mechanisms, while improved with linting, may still struggle with repeated failed edits, indicating a potential area for further enhancement.
4. The ACI seems to only support a limited number of tools -- how to expand it to other more advanced and useful tools are unclear.

**Questions:**

1. How does the ACI manage context history to maintain essential information while avoiding unnecessary context? Note that the context length would be a bottleneck for many open-source LLMs.
2. How does the system handle scenarios where the LM encounters repeated failed edits despite the linter's intervention?
3. It seems that SWE-Agent is not a SOTA framework for SWE-bench. I am curious what the authors think about the recent SOTA approaches, and what would be the most valuable and promising directions for the next-gen of SWE-agent?

**Limitations:**

See above.

---

> ### Author Rebuttal · Authors · 2024-08-07
>
> Thank you so much for your time and consideration. You’ve brought up some excellent points in your feedback that we try to address below.
>
> Regarding your questions / weaknesses:
>
> **W1, Q1: How does the ACI manage context?**
>
> > The main mechanism by which SWE-agent manages to keep memory short, is by “collapsing” old environment responses into a placeholder message, keeping most of the visible history at any given time either related to the system setup and initial problem, or to recent environment observations. For example, if the agent has had 20 turns of interaction with the computer, the content of the first 15 environment responses will not be shown in full, while the last 5 observations will be shown in their entirety. This practice keeps the total context used for each turn shorter and cheaper.
>
> > We’ve clarified the language regarding the ACI context management and expanded discussion around Figure 9, which shows a diagram of system inputs and outputs. We’ve also added a figure showing the exact contents of a “collapsed” environment response, and now indicate this in Section 3 of the main paper.
>
> **W2: The ACI configuration choices lack transparency**
>
> > Due to the time and expense of developing on state-of-the-art proprietary models, such as GPT-4 (1106) and Claude 2, and the complexity of designing ACIs for challenging tasks, such as software engineering, the development process involved substantial qualitative analysis based on manual evaluations on the SWE-bench development split. We provide extensive ablations, showing the individual impact of many design decisions. We’ve expanded Section A.2 to provide further discussion on this process.
>
> **W3, Q2: How does the system manage repeated failed edits?**
>
> > This is a good insight, and currently, the system does not specially handle repeated failed edits beyond automatic linting. However, your suggestion is sound and could have clear benefits for future research on SWE agents.
>
> **W4, Q3: How can SWE-agent and ACIs improve further, provided results since submission?**
>
> > At the time of submission, SWE-agent was the SOTA system for approaching SWE-bench tasks. Since then, we’ve seen numerous papers design their own ACI’s with new tools, building on SWE-agent’s design philosophy. While your suggestions and incorporating the improvements proposed by others is definitely possible for further improving SWE-agent, we see the primary contribution of this paper to be the introduction of the ACI concept and our design principles. Future research has great potential to improve further upon these principles and the design of interfaces, tools, and model designs for agents.
>
> We appreciate your feedback and we’ve made sure to incorporate your points to refine our paper. Thank you so much for your time and effort.

---

### Comment · Area_Chair_fCuB · 2024-08-14
**Reviewer BuFF & N4zh**

Please respond to the authors at your earliest opportunity. Today is the last day to engage.

---

### Decision · Program_Chairs · 2024-09-25

**Decision:**

Accept (poster)

**Comment:**

This paper proposes a new method, SWE-agent, which is built upon a large language model to help with software engineering tasks. The paper develops an agent-computer interface (ACI) to help facilitate interaction with a coding environment. The paper evaluates SWE-agent on two benchmarks, showing positive results.

The reviewers were all positive. One reviewer raised a score after the rebuttal while others did not. Some concerns remain regarding the over-focus on SWE Bench, as it calls into question the generalizability of ACI. Further, the proposed method could have been studied more exhaustively with better/additional ablation studies. However, the reviewers appreciated the importance of the problem and the potential impact of the paper.